# Modeling Land Use and Land Cover Change: Using a Hindcast to Estimate Economic Parameters in gcamland v2.0

Katherine V. Calvin[1], Abigail Snyder[1], Xin Zhao[1], Marshall Wise[1]

[1]Joint Global Change Research Institute, Pacific Northwest National Laboratory, College Park, MD, 20740, USA

*Correspondence to*: Katherine V. Calvin (katherine.calvin@pnnl.gov)

**Abstract.** Future changes in land use and cover have important implications for agriculture, energy, water use, and climate. Estimates of future land use and land cover differ significantly across economic models as a result of differences in drivers, model structure, and model parameters; however, these models often rely on heuristics to determine model parameters. In this study, we demonstrate a more systematic and empirically-based approach to estimating a few key parameters for an economic

model of land use and land cover change, gcamland. Specifically, we generate a large set of model parameter perturbations for the selected parameters and run gcamland simulations with these parameter sets over the historical period in the United States to quantify land use and land cover, determine how well the model reproduces observations, and identify parameter combinations that best replicate observations, assuming other model parameters are fixed. We also test alternate methods for forming expectations about uncertain crop yields and prices, including adaptive, perfect, linear, and hybrid approaches. In

particular, we estimate parameters for six parameters used in the formation of expectations and three of seven logit exponents for the USA only. We find that an adaptive expectation approach minimizes the error between simulated outputs and observations, with parameters that suggest that for most crops, landowners put a significant weight on previous information. Interestingly, for corn, where ethanol policies have led to a rapid growth in demand, the resulting parameters show that a larger weight is placed on more recent information. We examine the change in model parameters as the metric of model error changes,

finding that the measure of model fitness affects the choice of parameter sets. Finally, we discuss how the methodology and results used in this study could be used for other regions or economic models to improve projections of future land use and land cover change.

## 1 Introduction

Between 1961 and 2015, global agricultural production has increased substantially, including more than a tripling of wheat

production, a five-fold increase in maize production, and a twelve-fold increase in soybean production (FAO, 2020b). Agricultural area has increased, but by a smaller amount (10% increase in harvested area for wheat, 180% increase for maize, five-fold increase for soybeans), due to increases in agricultural productivity (FAO, 2020b). Total global cropland area has increased by 15% between 1960 and 2015, from 1377 million hectares (Mha) to 1591 Mha (Goldewijk et al., 2017). These changes have resulted in changes in natural land area, including declines in global forest area (Hurtt et al., 2020).

In the United States, crop production has increased substantially in the last several decades, but much of that increase in production is due to increases in yields (Babcock, 2015; Fuglie, 2010). Total cropland area in the United States has remained relatively constant between 1975 and 2015. Instead, there has been a shift in crop distribution, with an increasing share of corn and soybeans and a decreasing share of wheat and other grains (Figure 1, (FAO, 2020a; Taheripour and Tyner, 2013)).


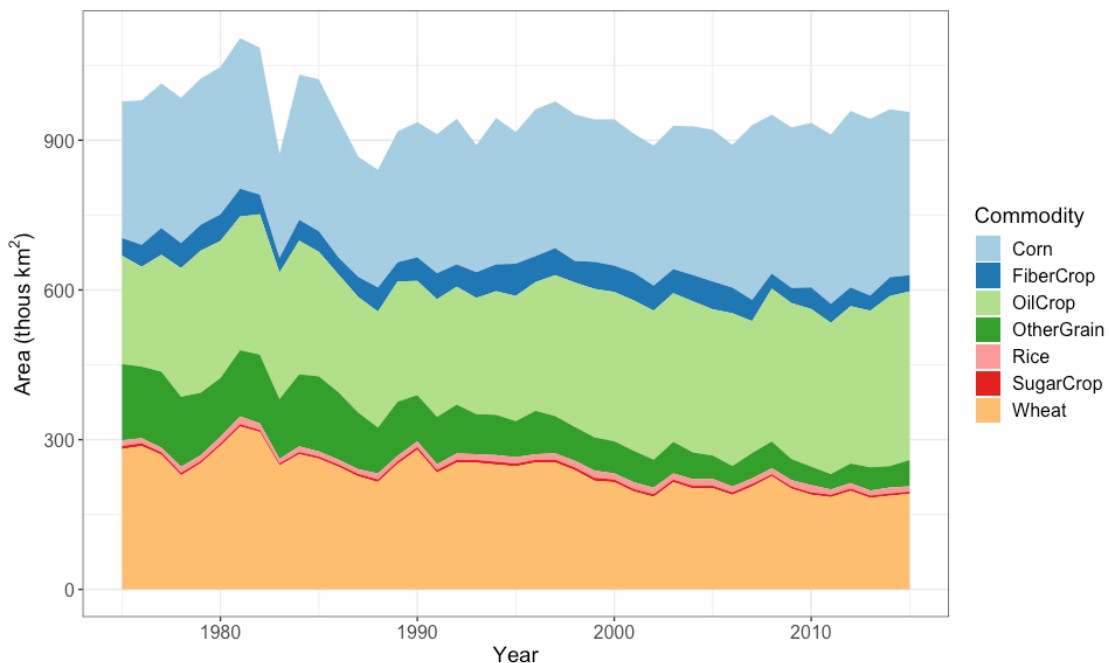

**Figure 1: Harvested area by crop for major commodities in the United States (1975-2015). Source: USDA.**

Future changes in land use and land cover have implications for agricultural production, energy production, water use, and climate. For example, changes in land cover can alter albedo, resulting in changes in local and global temperature and precipitation (Brovkin et al., 2013; Jones et al., 2013; Manoli et al., 2018). Similarly, changes in land use and land cover have implications for water withdrawals and water scarcity (Bonsch et al., 2016; Chaturvedi et al., 2013; Hejazi et al., 2014a, 2014b; Mouratiadou et al., 2016). However, there is significant uncertainty in the future evolution of land use and land cover, due to uncertainties in future socioeconomic conditions (e.g., population, income, diet) (Popp et al., 2017; Stehfest et al., 2019), technological change (Popp et al., 2017; Tilman et al., 2011; Wise et al., 2014), climate (Calvin et al., 2020; Nelson et al., 2014), and incentives for bioenergy, afforestation, reforestation (Calvin et al., 2014; Hasegawa et al., 2020; Popp et al., 2014, 2017).

Economic models are widely used to estimate future agricultural production and land use, and estimates of future land use and land cover also differ significantly across such models (Alexander et al., 2017; Von Lampe et al., 2014; Popp et al., 2017).

These models use economic equilibrium, statistical, agent-based, machine learning, and hybrid approaches (Engström et al., 2016; National Research Council, 2014). Even within each category, there are differences across models, both in terms of structure and parameters. For example, among economic equilibrium models of land use change (the approach most commonly used in integrated energy-water-land-climate models), some models use constrained optimization (e.g., GLOBIOM), while other models use a non-linear market equilibrium approach (e.g., GCAM) (Wise et al., 2014).


    Efforts to evaluate land use models over the historical period are limited. Baldos and Hertel (2013) compare the net change in cropland area, agricultural production, average crop yield, and crop price between 1961 and 2006 simulated by the SIMPLE model to observed changes. Their model matches observations better at the global scale than at the regional scale; additionally, they find that "even knowing yields with certainty does not allow us to predict cropland change accurately over this historical

period." Bonsch et al. (2013) compare simulated land-use change $CO_2$ emissions from MAgPIE to observations, finding that the choice of observation dataset matters for how well the model performs. Calvin et al. (2017) and Snyder et al. (2017) compare agricultural production and land area simulated by the GCAM model to observations, finding that the model does better for trends than annual values and that some region/crop combinations are better than others. The authors test the use of expectations about yield using a linear forecast as a driver of land use change instead of observed yield, finding that simulations

using expected yield better match observations than those using observed yield. Engstrom et al. (2016) use a Monte Carlo approach to sample parameters in PLUM, simulating agricultural production and land area over the historical period and comparing results to observations. The authors find the model performs better at larger regional aggregations, but the observed grassland and cereal land area falls outside the full range of their ensemble results. However, most land use models outside of these have not used historical simulations for evaluation/validation.


    Only a few studies have attempted to draw land use modeling parameters from econometric estimates of land supply elasticity (Ahmed et al., 2009; Lubowski et al., 2008). However, there is usually no fixed relationship between the land supply elasticities and land use modeling parameters in equilibrium models (Zhao et al., 2020a) and, more importantly, empirically estimated elasticities only provide a limited coverage of regions and land use categories (Barr et al., 2011; Lubowski et al., 2008). Thus,

the parameters used in land use models are often based on heuristics (Schmitz et al., 2014). For example, Taheripour and Tyner (2013) group regions into four categories based on historical land use change and assign substitution parameters based on those categories. Wise et al. (2014) choose model parameters to replicate empirically estimated parameters; however, there is no unique mapping between the empirical parameter (constant elasticity of land transformation) and the model parameter (logit exponent). While there are many examples of studies exploring sensitivity to drivers of land use change or sensitivity across

models, most studies exclude sensitivity to parameters. The small number of studies that do test alternative parameters find that it could significantly alter land use change (Engström et al., 2016; Taheripour and Tyner, 2013; Zhao et al., 2020b).

In this paper, we advance the science on parameterizing land use models by using hindcast simulations and statistical approaches rather than the heuristic approaches described in the previous paragraph. Specifically, we use a large perturbed parameter ensemble and a sensitivity analysis over different model structural assumptions to determine the model expectation configuration and parameter set that best replicate observed historical land use and land cover within the United States. Section 2 describes the methodology used in this study. The primary results and sensitivity analyses are discussed in Sections 3 and 4, respectively. Section 5 includes the discussion and conclusions.

## 2 Methodology

In this paper, we run hindcast simulations using gcamland to select the model parameters that best reproduce observations under different model specifications. The steps implemented are is as follows (see also Figure 2):

1. Sample Parameters: Using Latin Hypercube Sampling, randomly select a set of parameters from uniform distributions (see Section 2.2.1).

2. Run gcamland ensemble: Land allocation in the United States for the whole time period is estimated by running gcamland over the historical period (i.e., as a hindcast simulation) with each set of randomly chosen parameters (see Section 2.1 for a description of gcamland and Section 2.2.2 for a description of the ensemble).

3. Compare to Observations: Calculate a variety of metrics of goodness of fit from simulated land allocation from gcamland and observations of land allocation (see Section 2.2.3).

4. Select best parameters per expectation type: Determine the "best" set of parameters by choosing the set that optimizes a given goodness of fit metric for each expectation type (see Section 2.2.4).

5. Select overall best model: Select expectation type and parameter set combination that optimizes a given goodness of fit metric across all expectation types (see Section 2.2.5).

6. Repeat Steps 1 through 6 for different model specifications (see Section 2.2.6).

Section 2.1 describes gcamland, including its economic and mathematical approach to modelling land use and land cover. Section 2.2 describes each of the steps above in turn.

**Figure 2: Schematic depicting the overall methodology used in this paper. The Latin hypercube sampling is used to sample nine different parameters, but displayed in the left panel of this schematic as a two-parameter example.**

## 2.1 Land use modelling

### 2.1.1 gcamland

We use the gcamland v2.0 software package in this study (Calvin et al., 2019a). gcamland separates the land allocation mechanism in GCAM (Calvin et al., 2019b) into an R package.[1] The model calculates land allocation over time; changes in land use and land cover are driven by changes in commodity prices, yields, costs, and subsidies, all of which are inputs into gcamland. gcamland includes all land use and land cover types, with crops aggregated into 12 commodity groups[2] (see Table S1 for a mapping). gcamland can be run in several different modes, including hindcast and future scenario options and single and multiple ensemble options. For this paper, we utilize the ensemble and hindcast options, generating large ensembles of hindcast simulations (see Section 2.2.2). gcamland can be run for any of the 32 geopolitical regions within GCAM, but for this study we focus on the United States.

---

[1] GCAM and gcamland are separate models. While gcamland replicates the land allocation mechanism in GCAM, it is not run within GCAM. Similarly, GCAM is not run as a part of gcamland. gcamland only includes a representation of land allocation. GCAM includes representations of agricultural supply and demand, land allocation, and other sectors (energy, water, economy, climate). The land allocation mechanism within gcamland uses price, yield, cost, subsidy, logit exponents, expectation parameters, and initial land area as exogenous inputs and endogenously determines land area in subsequent years. Changes in demand are explicitly represented in GCAM. In gcamland, changes in demand are captured through changes in price. For example, the increase in demand for corn and soybean due to biofuels policy is captured through changes in the prices of these goods.

[2] gcamland technically includes a 13th crop (biomass) which represents lignocellulosic energy crops (e.g., switchgrass and miscanthus). However, since these were not grown at commercial scale in the historical period, its land area is zero in the simulations described in this paper.

## 2.1.2 Economic approach in gcamland

Land allocation in gcamland (and GCAM) is determined based on relative profitability, using a nested logit approach (McFadden, 1981; Sands, 2003; Wise et al., 2014). The logit land supply is presented in equation (1). All else equal, an increase in the rental profit rate ($r_i$) of one land type will result in an increase in the land area ($X_i$) allocated to that land type. The magnitude of the land supply response is dependent on the positive logit exponent ($\rho$) and share-weight parameters ($\lambda_i$). These parameters influence the land supply elasticity, which is non-constant (i.e., it varies depending on the relative profitability as described in Wise et al., 2014). $Y$ is the total land supply, i.e., $\sum_i X_i = Y$. The logit formulation assumes that there is a distribution of profit rates for each land type, and the resulting land allocation for a given land type is the probability that land type has the highest profit (Zhao et al., 2020b). The logit share-weights (the scale parameters in the distribution) are calculated to perfectly reproduce the data in a base year. The logit exponent (the shape parameter in the distribution which governs the magnitude of land transformation given relative profit shocks) is one of the parameters of interest in our study (see Section 2.2, Table S2).

$$X_i = \frac{(\lambda_i r_i)^\rho}{\sum_j (\lambda_j r_j)^\rho} \cdot Y \,, \tag{1}$$

The logit approach is advantageous compared with the constant elasticity of transformation (CET) approach widely used in Computable General Equilibrium (CGE) models as it can directly provide traceable physical land transformation. But like the CET function, the logit land sharing function is parsimonious and a nested structure can be used. In gcamland, all crops are nested under cropland. Cropland is nested with forest and then pasture, see Figure S1. In a nested logit, the area of a particular land type is determined by not just the logit of its nest, but also by the logit of the nests above that. Thus, there are three logit exponent parameters governing land transformation for crops in gcamland. In the nested version, land allocation at each of these nests is determined by equation 2 (a modified version of equation 1, where Y is replaced by the land allocated to that particular nest). The land allocated to a particular nest is dynamic and varies over time. In equation 2, dynamic variables are indicated with subscript *t*.

$$\begin{cases} X_{jt}^C = \dfrac{\left(\lambda_j^C r_{jt}^C\right)^{\rho^C}}{\sum_j \left(\lambda_j^C r_{jt}^C\right)^{\rho^C}} \cdot Y_t^C & for \ \text{C the cropland nest including all crops and other arable land (see footnote 6)} \\[2em] X_{jt}^A = \dfrac{\left(\lambda_{jo}^A r_{jt}^A\right)^{\rho^A}}{\sum_j \left(\lambda_j^A r_{jt}^A\right)^{\rho^A}} \cdot Y_t^A & for \ \text{A the ag, forest and other nest} \\[2em] X_{jt}^R = \dfrac{\left(\lambda_j^R r_{jt}^R\right)^{\rho^R}}{\sum_j \left(\lambda_j^R r_{jt}^R\right)^{\rho^R}} \cdot Y_t^R & for \ \text{R the gcamland dynamic modeling nest} \end{cases} \tag{2}$$

Profit rates ($r_j$) at the lowest level of the nest are computed based on price, cost, yield, and subsidy (if included) for commercial land types (crops, pasture, commercial forest); profit rates for non-commercial land types are input into the model and are based on the value of land (see also Table S1). Profit rates for commercial lands evolve over time as price, cost, yield, and

subsidy change. Profit rates for non-commercial lands are constant over time. The logit approach effectively depicts a supply curve for non-commercial land with the land supply elasticity implicitly determined by the logit exponent and the assumed rental profit rates (i.e., implying a cost of land transition). The supply curve approach, which views the amount of land available as endogenous, offers more modelling flexibilities with traceable results compared to approaches of assuming non-commercial lands to be inaccessible and fixed over time or aggregating non-commercial lands with commercial lands (Dixon et al., 2016). Profit rates for higher levels of the nest ($r_{node}$) are determined by:

$$r_{node} = \left[ \sum_{j=1}^{n} (\lambda_j r_j)^\rho \right]^{1/\rho}. \tag{3}$$

gcamland tracks both physical area and harvested area for crops. Physical area is determined by the logit-based land allocation scheme described in this section. Harvested area is calculated using physical area and a fixed harvested-to-physical area ratio, estimated in the base year, and held constant in the future. Note that, since forestland is not an annually planted and harvested commodity, GCAM, gcamland, and other similar models assume that land must be set aside at every timestep to ensure enough commercial forestland is available to meet harvest demand at the time the forest matures. To do this in gcamland, we assume that the amount of land allocated to forest depends on the harvest yield and the rotation length.

### 2.1.3 Means of forming expectations

There are multiple means of forming expectations in the literature. With perfect foresight, the expected value of a given variable is equal to its realized value:

$$\mathbb{E} x_t = x_t. \tag{4}$$

In an adaptive expectation approach (Nerlove, 1958), the expected value is a linear combination of the previous expectation and the new information acquired, with $\alpha$ being the coefficient of expectations:

$$\mathbb{E} x_t = (1-\alpha) x_{t-1} + \alpha \mathbb{E} x_{t-1}. \tag{5}$$

Finally, a linear expectation approach uses a linear extrapolation of previous information to form the expectation:

$$\mathbb{E} x_t = \frac{Cov[\boldsymbol{x}(n), \boldsymbol{year}(n)]}{Var[\boldsymbol{year}(n)]} year_t, \tag{6}$$

where $n$ is a fixed number of previous years considered in forming the expectation, $\boldsymbol{x}(n)$ and $\boldsymbol{year}(n)$ are vectors of the variable and year index, respectively, with historical information from year $t-1$ to $t-n$. That is, instead of using all available historical information, forward-looking producers are assumed to rely on only information of the most recent $n$ years.

In our study, we combine these basic approaches into four different expectation types, specifying the means of calculating expected price and expected yield (Table 1).[3] The expected prices and yield would affect farmers' expected rental profits and, thus, land use decisions. Note that most previous studies only include price expectations. We also include yield expectations, which is important in explaining landowner's behavior and supply responses (Roberts and Schlenker, 2013).

180

**Table 1: Expectation types tested in this study.**

| Expectation type | Price expectations | Yield expectations | Examples in the literature |
|---|---|---|---|
| Perfect | Perfect expectations | Perfect expectations | All integrated models and most agriculture economic models |
| Adaptive | Adaptive expectations | Adaptive expectations | (Féménia and Gohin, 2011; Lundberg et al., 2015; Mitra and Boussard, 2012) |
| Linear | Linear expectations | Linear expectations | (Calvin et al., 2017; Snyder et al., 2017) |
| Hybrid Linear Adaptive | Adaptive expectations | Linear expectations | Tested in this paper |

### 2.1.4 Initialization data

To initialize gcamland in this study, we started from the GCAM v4.3 agriculture and land use input data (see Table S1). The

185    GCAM data processing reconciles land use data from FAO with land cover data, ensuring that total areas do not exceed the amount of land in a region. Thus, we chose to use this reconciled data instead of using FAO data directly. We have made two changes to the GCAM v4.3 initialization data.

First, since GCAM has a five-year time step, it uses five-year averages of land use and agricultural production for initialization.

190    For this study, we have updated the input data to remove the averaging since we are primarily focused on annual time steps in gcamland; that is, the initialization data in gcamland for a particular year is the data for that year only and not a five-year average around that year as it is in GCAM.

Second, GCAM models land use and land cover at the subnational level (v4.3 used Agro-Ecological Zones; v5.1 and

195    subsequent versions use water basins). However, much of the comparison data is provided at national level. For this study, we aggregate the initialization data to the national level, representing the USA as a single region. The qualitative insights in this paper would not change if we disaggregated to subnational level, but the exact quantitative results would.

---

[3] Note that other expectation types can be tested within gcamland, e.g., expectation types that are a hybrid of past and perfect information. Such expectations types can be useful for understanding the value of additional information. However, we exclude them in this paper as they are unlikely to explain past behavior and are not covered in the literature on land use decision making.

Third, GCAM uses constant costs over time. For this study, we have updated the costs to use time-evolving cost data (see next section).

### 2.1.5 Scenario data

We use data for producer price and yield from the U.N. Food and Agricultural Organization (FAO, 2018a, 2018b, 2020b), with data available for all non-fodder commodities for 1961-2018. Data was aggregated from individual crops to the GCAM/gcamland commodity groups, weighting non-fodder crops by their production quantity. In some cases, data prior to 1961 are required to generate expectations for the model years (1975-2015); in these cases, we assume that prices and yields prior to 1961 are held constant at their 1961 values. For cost, we use data provided by the U.S. Department of Agriculture (USDA, 2020a), with data available for major crops from 1975-2018. We only include the variable costs as reported by USDA and exclude the allocated overhead costs. We use a representative crop from USDA for each GCAM/gcamland commodity group, as data does not exist for all crops (i.e., we use soybean cost from USDA as a proxy for the cost of OilCrop in gcamland). The producer prices used in gcamland are defined as "prices received by farmers…at the point of initial sale" or "prices paid at the farm-gate" (FAO, 2018a) and thus do not reflect subsidies. However, subsidies are a reality of crop agriculture in the United States. However, there are not continuous, complete, and consistent data sets for all types of subsidies paid to farmers. Additionally, crop-specific information (of the type needed for gcamland) is only available for direct payments, making the inclusion of other types of subsidies difficult.  Therefore, for subsidies, we combine two different data sets from USDA: the Federal government direct payments (USDA, 2020c) and the farm business income (USDA, 2020b). We only include direct payments from these two reports; thus, our subsidy data is missing many other forms of payment. Additionally, we only have data for a subset of crops and the categories reported change over time across the two data sets. Because this data is inconsistent and incomplete, we only use it as a sensitivity in this paper and do not include it in the primary analysis.[4]

### 2.2 Using ensembles to estimate gcamland parameters

#### 2.2.1 Parameter samples

In total, gcamland has between 29 and 35 parameters (depending on the expectation type) that are used to calculate land allocation in each year (see equation 2). This study samples all six parameters used in the expectation calculation and three of the seven logit exponents. The remaining four logit exponents are specified exogenously, as these exponents have minimal impact on the outcomes of interest in this paper (see Section S1 and Table S2). The values of those four logit exponents have not been obtained from an explicit statistical analysis and instead were selected based on authors' judgment (see Section S1). The remaining 22 parameters are share-weight parameters ($\lambda_i$ in equations 1 and 2). These parameters are calculated from the

---

[4] Note that our choice to use it as a sensitivity and not the default is because it does not improve NRMSE and did not alter the parameter set that minimized NRMSE between simulated and observed land allocation (as discussed in Section 4).

observed land allocation in the initial model year and the other specified parameters to ensure that land allocation in the initial year exactly matches observations.

Within this study, we vary a total of nine parameters (Table 2), including three logit exponents, the coefficient of expectations ($\alpha$) for the *Adaptive* expectation and the number of years ($n$) used in the *Linear* expectation. In addition, we allow $\alpha$ and $n$ to vary across commodity groups, resulting in three separate realizations for each parameter. We group the commodities to minimize the number of free parameters. The first group includes Corn and OilCrop, which are used for biofuels in the United States and have had shifts in the demand over time as a result of biofuels policies. The second group includes the other two

large commodities produced in the United States, Wheat and OtherGrain. The third group includes all other crops. The range of values spanned in the ensemble was chosen to cover all plausible values of each parameter but avoid potential numerical instabilities. Those ranges and their justification are described in Table 2. We use a Latin Hypercube Sampling[5] strategy to generate the ensembles, with 10,000 ensemble members per expectation type and model configuration. Latin Hypercube Sampling draws all nine parameters simultaneously from uniform distributions.


**Table 2: Parameters perturbed in this study, including the range of values tested.**

| Type | Parameter | Description | Range | Rationale for Range |
|---|---|---|---|---|
| Logit | Dynamic Land | Logit exponent ($\rho^R$) dictating competition between the "ag, forest, and other" and pastureland nests, which include all dynamic land types within gcamland[6] | 0.01-3 | The minimum value is chosen to be close to zero (which would result in no shifts in land) but without causing numerical instability. Very large logit exponents result in winner-take-all behavior (Wise et al., 2014; Zhao et al., 2020b). Such behavior may be reasonable at a small scale but not for the United States as a whole, so an upper bound of 3 is chosen to prevent this. |
| | Ag, Forest, and Other | Logit exponent ($\rho^A$) dictating competition between cropland, forestland, and grass/shrubs | | |

---

[5] We use the R `lhs` package for the sampling (Carnell, 2020; R Core Team, 2020).

[6] A small amount of land (~4%) is considered unsuitable for cropland, pasture, or other vegetation expansion in gcamland in the United States, including urban, tundra, rock, ice, and desert (Table S1). This land is held constant throughout the simulation time period by setting the logit exponent dictating competition between these land types to zero (Table S2). Such a parameterization means that no cropland can be converted to urban, rock/ice/desert or tundra and no urban, rock/ice/desert or tundra can be converted to cropland.

| | | | | |
|---|---|---|---|---|
| | Cropland | Logit exponent ($\rho^C$) dictating competition among crops[7] | | |
| Share of Past Information | Corn, OilCrop | Weight on previous expectations ($\alpha$) for Corn and OilCrop in the *Adaptive* expectations | 0.1-0.99 | Parameter is restricted to the range [0, 1]. A value of 1 would keep expected profit constant at its initial value, so we choose a value slightly smaller for the upper bound. Very small values of this parameter have been shown to result in divergence of the system (Féménia and Gohin, 2011).[8] A lower bound of 0.1 is chosen to prevent this. |
| | Wheat, OtherGrain | Weight on previous expectations ($\alpha$) for Wheat and OtherGrain in the *Adaptive* expectations | | |
| | All Other Crops | Weight on previous expectations ($\alpha$) for all other crops (see footnote 6, Figure S1, or Table S1 for a full list of crop categories) in the *Adaptive* expectations | | |
| Number of years | Corn, OilCrop | Number of previous years (n) used in the linear extrapolation in the *Linear* expectations for Corn and OilCrop | 2-25 | Linear extrapolation is undefined for values less than 2. Only integer values allowed. |
| | Wheat, OtherGrain | Number of previous years (n) used in the linear extrapolation in the *Linear* expectations for Wheat and OtherGrain | | |

---

[7] gcamland includes twelve crop categories (Corn, FiberCrop, FodderGrass, FodderHerb, MisCrop, OilCrop, OtherGrain, PalmFruit, Rice, Root_Tuber, SugarCrop, Wheat). In addition, other arable land (which includes fallow and idled cropland) is included in this nest.

[8] Note that Femenia and Gohin (2011) define their parameters differently than is done in this paper. Thus, an $\alpha$ value of 1 in their study is equivalent to a value of 0 here.

| All Other Crops | Number of previous years (n) used in the linear extrapolation in the *Linear* expectations for all other crops (see footnote 6, Figure S1, or Table S1 for a full list of crop categories) | | |
| --- | --- | --- | --- |

## 2.2.2 Running the gcamland hindcast ensemble

Hindcast simulations are experiments where a model simulation is conducted for a time period in which observational data are

available but in which the observational data are specifically not used in the model simulation. In the example of gcamland running a hindcast from 1990-2015, this would correspond to a gcamland forecast of land allocation from 1990-2015. When 1990 is used as the initial model year, observed data from 1991-2015 are not used at any point in the gcamland simulation of land allocation, and 1990 observed data is only used to initialize gcamland.

We use each of the 10,000 parameter sets to run a gcamland hindcast for each of the four expectation types described, resulting in 40,000 simulations. Each parameter set includes 9 parameters (see Table 2); the three logit parameters are used for all expectation types, but the expectation parameters are only used in expectation types requiring them (e.g., perfect expectations only uses the three logit exponents; the hybrid linear-adaptive expectation type uses all nine parameters).

## 2.2.3 Comparing to observations

### 2.2.3.1 Observation data

We compare model outputs to observation data to evaluate the performance of gcamland under each expectation type and parameter set. Ideally, the observation data would be completely independent of the model. However, due to limited availability of data sets,[9] we use the FAO harvested area for crops as the observational dataset, despite the fact that it is used to calculate the initial model year land allocation in gcamland. Only a single year of data is used for this initialization, so the comparison

to the FAO time series is still valid (Sections 2.1.4 and 2.2.1 for more details). FAO includes harvested area for the entire time series considered in this paper (1975-2015) for most crops; however, FAO does not have a full time series of harvested area for fodder crops so we exclude it from our error calculation. For land cover, an independent data set is available for use in

---

[9] The only other data set we are aware of the provides a time series of cropland area by crop is the USDA. However, since FAO basis their reporting for the United States on submissions from the USDA, these two datasets are identical.

gcamland; specifically, we use satellite data from the European Space Agency (ESA) Climate Change Initiative (CCI), as reported by the FAO (FAO, 2020a) and aggregated to the gcamland land cover classes. Due to differences in definitions and

classifications, the grassland and shrubland reported by CCI differ substantially from the gcamland areas even in the initial model year. Additionally, CCI data is not available prior to 1992. For these reasons, we include the comparison to observations of land cover as a sensitivity only.

### 2.2.3.2 Measures of goodness of fit

Different measures of model performance are used to select parameter sets that optimize different aspects of model

performance.[10] We consider normalized and unnormalized metrics, as well as a metric based on comparing summary statistics between simulated and observed time series.

Normalized root mean square error (NRMSE) considers all deviations between simulated and observed values, and places them in the context of the variance seen in the observational data. For crop $i$,

$$NRMSE_i = \frac{\sqrt{mean_i(obs_{i,t} - sim_{i,t})^2}}{\sqrt{mean_i(obs_{i,t} - \overline{obs_i})^2}}. \tag{7}$$

One benefit of this measure is that it includes a natural benchmark of acceptable model performance. While $NRMSE = 0$ corresponds to perfect model performance, any $NRMSE < 1$, is considered acceptable model performance (e.g. (Tebaldi et al., 2020) and the review of metrics in (Legates and McCabe, 1999)). Using the standard deviation of observation as an error baseline puts the deviations between simulation and observation for each crop in the context of that crop's historical variations. If errors in a 1990-2015 gcamland hindcast simulation are greater than the historic standard deviation, then by definition,

simply using the 1990-2015 mean value of land allocation in every simulated year 1990-2015 would have resulted in better errors than the model under consideration. Note, however, that in a hindcast approach, one would not actually have access to the 1990-2015 mean observed land allocation to use as a model to simulation 1990-2015 land allocation; it is simply an easy conceptual counterfactual model. Even when comparing two different model results that each have $NRMSE > 1$, the model with the smaller $NRMSE$ value is considered better.


We also consider the root mean square error (RMSE),

$$RMSE_i = \sqrt{mean_i(obs_{i,t} - sim_{i,t})^2}, \tag{8}$$

and bias,

$$bias_i = (\overline{obs_i} - \overline{sim_i}). \tag{9}$$


---

[10] For this analysis, we use the R `stats` package (R Core Team, 2020).

These un-normalized measures make no distinction between different crops; a bias or RMSE of 200 km$^2$ means exactly the same for Corn as it does for Rice, despite the fact that Corn represents a larger proportion of harvested area in the United States in the historical period. While RMSE is concerned with all deviations between observation and simulation for a crop, bias simply compares the means between observation and simulation. While these means tend to be determined more by the smoothed trend in a time series than variations about the trend, it is important to note that bias specifically does not penalize volatility the way that RMSE and other measures may.

Finally, the Kling-Gupta Efficiency score (Knoben et al., 2019) is also implemented for each crop,

$$KGE_i = 1 - \sqrt{(r-1)^2 + \left(\frac{\sigma_{sim}}{\sigma_{obs}} - 1\right)^2 + \left(\frac{\mu_{sim}}{\mu_{obs}} - 1\right)^2} \ , \tag{10}$$

for correlation coefficient $r$, standard deviation $\sigma$, and mean $\mu$). While a perfect simulation (NRMSE=RMSE=bias=0) would by definition give perfect KGE (KGE=1), KGE is defined by penalties between different time series summary statistics, as opposed to the penalties based on simple deviations between simulation and observation at each time point in the other error metrics considered here.

For a given error measurement, the metric is calculated for each crop in each ensemble member. For NRMSE and RMSE, the average value across crops is then minimized to select the ensemble member with the most optimal parameters for matching observation. For bias, it is the average across crops of the magnitude of bias that is minimized, to avoid cancellation of errors between crops. For KGE, it is the average across crops of the quantity $1 - KGE_i$ that is minimized so that the average across crops of $KGE_i$ is optimized as needed. As an additional sensitivity, the actual land types included in this average metric can be adjusted to include all crops, simply one individual crop, or any combination of land types of interest. By default, we include any land type where we have observations for the full time series of the simulation, which effectively means all crops excluding fodder crops (see Section 2.4.3 and Table S1); however, we include a sensitivity on the set of land types included in Section 5.2.2.

### 2.2.4. Selecting the best parameters by observation type

We calculate goodness of fit for each land type of the gcamland ensemble members and each metric of goodness of fit. We then choose the ensemble member that optimizes the average across land types of interest for each measure of goodness of fit for each expectation type. The parameter set used to generate that ensemble is considered the "best parameter" set for that expectation type. Our default is to use NRMSE as a measure of goodness of fit, but we discuss sensitivity to measure of goodness of fit in Section 4.2.

**2.2.5 Select the overall best model**

The previous step generates four parameter sets, one for each expectation type. In this step, we choose the expectation type and parameter set that optimizes average goodness of fit across all land types, resulting in a single "best model".

**2.2.6. Simulations and sensitivities**

The default ensemble analyzed in this paper uses 1990 as the initial model year, runs annually through 2015, excludes
subsidies, and differentiates the expectation parameters ($\alpha$ and n) by crop groups. To test the sensitivity of the results to each of these assumptions, we re-run the ensemble with alternative specifications for each assumption (Table 3).

**Table 3: Model specifications used in this study.**

| Name | Initial Model Year | Time Step | Subsidies? | Parameters differentiated by crop group? |
|---|---|---|---|---|
| Default | 1990 | Annual | No | Yes |
| Same parameters | 1990 | Annual | No | No |
| With subsidy | 1990 | Annual | Yes | Yes |
| 1975 | 1975 | Annual | No | Yes |
| 2005 | 2005 | Annual | No | Yes |
| 5 year timestep | 1990 | Five year | No | Yes |

Over the last several decades, yields have increased in the United States; prices and profits are more variable (Figure S2). Changes in the area of a particular crop, however, are not always correlated with in year profit (Figure S3, S4). There are several potential reasons for this:

1. Farmers do not know the profit at the time of planting and instead are basing their planting decisions on expectations.
2. The profit calculated here is missing some other factor (e.g., a government subsidy).
3. Profit relative to another commodity may be a better predictor (e.g., if two crops have increases in profit, a farmer might shift to the one with faster increases, resulting in a decline in land area for the other despite its increase in profit).
4. Different crops may have undergone very different improvements in yields over time.
5. Other non-economic factors (e.g., distance to markets) might drive land use decisions.

We explicitly test the first two explanations in this paper. The third and fourth are captured in all of our simulations. The fifth
is implicitly captured in the calibration routine in gcamland but we do not vary this over time.

## 3. Results

This section describes the results from the default gcamland ensemble. This ensemble assumes an initial model year of 1990, an annual time step, subsidies are excluded, and the parameter sets are chosen to minimize the average NRMSE across all crops. Sensitivity to each of these assumptions is presented in the next section. Note that throughout the results and sensitivity sections the default configuration, with the numerically optimal parameter set and expectation type are shown in thick magenta lines for consistency.

### 3.1. Parameter Sets that Minimize NRMSE in gcamland

NRMSE varies across expectation types, ranging from 1.399 with Adaptive expectations to 1.874 with Linear expectations. The parameters that minimize NRMSE vary by expectation type (see Figure 3), including the ordering of the logit exponents. In the Adaptive expectations, the logit exponent dictating substitution among crops is larger than the logit exponents determining substitution between crops and other land types. This rank ordering of logit exponents is consistent with the intuition from historical trends in USA land allocation (Figure 1); specifically, the larger changes in crop mix than total crop area in the observations suggest that the logit for the cropland nest should be larger than the other logits. In all models with imperfect expectations, expected profits are heavily weighted toward previous information, as evidenced by the large values for the share of past information and the number of years in the linear forecast (see also Table S3 and Figure S5). However, these values vary across crop groups. For example, Corn and OilCrop rely less on past information than other crops in the Adaptive expectations and for prices in the Hybrid Linear Adaptive expectations, likely due to changes in the market due to the introduction of biofuels policies circa 2005.

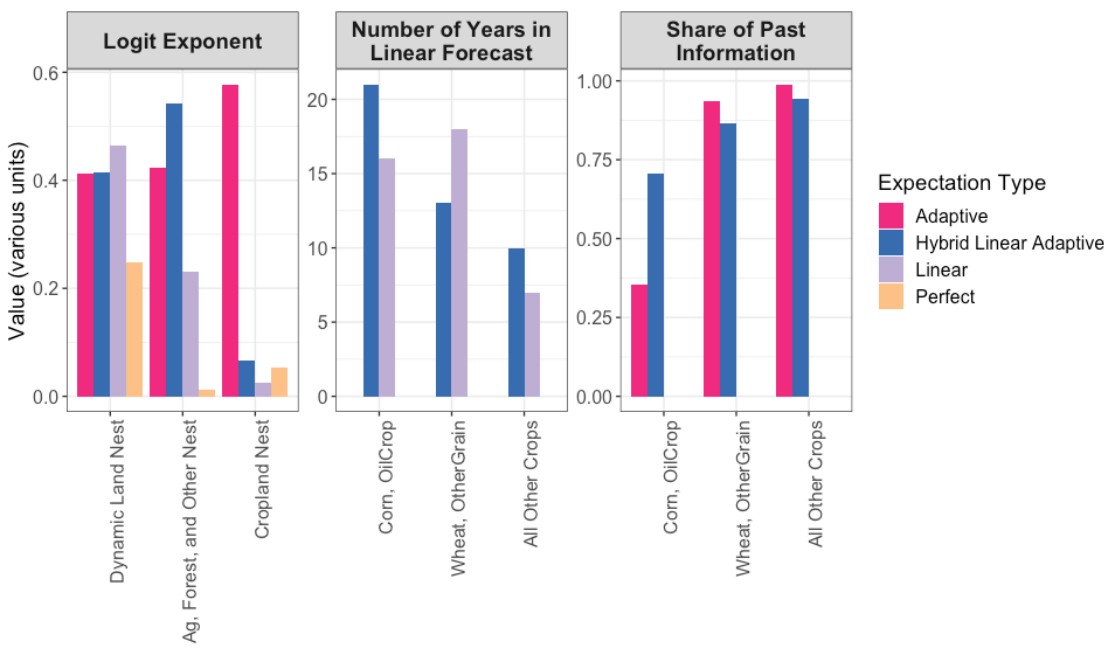


**Figure 3: Parameters that minimize NRMSE by Expectation Type.**

## 3.2. Comparing modeled land area to observations

The full ensemble of gcamland simulations results in a large range of land allocated to crops, covering +/-100% of the observed

area. The parameter sets that minimize NRMSE in gcamland replicate total harvested cropland area over time in the United States fairly well (Figure 4, left panel). However, gcamland misses some of the transitions in crops shown in Figure 1. In particular, for Adaptive expectations (the numerically optimal expectation type and parameter set), gcamland underestimates the growth in OilCrop in the mid-1990s and overestimates the growth in Corn in recent years (Figure 4). The insights from Figure 4 are confirmed when examining the crop-specific NRMSE in this simulation. The NRMSE for Corn and OilCrop are

larger (1.88 and 1.67, respectively) than the NRMSE for other Wheat and OtherGrain (1.16 and 0.7, respectively) (see also Figure S12). Similar comparisons are shown for all 12 GCAM crop types in the supplementary material (Figure S6 and S7), including the four types plotted in Figure 4, as well as for land cover types (Figure S9, S10, S11). Time series of the cropland share over time for these four crops are also included in the supplementary material (Figure S8).

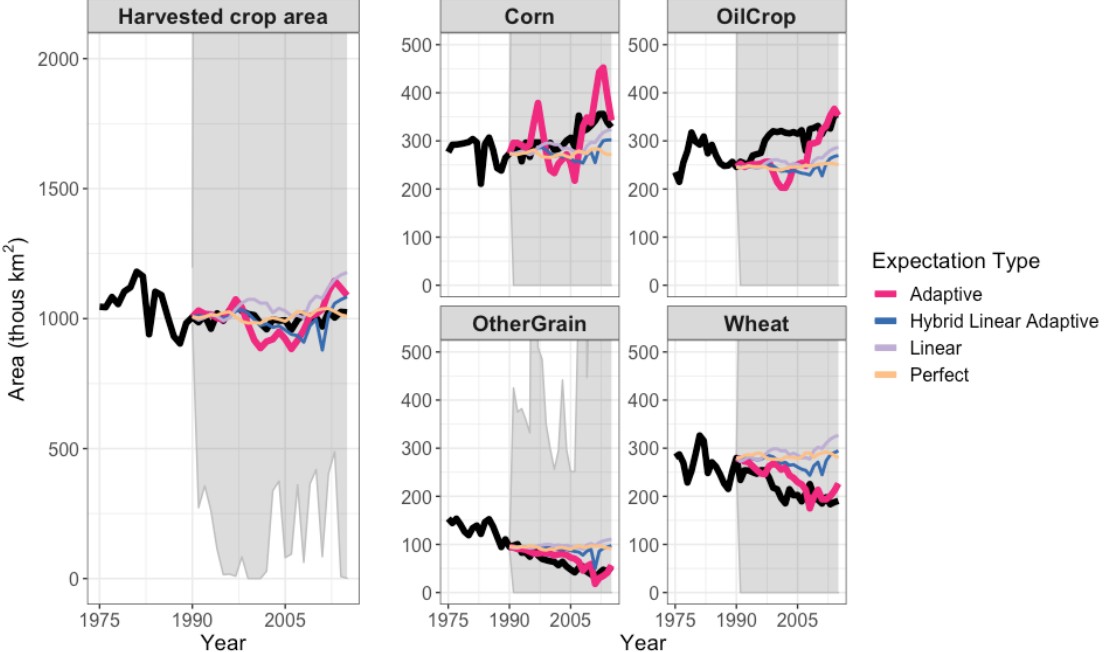

**Figure 4: Harvested crop area (total and by crop) over time by expectation type. Black line is observations (FAO). Colored lines are gcamland results for the models that minimize NRMSE. The expectation type with the minimum NRMSE (Adaptive) is shown with a thicker line. Gray area is the range of all gcamland simulations. Note that fodder crops are included in gcamland but are excluded from total cropland area in this figure due to data limitations. Figure S6 shows this same information for all 12 GCAM crop types and Figure S9 shows this for land cover types.**

## 4. Sensitivity Analysis

In this section, we describe the sensitivity of the results above to several different assumptions, including those related to the configuration of the model, the initial model year, the model timestep, and the objective function used. For the model configuration, initial model year, and timestep sensitivities, we generate new ensembles of gcamland results with the appropriate assumption altered. For the sensitivity to objective function, we filter the original ensemble using different criteria to determine the numerically optimal parameter sets.

### 4.1. Sensitivity to Model Assumptions

First, we test the sensitivity of the analysis to two different assumptions: (1) whether subsidies are included in the expected profit for crops, and (2) whether the expectation-related parameters differ across crops. For all three sets of assumptions, Adaptive expectations minimizes NRMSE. Varying these assumptions results in differences in cropland area (Figure 5) and in parameters for the "Same Parameters" sensitivity; however, the parameters for the "With Subsidies" sensitivity are identical to the Default model (Table S5). Including subsidies increases the NRMSE (from 1.399 in the Default case to 1.46 with subsidies). This is likely due to the quality of the subsidy data. Including all factors that affect profit should improve the model;

however, the subsidy data is incomplete (only direct payments were included for crops where these were reported) and inconsistent (reporting changed over time). In addition, previous studies have shown that direct payments have little effect on crop production or land area in the United States (Weber and Key, 2012), suggesting that better subsidy data may not change land allocation decisions substantially.

Using the same expectation parameters across commodity groups increases NRMSE (from 1.399 in the Default case to 1.531 with uniform parameters). There are several reasons why different crops could require different parameters. First, one would expect differences between annual and perennial crops due to the lag between planting and harvesting and the multi-year investment required by perennial crops. Second, some crops (e.g., Corn and OilCrop) have had shifts in policy or demand over time (e.g., for biofuels). Such shifts may lead landowners to prioritize newer information. Finally, there could be differences in how markets are structured (e.g., futures contracts) or region-specific differences. These effects are difficult to disentangle in gcamland. Perennial crops are all included in the "All other crops" group. This group is a mix of both perennial and annual, but we do see higher shares of past information in this group than in the other commodity groups in the Default model. Corn and OilCrop rely more heavily on new information when parameters vary, which is consistent with the market shifting hypothesis.

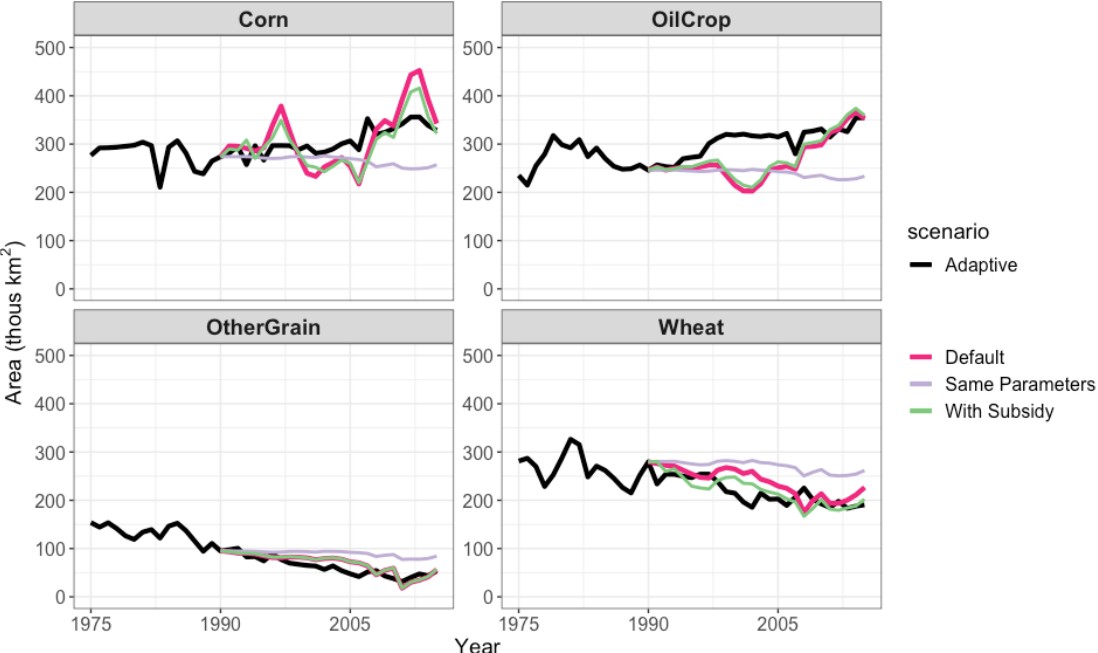

**Figure 5: Harvested area by crop under different model assumptions. Black line is observations (FAO). Colored lines are gcamland results for the models that minimize NRMSE.**

## 4.2. Sensitivity to the Objective Function

The analysis above uses the average NRMSE across all crops as an indicator of "goodness of fit", but other objective functions are possible. In this section, we discuss alternative measures of "goodness of fit", including bias, RMS, and KGE. Additionally, we examine the implications of minimizing NRMSE for an individual crop as opposed to the full set of crops.

### 4.2.1. Optimizing for different objective functions

The parameter sets (Table S6) and cropland time series (Figure 6) that are numerically optimal for KGE are somewhat similar to those of NRMSE and the parameter set that minimizes RMSE is identical to that of NRMSE.[11] The NRMSE and RMSE minimize objective function values with the Adaptive expectation, while the KGE minimizes values with the Hybrid Linear Adaptive expectation. All three rely less on past price information for Corn and OilCrop (share ranges from 0.36 with NRMSE and RMSE to 0.61 with KGE) than for all other crops (share of past information > 0.93). The logit exponents are relatively small (0.05 to 0.58 across all three objective functions and all three nests), with modest substitution allowed in the cropland nest (logit exponent of 0.37 in KGE and 0.58 in NRMSE and RMSE).

The parameter set that minimizes bias, however, is fundamentally different. The logit exponents dictating the substitution between crops and other land types are large (2.18 for the Dynamic Land nest; 1.38 for the Ag, Forest, and Other nest). The parameter set that minimizes bias also includes the lowest Cropland nest logit value of any objective function (0.28). The resulting simulations for bias exhibit large volatility in land area. Given that bias simply compares the model mean across time to the observation mean across time, this volatility is not penalized in the bias metric, whereas it is penalized for KGE, RMSE and NRMSE. For example, the parameter sets that minimize bias result in an average simulated Corn area of 307 thous km$^2$ compared to an average observed Corn area of 306 thous km$^2$, resulting in a bias of less than 1 thous km$^2$. This bias is much lower than the bias for Corn in the other objective functions (NRMSE and RMSE have a bias of 6 thous km$^2$; KGE has a bias of 16 thous km$^2$). Bias is effectively assessing whether the model is correct on average and not whether it captures the trends or volatility; such an objective function is less useful in systems where trends are significant or where the goal is to capture the volatility. From a mechanistic perspective, we hypothesize that the difference in the cropland area volatility when bias is minimized is due to the differences in the Ag, Forest, and Other logit.

---

[11] Note that this is not true in general but is true for the Default model. Other configurations of the model have different parameter sets that minimize NRMSE than those that minimize RMSE.

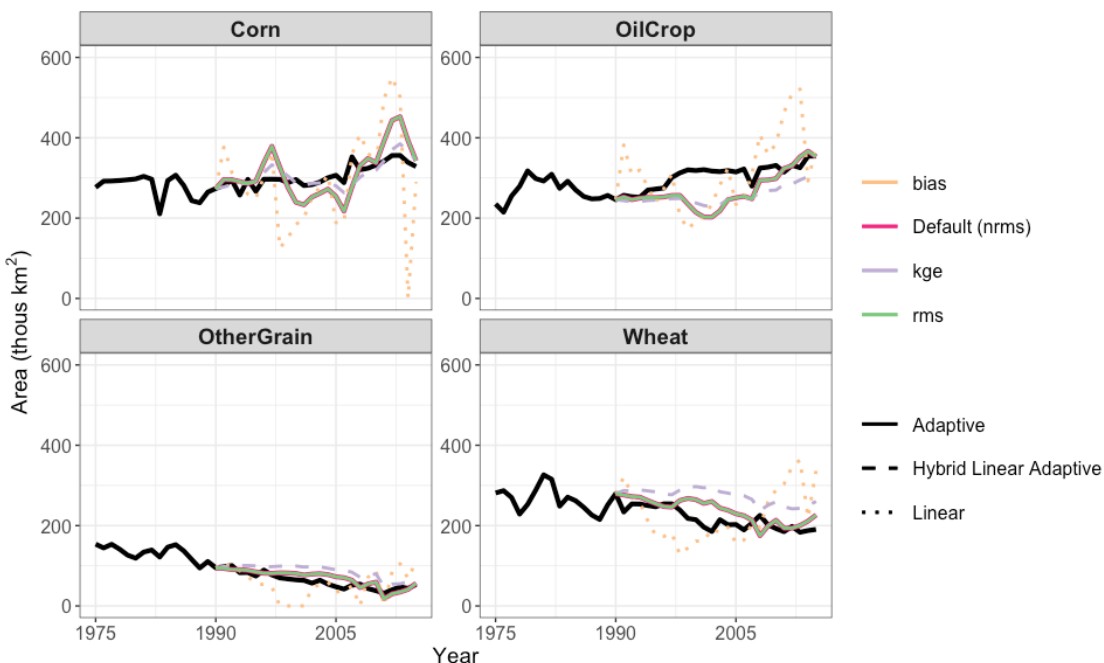

**Figure 6: Harvested area by crop when optimizing for different objective functions. Colors indicate objective function. Line type indicates the expectation type that minimizes that objective function. Only the objective function minimizing expectation type is shown. Note that NRMSE and RMSE result in identical parameter sets in the default model and thus have identical land allocation in this figure.**

### 4.2.2. Optimizing for different land types

Figure 7 shows the difference in the best models when we optimize for a particular set of land types or crops. As seen in this figure, gcamland can track land area for any given crop very well when the ensemble with optimal parameters is chosen specifically for that crop. However, matching all crops at once is more challenging. For example, the parameter sets that minimize NRMSE for Corn result in an excellent match between observations and model output for Corn; however, those parameters result in an overestimation of Wheat land by 250 thous km$^2$ in 2015 (or ~1/2 of the actual area). The insights from

this figure are also confirmed numerically. The NRMSE for Corn is reduced from 1.88 to 0.72 when we go from minimizing NRMSE across all crops to minimizing NRMSE for Corn only. Similarly, the NRMSE for OilCrop is reduced from 1.67 to 0.54 when we go from minimizing NRMSE across all crops to minimizing NRMSE for OilCrop only. Optimizing for a single crop has less effect on the NRMSE for Wheat and OtherGrain (from 1.16 to 0.79 for Wheat, and from 0.7 to 0.43 for OtherGrain). Finally, including all dynamic land cover types where observations are available for any period of the simulation

years (e.g., non-fodder crops, grassland, shrubland, and forest) in the calculation of NRMSE increases the NRMSE substantially (from 1.4 to 75) due to definitional differences in land cover types. The change in land area for land cover types is reasonably consistent with observations (Figure S10); however, the absolute area for grassland and shrubland differs

substantially (Figure S9). Despite the increase in NRMSE, the inclusion of land cover types does not alter the parameter sets that minimize NRMSE.

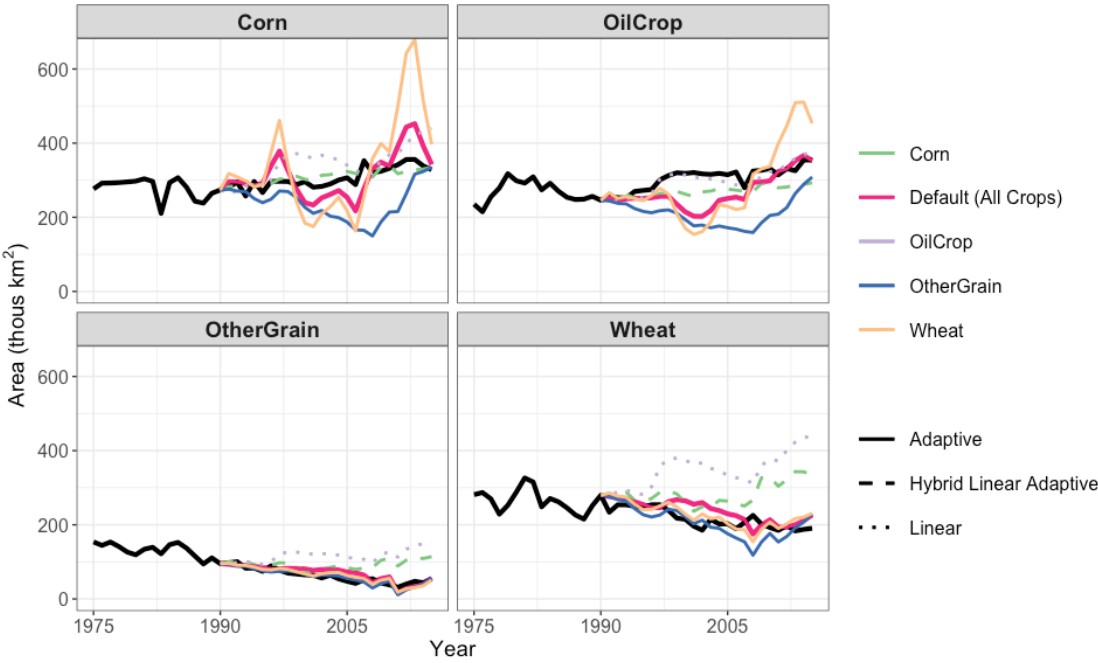


**Figure 7: Harvested area by crop when optimizing for different land types. Colors indicate crops included in the objective function. Line type indicates the expectation type that minimizes NRMSE for that set of crops. Only the NRMSE minimizing expectation type for each set of crops is shown.**

### 4.3. Different initial model years

The calibration routine in gcamland calculates share weight parameters ($\lambda_i$ in equation 1) to ensure that the land area is exactly replicated in the specified base year. Those parameters are held constant in all subsequent periods. Changing the base year could result in different calculated share weight parameters and thus different land allocation, even if all other parameters are the same. In this section, we test this sensitivity, using 1975 and 2005 as alternative initial model years. Figure 8 shows the difference in cropland area for the parameter sets that minimize average crop NRMSE for each initial model year. Those

parameter sets are shown in Figure S13. The resulting parameters and land use are relatively similar between variants with initial model years of 1990 (the default described above) and 2005. The logit exponents are small for all three nests, with the largest value over the cropland nest. Both models use more past information for All Other Crops than for Corn and OilCrop, but they differ in the degree of past information used for Wheat and OtherGrain. The variant with a 1975 initial model year, however, has large differences in parameters and behavior from those with 1990 and 2005 initial model years. We hypothesize

two reasons for these differences. First, we have a limited time series prior to 1975, which results in erroneous estimates of expected price and expected yields for parameter sets with large reliance on past information. Second, there is a discrepancy between FAO harvested area and the land cover data sets used in GCAM in 1975 (this discrepancy exists but is much smaller

from 1990 onwards). In particular, FAO harvested area is larger than the physical crop area. We correct this in gcamland by assuming that some areas are planted more than once in a year. However, this results in larger annual yields in gcamland than

the harvest yield provided by FAO. This results in higher profit rates that could affect the land allocation. Note that this issue is not a problem in future simulations, like those typically run in GCAM, since the calibration information used in future periods is the information calculated from a more recent year without these data challenges (2010 or 2015 depending on the version of GCAM).

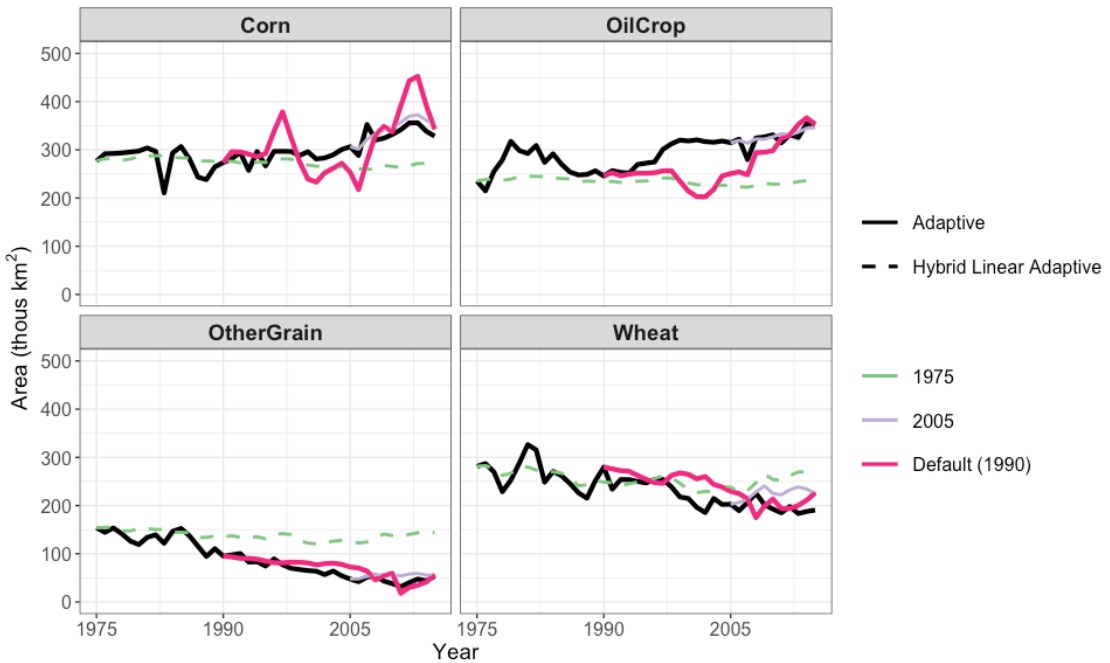


**Figure 8: Harvested area by crop when using different initial model years. Colors indicate initial model year. Line type indicates the expectation type that minimizes that NRMSE for that initial model year. Only the NRMSE minimizing expectation type for each initial model year is shown.**

### 4.4. Short-run versus long-run parameters

Finally, we examine the sensitivity of the results to the time step. Most studies using GCAM use a five-year time step with perfect expectations. However, we are increasingly interested in quantifying the implications of climate variability and change on agriculture, land use, and the coupled human-Earth system, which requires higher temporal resolution. For purposes of this comparison, we focus on RMSE instead of NRMSE. NRMSE and RMSE differ in that NRMSE is normalized by standard deviation; however, the inclusion of standard deviation introduces inconsistencies when comparing across time steps. For the

Default model, the choice of RMSE or NRMSE has no effect on results, but for the five year time step it does. We note any differences that would emerge from using NRMSE in this discussion.

Our hypothesis was that longer time steps would result in larger logit exponents since farmers would have more time to make adjustments and that expectations would matter less with longer time steps. Using RMSE, the former is true, but the latter is not.[12] The five-year timestep results in higher logit exponents, particularly in the Dynamic Land nest and the Cropland nest; the expectation parameters are similar though (Figure S14). However, the Hybrid Linear Adaptive expectations minimizes RMSE in the five-year time step model (Table 4), suggesting that expectations are still important for longer time steps (see also Figure S16).[13] We find that the one-year time step results in a lower RMSE than the five-year time step model, even when the differences in comparison years are taken into account (Table 4): the RMSE computed over five-year increments in the one-year model is still lower than the RMSE in the five-year model. In the five-year time step model, farmers use five-year averages of price and yield when forming expectations. As a result, the five-year time step model will produce different expectations (Figure S17) and different land allocation results (Figure S18) than the one-year time step model even when the same parameters are used. The fact that annual time steps reduce RMSE suggests that interannual variability may have a noticeable influence on expectations and the resulting land allocation; that is, farmers consider not just the trend in yield and price but also the variability around that trend. This is particularly true for Corn and OilCrop where more recent information has a larger effect on expectations.

Table 4: The effect of time step, expectations, and comparison years on RMSE

| Time Step | Expectations | Comparison Years | RMSE |
|---|---|---|---|
| 1 year[*] | Adaptive | Annual, 1990-2015 | 16.1 |
| 1 year | Adaptive | 5-year increments, 1990-2015 | 14.5 |
| 5 year | Hybrid Linear Adaptive | 5-year increments, 1990-2015 | 18.7 |
| 5 year | Perfect | 5-year increments, 1990-2015 | 25 |

[*] This variant is equivalent to the Default shown earlier in the paper.

## 5 Discussion and Future Work

In this paper, we have explored structural sensitivities and used a perturbed parameter ensemble of simulations of land use and land cover over the historical period to guide the selection of structural economic assumptions and associated parameters for an economic model, gcamland, in the United States. The exploration of different expectation types using a perturbed parameter ensemble and then selecting the optimal combination by comparing hindcast simulations to historical observed data not used in those simulations is a key part of this study, and an addition to the economic land use modelling literature. In addition to

---

[12] Using NRMSE, the logit exponents are slightly smaller in the five-year timestep model than in the one-year timestep model (Figure S10), but expectations reduces error in the five-year timestep model under both RMSE and NRMSE. The resulting land allocation in the five-year time step model for both RMSE and NRMSE is shown in Figure S12.
[13] With NRMSE, Adaptive expectations minimizes error. Like RMSE, we still find that expectations are important for longer timesteps.

exploring expectation types, we also explored structural sensitivities to the objective function used for comparison to historical observed data, the historical period over which the hindcast simulation is run, and the inclusion of subsidy data. We find that adaptive expectations minimize the error between simulated outputs and observations, consistent with empirical evidence (Mitra and Boussard, 2012). The resulting parameters suggest that for most crops, landowners put a significant weight on previous information. For Corn and OilCrop, however, a large weight is placed on more recent information. This is consistent with an observation by Kelley et al. (2005): "In the case of agriculture, anecdotal evidence suggests that some farmers are more myopic, weighing recent information more than is efficient."

The optimal expectation type and set of parameters is sensitive to the choice of objective function, with differences emerging either when the mathematical formulation of the error is altered or when the set of land types included in the calculation of error is changed. For the former, we find that using bias as an objective function leads to the largest volatility in annual land allocation. While GCAM has historically performed better at capturing overall trend behavior than annual variations and this has been considered acceptable model behavior (Calvin et al., 2017; Snyder et al., 2017), the results of this study highlight the importance of penalizing variations about the trend as well. For the latter, it is possible to significantly improve the performance for the model for any single crop by optimizing for that crop; however, the resulting parameters may lead to a larger error for a different crop. For example, the parameter sets that minimize NRMSE for Corn result in an excellent match between observations and model output for Corn; however, those parameters result in an overestimation of Wheat land by approximately 250 thous km$^2$ in 2015.

We hypothesize that limitations of data affect the performance of some variants of the model. For example, the variant that explicitly excludes subsidies outperforms the one with subsidies, likely due to the poor quality of the subsidy data. Similarly, the variant with 1975 as an initial model year is fundamentally different from the variant with 1990 or 2005 as an initial model year, likely due to discrepancies between harvested and physical area in 1975 and limited availability of the data prior to 1975 that is needed to form expectations. Similarly, the land cover data provides little constraint on the model due to the short time series and difference in definitions of land categories. Future work could include improvements in the data and the addition of new data sets to constrain the model. In theory, any change in the data or in the profit calculation, like the inclusion of subsidies, could alter the error and the set of parameters that minimize error (i.e., conversely, the exclusion of those factors could introduce biases in the estimated parameters). However, in our study, we found that the inclusion of subsidies increased NRMSE, but did not alter the parameters that minimized NRMSE. Additionally, we have focused on the United States, using national level data. Because the United States is a data-rich region, it was chosen as an initial focus for developing this methodology and identifying important structural sensitivities. Future work could replicate this hindcast-based analysis for subnational regions or for other countries around the world with a more streamlined approach to some of the sensitivities explored (e.g., only running the 1990-2015 annual variant and focusing on NRMSE). This is particularly practical in gcamland, in which exploring structural sensitivities and estimating parameter values for country-level or larger regions is an independent

exercise: given historical price and yield data for that region, the land allocation model can have decision parameters estimated independently in each region. Our expectation is that we would find qualitatively different combinations of parameters best replicate observations in other countries, similar to what is asserted in Taheripour and Tyner (2013).

Other potential research directions include testing other assumptions in gcamland (e.g., the nesting structure, multi-cropping), new explanatory variables (e.g., crop insurance, speculative storage), alternative decision-making frameworks (e.g., non-logit approaches), or additional behavioral processes (e.g., learning, diffusion). For the nesting structure, we have only tested the default GCAM nesting structure here. Taheripour and Tyner (2013) test an alternative nest and find that it has implications for the share of forest cover (14% vs. 3% depending on the nest). For multi-cropping, gcamland includes both harvested and physical area; however, the ratio between the two is held constant. Intensification through multi-cropping could be more important when extending the study outside of the United States; however, additional data and investigation is needed. For explanatory variables, studies have indicated that some programs, like crop insurance, are likely to have a direct impact on area planted and production (Young and Westcott, 2000). For alternative decision-making frameworks, Zhao et al. (2020b) demonstrate that the resulting change in land use due to a shock differs depending on the combination of functional form (logit, constant elasticity of transformation, constrained optimization) and parameter value. Finally, our study focused on land supply responses and did not identify the sources of price changes. Future studies could extend our model structurally to explicitly identify demand shocks, responses, and their effects on prices.

In this paper, we have focused on the historical period, simulating land allocation in gcamland over this period and comparing it to observations that were not used for the simulation. However, these structural assumptions and parameter estimates could be used in a simulation of future land use and land cover change to better understand their implications. Because gcamland implements the same land allocation equations and structure as GCAM, expectation structures and parameter values estimated for gcamland can have utility in future GCAM experiments, when they have been estimated for all 32 geopolitical regions shared by gcamland and GCAM. It would not be computationally feasible to perform this extensive and systematic exploration of economic expectations (and other structural sensitivities) and parameters directly in GCAM. In the future, the methodology established in this paper will be repeated with gcamland for all 32 regions, and the resulting optimal parameters will be run through GCAM in a hindcast to see if the data-informed economic expectations and parameters result in an overall better evaluation of model performance than the default decision parameters and expectation structure (as in (Calvin et al., 2017; Snyder et al., 2017)). Finally, while other modeling teams are unlikely to be able to use the exact parameters due to differences in model structure and inputs, the methodology described here and the lessons learned could be used by other economic models.

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

## Author Contributions

KC and MW designed the experiment. KC and AS developed the model code. KC performed the simulations. KC, AS, and XZ analyzed results. KC prepared the manuscript with contributions from all authors.

## Acknowledgments

This research was supported by the US Department of Energy, Office of Science, as part of research in MultiSector Dynamics, Earth and Environmental System Modeling Program. The Pacific Northwest National Laboratory is operated for DOE by Battelle Memorial Institute under contract DE-AC05-76RL01830.

## Data Availability

gcamland code and inputs are available at https://github.com/JGCRI/gcamland and https://zenodo.org/record/4071797. All outputs and the code used to generate the figures in this paper are available at https://github.com/JGCRI/calvin-etal_2021_gmd.