# Peer review of "Modeling Land Use and Land Cover Change: Using a Hindcast to Estimate Economic Parameters in gcamland v2.0"

_Geoscientific Model Development, 2020_

## Referee Comment (RC1) · Anonymous Referee #1 · 28 Dec 2020

GENERAL COMMENTS

In this manuscript, Calvin et al. conduct a large ensemble of simulations for the United States with the gcamland land use model to optimize model structure and parameters related to landowner decisionmaking. They find that an adaptive expectation structure performs best, in contrast to the linear structure previously used in GCAM. The authors also test the sensitivity of the optimization to various setup choices—including objective function, timestep, and calibration year—finding that, where the results are not robust across setups, they differ in mostly predictable and understandable ways.

This manuscript represents two important contributions to the development of gcamland, which seem to not often be explored by other land models: The challenging of the model with historical data, and the testing of alternative structural formulations. (I am admittedly not familiar enough with the land use modeling literature to say for certain that the authors have included all relevant previous work in their review of previous literature, however.) It also provides a useful blueprint through which similar work could be performed for other land models. The manuscript is well-written and clear from the introduction through the conclusions, although I have some minor comments and suggestions (listed below).

I thus recommend that this manuscript be accepted subject to minor revisions.

SPECIFIC COMMENTS

- The authors should better tie the paragraph at lines 69–79 (re: previous attempts to parameterize land use models based on real-world information) into the rest of the paper. In the following paragraph, they could more explicitly lay out the novelty of their work relative to what's been done before in this area. Then, in section 2.1.2, they should tie back more explicitly into the idea of elasticity, which is how they characterize much of the work pointed to at 69–79.

- Did the authors use any R (or other) packages to perform the Latin Hypercube sampling and/or to evaluate the results? If so, it would be helpful for reproducibility to specify those.

- Fig. 4 needs at least one other marking on the X-axis. Ideally, match Figs. 5 and 6.

- L334: "the difference in the bias area volatility" is confusing. Maybe a word is missing?

- Fig. S2: A 1:1 line would be helpful.

- Fig. 6: It would be helpful to have the thick lines below the rest, since that would allow the thinner lines to be seen even when they overlap it (which they frequently do).

- Fig. S11: Middle panel only includes bars for "5 year timestep (RMSE)" because that

[Figure]

was the only setup where the error-minimizing setup used that parameter, right? That should be mentioned in the caption.

TECHNICAL CORRECTIONS

- Lines 152–153: NRMS should be changed to NRMSE for consistency with the rest of the manuscript.

- Fig. 3: "... fodder crops *are* excluded..."

- L329–330: Units should be added to the first part of the sentence. Additionally, it feels weird to say "the parameter sets... result in an average observed Corn area", as the parameter sets don't have anything to do with the observations.

- Figs. S4, S5, S7, S9, S12, and S13 should be mentioned in the main text.

- Rounding inconsistencies between Sect. 5.2.1 and Table S3 should be corrected.

- "Ag, Forest, and Other" is referred to in the Supplement as "AgForest_NonPasture". Ideally, one should be chosen for consistency.

---

## Referee Comment (RC2) · Anonymous Referee #2 · 3 Jan 2021

Summary of the paper This paper uses gcamland/GCAM to calibrate/estimate/tune the land distribution parameters of a nested logit land allocation function used in this model. In the lack of econometrically estimated values for these parameters, it is an important effort to accomplish this task. However, the paper suffers from some important deficiencies and lack of clarity, in particular for those who do not know this mode.

Some important comments:

1. I am not a GCAM modeler but is seems gcamland operates under GCAM. For non GCAM community the links and interactions between these two models are not clear. How they linked and interact. A simple chart can help.

[Figure]

2. The model clearly uses a nesting logit format, perhaps three nests. Equation 1 of the paper shows only on nest. The formula should be replaced with a formula for the full nest.

3. How the land constraint/constraints is/are defined? Does a simple land constraint directly add all types of land: Total land =forest + pasture + corm + soy+ etc.? or each nest has its own land constraint?

4. It is not clear how the estimation process is defined to estimate these parameters. Does the process estimate all the distribution parameters (s) simultaneously or individually?

5. How distribution parameters () were perturbed? Are they coming from given distribution? If yes, what type of distribution? Is this a random selection of three values limited between 0.01 and 3?

6. Over time total area of agricultural land in the US has declined sharply, due to conversion to non-agricultural uses of land (urbanization, infrastructure, ….). How gacamland handles conversion of land to non-forestry-ag land. How land availability land has been taken care off over time? Is it an exogenous variable in each year?

7. A big issue in land use modeling is marginal cropland (idled land under CRP, cropland pasture, other types of idled). Area of idled cropland in the US have changed a lot. The definition of "cropland pasture" has also changed over time. How idled land is treated?

8. It is noted that harvested area from FAO is used. FAO is missing many feed crops since 2011, including million hectares of those crops. without proper steps to cover missing crops in FAO, the estimated parameters will be subject to major issues and biases. Figure three suggest that those feed crops is missed. That is a major issue.

9. GCAM is using commodity price to model land allocation. It seems wholesale farm prices is used. That is a bad proxy for exporting crops such as cone and soybeans.

For example, half of soybean is exported at much higher price farm price.

10. In this paper, in one case, subsidy has been examined in a sensitivity test. Subsidy is the key item in deriving land use, land rent, and the price received by farmers. The distribution parameters of the logit should be evaluated with subsidies. Sensitivity test is meaningless. The key here is to capture all types of subsidies paid to famers in the estimation processes.

11. How biofuels were included in the simulations? Biofuels and biofuel policies were major drives of land use. How that included in your simulations

12. Th dapper highlights that gcamland uses commodity prices in land allocation. But the model allocates land across land cover items. What prices are used for forest products, livestock products, etc.? The paper is silent on these prices. What prices were used for land cover items

13. Regarding forestry, how gcamland treats forest land. Is it operates based on managed forest? Managed + unmanaged? How it treats unmanaged forest with no economic output.

14. GCAM and gcamland are not forestry models. Forestry is not an annual crop. How these models take care of forestry in a dynamic setting. Do these models treat forestry as an annual crop?

15. In each case, the model is solved for a range of parameters. Then a set of parameters that minimizes NRMSE is selected. But NRMSE is defined for a single crop. How this variable is aggregated over crops? How NRMSE is calculated for non-cropland (e.g. forest, pasture, grass land, and etc.)? How cropland and non-cropland aggregated?

16. How productivity of non-cropland is measured?

17. GCAM aggregates crops into some specific categories? How prices were generated for those categories. In many cases there is no data on crop prices?

18. The paper provides mixed messages on endogenous and exogenous variable. In determining targeted distribution parameters, what variables were targeted and what variables were determined in the model. It seems prices, areas, and yields were exogenous. Be more specific.

19. The whole practice implicitly assumes that other model parameters are accurate and valid. This is a strong assumption. The land supply parameters were determined while demand parameters held constant. The estimated supply parameters will be entirely wrong if the demand parameters (e.g. income and price elasticities for crops, livestock products, and forestry) are not valid. Any change in the demand parameters could alter your estimated parameters for the land supply. Can you test sensitivity of your results with respect to changes in other elasticities of the model?

20. The results are counterintuitive. Let me explain using figure 2. In the adaptive case, for the first two nests the  values are about 0.4 and for the last nest (cropland) the value of  is about 0.6. Given that limited land movements among land cover items occurred at national level in the US and lots of change occurred in the crop nest, one could justify this outcome. However, for the other three cases (hybrid, linear, and perfect) the ranking is of  values shows revers. Meaning that land conversion is easier at the land cover nests than the cropland land nest. These outcomes do not make sense. Am I missing something?

21. In showing the results, level variables were used to show errors. For example, figure 2 compares estimated harvested areas with their observations for four types of expectations. This hides the errors involved. It is better to calculate errors as percent difference between the estimated and observed areas.

22. The main manuscript only presents comparison of the projected and observed harvested areas and provided no comparison for other land types.

23. Results are highly aggregated into four groups of crops. how about the 12 categories of crops in GCAM?

24. The figure S5 of SI shows major errors for the change in forest area. This show that the model fails to represent changes in forest area correctly.

25. The figure S5 show increases in all land cover types and harvested areas. How that could be possible?

26. Figure S7 shows no results for land cover items including grassland and shrubland for three types of expectations. Why?

27. Figure S7 shows major errors for grassland and shrubland in the adaptive approach project huge errors. Why?

28. It seems the whole practice has failed to take care of land cover changes.

29. The examined practice estimated a few parameters of the model for land use. A good way to test the outcomes of this practice is to run the GCAM model with the estimated parameters and compare the model results for land use changes, land cover changes, changes in crop prices, and changes in yield with actual observations over the examined period.

30. Finally, the whole work could be a valuable practice for the CGAM community. It uses "hindcast" to estimate the logit distribution parameters for this model. Hindcast Is not a new approach. The outcome of this practice may help the GCAM community to improve their work on land use modeling. However, the results of this practice may be not useable for other models. As they may follow very different modeling structure and assumptions. The author of this paper should make this point very carefully.

31. The abstract provides trivial information. It is not an abstract of this paper.

32. Following a summary of land use change at the global scale, the second paragraph of the introduction begins with: "Similar trends occurred in the United States". This is not an accurate statement. The US land use change did not follow the global land use changes in terms of land conversion to crop production. No expansion in cropland has been observed for the cases of US.

---

## Author Comment (AC1) · 23 Feb 2021

GENERAL COMMENTS In this manuscript, Calvin et al. conduct a large ensemble of simulations for the United States with the gcamland land use model to optimize model structure and parameters related to landowner decisionmaking. They find that an adaptive expectation structure performs best, in contrast to the linear structure previously used in GCAM. The authors also test the sensitivity of the optimization to various setup choices—including objective function, timestep, and calibration year—finding that, where the results are not robust across setups, they differ in mostly predictable and

understandable ways. This manuscript represents two important contributions to the development of gcamland, which seem to not often be explored by other land models: The challenging of the model with historical data, and the testing of alternative structural formulations. (I am admittedly not familiar enough with the land use modeling literature to say for certain that the authors have included all relevant previous work in their review of previous literature, however.) It also provides a useful blueprint through which similar work could be performed for other land models. The manuscript is well-written and clear from the introduction through the conclusions, although I have some minor comments and suggestions (listed below). I thus recommend that this manuscript be accepted subject to minor revisions.

Author Response: Thank you for your feedback and helpful suggestions.

Author Changes: We have made a number of changes to the paper in response to your comments and those of another reviewer. Those changes are detailed below.

SPECIFIC COMMENTS - The authors should better tie the paragraph at lines 69–79 (re: previous attempts to parameterize land use models based on real-world information) into the rest of the paper. In the following paragraph, they could more explicitly lay out the novelty of their work relative to what's been done before in this area. Then, in section 2.1.2, they should tie back more explicitly into the idea of elasticity, which is how they characterize much of the work pointed to at 69–79.

Author Response: Thank you for the suggestion. We have made both suggested changes.

Author Changes: We have added "we advance the science on parameterizing land use models by using statistical approaches rather than the heuristic approaches described in the previous paragraph" to the last paragraph of the introduction (starting around line 83). We have also added text in 2.1.2 to tie the approach in gcamland to the land supply elasticity: "These parameters influence the land supply elasticity, which is non-constant (i.e., it varies depending on the relative profitability as described in Wise et

al., 2014)."

- Did the authors use any R (or other) packages to perform the Latin Hypercube sampling and/or to evaluate the results? If so, it would be helpful for reproducibility to specify those.

Author Response: We used the R lhs package for the sampling. We also use the R stats package for analysis.

Author Changes: We have added this information to the methodology section of the paper.

- Fig. 4 needs at least one other marking on the X-axis. Ideally, match Figs. 5 and 6.

Author Response: Thank you for the suggestion.

Author Changes: We have added additional x-axis labels.

- L334: "the difference in the bias area volatility" is confusing. Maybe a word is missing?

Author Response: We agree this was confusing.

Author Changes: We have revised this phrase to say "the difference in the cropland area volatility when bias is minimized"

- Fig. S2: A 1:1 line would be helpful.

Author Response: We have added a line.

Author Changes: We have added a line.

- Fig. 6: It would be helpful to have the thick lines below the rest, since that would allow the thinner lines to be seen even when they overlap it (which they frequently do).

Author Response: Thank you for the suggestion.

Author Changes: We have made this change.

- Fig. S11: Middle panel only includes bars for "5 year timestep (RMSE)" because that was the only setup where the error-minimizing setup used that parameter, right? That should be mentioned in the caption.

Author Response: We used RMSE to compare results from the 1-year timestep model configuration to the 5-year timestep model configuration. We chose this instead of NRMSE because NRMSE normalizes by standard deviation which will change depending on timestep.

Author Changes: We have added an explanation to the caption.

TECHNICAL CORRECTIONS - Lines 152–153: NRMS should be changed to NRMSE for consistency with the rest of the manuscript.

Author Response: We have made this correction.

Author Changes: We have made this correction.

- Fig. 3: "... fodder crops *are* excluded..."

Author Response: We have corrected the grammatical error in this sentence.

Author Changes: This phrase now says: "fodder crops are excluded from..."

- L329–330: Units should be added to the first part of the sentence. Additionally, it feels weird to say "the parameter sets... result in an average observed Corn area", as the parameter sets don't have anything to do with the observations.

Author Response: Thank you for catching this.

Author Changes: We have added units to the first part of the sentence. We have also revised the sentence to say link the parameter sets to simulated area only: "the parameter sets that minimize bias result in an average simulated Corn area of 307 thous km2 compared to an average observed Corn area of 306 thous km2..."

- Figs. S4, S5, S7, S9, S12, and S13 should be mentioned in the main text.

[Figure]

Author Response: We have added references to all supplemental figures in the main text.

Author Changes: We have added references to all supplemental figures in the main text.

- Rounding inconsistencies between Sect. 5.2.1 and Table S3 should be corrected.

Author Response: We have corrected these inconsistencies.

Author Changes: We have corrected these inconsistencies.

- "Ag, Forest, and Other" is referred to in the Supplement as "AgForest_NonPasture". Ideally, one should be chosen for consistency.

Author Response: We have corrected this.

Author Changes: We have updated the Supplement to say "Ag, Forest, and Other"

Anonymous Referee #2

Summary of the paper This paper uses gcamland/GCAM to calibrate/estimate/tune the land distribution parameters of a nested logit land allocation function used in this model. In the lack of econometrically estimated values for these parameters, it is an important effort to accomplish this task. However, the paper suffers from some important deficiencies and lack of clarity, in particular for those who do not know this mode.

Author Response: Thank you for your detailed review of our paper. Your comments have pointed out a number of places where we did not clearly describe our methodology.

Author Changes: We have made a number of changes to the paper in response to your comments and those of another reviewer. Those changes are detailed below.

Some important comments: 1. I am not a GCAM modeler but is seems gcamland operates under GCAM. For nonGCAM community the links and interactions between

[Figure]

these two models are not clear. How they linked and interact. A simple chart can help.

Author Response: GCAM and gcamland are completely separate models. GCAM includes representations of energy, water, land, and climate. It includes a land allocation mechanism where land use and land cover are calculated based on changes in profit. gcamland only includes this land allocation calculation. gcamland is not run when GCAM is run; GCAM is not run when gcamland is run. Instead, gcamland replicates the land allocation equations used in GCAM so that we can isolate that part of the code for analysis and uncertainty quantification. See also response to comment #19.

Author Changes: We have added a footnote to the methodology section clarifying the relationship between GCAM and gcamland: "GCAM and gcamland are separate models. While gcamland replicates the land allocation mechanism in GCAM, it is not run within GCAM. Similarly, GCAM is not run as a part of gcamland. gcamland only includes a representation of land allocation. GCAM includes representations of agricultural supply and demand, land allocation, and other sectors (energy, water, economy, climate). The land allocation mechanism within gcamland uses price, yield, cost, subsidy, logit exponents, expectation parameters, and initial land area as exogenous inputs and endogenously determines land area in subsequent years. Changes in demand are explicitly represented in GCAM. In gcamland, changes in demand are captured through changes in price. For example, the increase in demand for corn and soybean due to biofuels policy is captured through changes in the prices of these goods."

2. The model clearly uses a nesting logit format, perhaps three nests. Equation 1 of the paper shows only on nest. The formula should be replaced with a formula for the full nest.

Author Response: We have added information on how profit and shares are calculated for the other nests.

Author Changes: We have added additional text and an equation "In the three-level nest version, land allocation at each level is determined by a modified version of equation 1, where Y is replaced by the land allocated to that particular nest. The land allocated to a particular nest is dynamic and varies over time. Profit rates (rj) at the lowest level of the nest are computed based on price, cost, yield, and subsidy (if included) for land use types (crops, pasture, commercial forest); profit rates for land cover types are input into the model and are based on the value of land. Profit rates for higher levels of the nest (rnode) are determined by: r_node= $[\sum\_(j = 1\Theta n(\lambda\_j\ r\_j\ )\hat{}\ ]\hat{}(1/)..$"

3. How the land constraint/constraints is/are defined? Does a simple land constraint directly add all types of land: Total land =forest + pasture + corm + soy+ etc.? or each nest has its own land constraint?

Author Response: The only explicit constraint on land in the model is on total land. That is, we require the sum of all land types (forest, pasture, grassland, shrubland, urban, crops, etc.) to equal the total area in the United States. We parameterize the model to prevent expansion of cropland into non-arable lands (urban, tundra, and rock/ice/desert).

Author Changes: We have clarified this in section 2.1.2: "The land allocated to a particular nest is dynamic and varies over time."

4. It is not clear how the estimation process is defined to estimate these parameters. Does the process estimate all the distribution parameters (s) simultaneously or individually?

Author Response: We are using Latin Hypercube Sampling, which estimates all parameters simultaneously.

Author Changes: We have added the following note to section 2.2: "Latin Hypercube Sampling draws all parameters simultaneously from uniform distributions."

5. How distribution parameters () were perturbed? Are they coming from given distribution? If yes, what type of distribution? Is this a random selection of three values limited between 0.01 and 3?

Author Response: The parameters were sampled assuming a uniform distribution. For logit exponents, we choose values for each of the three exponents randomly between 0.01 and 3. Each of the three exponents can have a different value and we do 10,000 samples for each expectation type and model configuration.

Author Changes: We have added the following note to section 2.2: "Latin Hypercube Sampling draws all parameters simultaneously from uniform distributions."

6. Over time total area of agricultural land in the US has declined sharply, due to conversion to non-agricultural uses of land (urbanization, infrastructure, . . ..). How gacamland handles conversion of land to non-forestry-ag land. How land availability land has been taken care off over time? Is it an exogenous variable in each year?

Author Response: While urban land has grown over time, the definition of urban land in gcamland accounts for only 1% of total land area in the United States. We hold this area constant in the simulations presented here (equal to 1975, 1990 or 2005 values depending on the calibration year used in the simulation). Allowing this to change over time would not have a noticeable impact on results given how small the area is. Similarly, we hold tundra and rock/ice/desert constant in time, but they account for very small amounts of land in the United States (2.6% and 0.4%, respectively). All other land is included in the economic competition.

Author Changes: We have added a footnote to the methodology section: "A small amount of land ($\sim$4%) is considered non-arable in gcamland in the United States, including urban, tundra, rock, ice, and desert. This land is held constant throughout the simulation time period."

7. A big issue in land use modeling is marginal cropland (idled land under CRP, cropland pasture, other types of idled). Area of idled cropland in the US have changed a lot. The definition of "cropland pasture" has also changed over time. How idled land is treated?

Author Response: Idled land is called "Other Arable Land" in gcamland. The amount of land in this category can change over time based on economic signals. Since idled cropland does not produce a product, its profit rate is exogenously specified like other land cover types (see response to comment #12). Note that this exogenous value is similar to a CRP payment, as it represents a marginal benefit of keeping land fallow.

Author Changes: We have added a footnote to the methodology section: "Fallow cropland (called other arable land in gcamland) is also included in this nest." We have also added text to the supplemental material indicating that idled cropland is included in gcamland.

8. It is noted that harvested area from FAO is used. FAO is missing many feed crops since 2011, including million hectares of those crops. without proper steps to cover missing crops in FAO, the estimated parameters will be subject to major issues and biases. Figure three suggest that those feed crops is missed. That is a major issue.

Author Response: gcamland includes fodder and feed crops, using data from FAO prior to 2011. We have excluded it from the comparison and statistics because the data is not available after 2011 as you noted, but it is included in the modelled results.

Author Changes: We have revised the Figure 3 caption to clarify this: "Note that fodder crops are included in gcamland but are excluded from total cropland area in this figure due to data limitations." We have also added figures to the supplemental material showing all crops, including fodder.

9. GCAM is using commodity price to model land allocation. It seems wholesale farm prices is used. That is a bad proxy for exporting crops such as cone and soybeans. For example, half of soybean is exported at much higher price farm price.

Author Response: The producer price is the relevant price signal to be used for planting decisions. The market price, and thus the producer and consumer prices, is a function of the demand sectors as well, which includes domestic demand and exports. However,

the resulting equilibrium price paid to producers is the relevant price regardless of how the demand is determined. Therefore, we feel that producer price is the right input into gcamland for this analysis.

Author Changes: We have revised the manuscript to better document what is included in gcamland. See also the responses to comment #1 and comment #19.

10. In this paper, in one case, subsidy has been examined in a sensitivity test. Subsidy is the key item in deriving land use, land rent, and the price received by farmers. The distribution parameters of the logit should be evaluated with subsidies. Sensitivity test is meaningless. The key here is to capture all types of subsidies paid to famers in the estimation processes.

Author Response: As noted in the manuscript, the subsidy data does not improve the estimation either because these subsidies do not affect cropland area (as suggested by Weber and Key 2012 and discussed in Section 5.1) or due to the quality of the data. There are not continuous, complete, and consistent data sets for all types of subsidies paid to farmers. Additionally, we only found crop-specific information on direct payments, making the inclusion of other types of subsidies difficult.

Author Changes: We added more information to Section 2.4.2 describing the limitations of the data: "Subsidies are a reality of crop agriculture in the United States. However, there are not continuous, complete, and consistent data sets for all types of subsidies paid to farmers. Additionally, crop-specific information (of the type needed for gcamland) is only available for direct payments, making the inclusion of other types of subsidies difficult" and our choice to make this a sensitivity "Because this data is inconsistent and incomplete, we only use it as a sensitivity in this paper and do not include it in the primary analysis."

11. How biofuels were included in the simulations? Biofuels and biofuel policies were major drives of land use. How that included in your simulations

[Figure]

Author Response: Biofuels and biofuels policy are reflected in our model through changes in producer prices of crops.

Author Changes: We have added this to the footnote explaining differences between GCAM and gcamland: "Changes in demand are explicitly represented in GCAM. In gcamland, changes in demand are captured through changes in price. For example, the increase in demand for corn and soybean due to biofuels policy is captured through changes in the prices of these goods."

12. Th dapper highlights that gcamland uses commodity prices in land allocation. But the model allocates land across land cover items. What prices are used for forest products, livestock products, etc.? The paper is silent on these prices. What prices were used for land cover items

Author Response: We calculate land rental prices for commercial forest and pasture using their product (forest or livestock products) prices and related productivity and cost information. For non-commercial land cover only items, effective profit rates are derived during the calibration process to ensure that the amount of land area in the base year predicted by the logit equation matches the read in value. For subsequent years, these effective profit rates are held constant. This estimation is described in detail in Wise et al. (2014).

Author Changes: We have added this information to the methodology section: "Profit rates ($r_j$) at the lowest level of the nest are computed based on price, cost, yield, and subsidy (if included) for land use types (crops, pasture, commercial forest); profit rates for land cover types are input into the model and are based on the value of land."

13. Regarding forestry, how gcamland treats forest land. Is it operates based on managed forest? Managed + unmanaged? How it treats unmanaged forest with no economic output.

Author Response: We include both managed and unmanaged forestland. For managed forestland, we use price and yield to calculate profit. For unmanaged, see response to comment #12.

Author Changes: See response to comment #12.

14. GCAM and gcamland are not forestry models. Forestry is not an annual crop. How these models take care of forestry in a dynamic setting. Do these models treat forestry as an annual crop?

Author Response: GCAM, gcamland, and other similar models assume that you need to set aside land at every timestep to ensure that you will have enough commercial forestland to meet harvest demand at the time the forest matures. To do this in GCAM, we assume that the amount of land needed for forest is equal to the wood product demand divided by the yield times the rotation length. So, if you need 1 Ha of land to meet wood product demand in a given year and the rotation length is 25 years, we set aside 25 Ha of land in that year. gcamland uses a similar paradigm, only we don't model demand, just yield and area. So, gcamland uses a yield that is equal to the harvest yield divided by the rotation length.

Author Changes: We have added this information to the methodology section: "Note that, since forestland is not an annually planted and harvested commodity, GCAM, gcamland, and other similar models assume that land must be set aside at every timestep to ensure enough commercial forestland is available to meet harvest demand at the time the forest matures. To do this in gcamland, we assume that the amount of land allocated to forest depends on the harvest yield and the rotation length."

15. In each case, the model is solved for a range of parameters. Then a set of parameters that minimizes NRMSE is selected. But NRMSE is defined for a single crop. How this variable is aggregated over crops? How NRMSE is calculated for non-cropland (e.g. forest, pasture, grass land, and etc.)? How cropland and non-cropland aggregated?

Author Response: We take the average of crop-specific NRMSE to get the aggregated NRMSE. As noted in the manuscript, we do not include non-cropland in our calculation of NRMSE.

Author Changes: We have clarified this in the methodology by adding "For NRMSE and RMSE" before the sentence that describes the averaging across crops.

16. How productivity of non-cropland is measured?

Author Response: We include productivity of pasture and forest, but not of other non-cropland types.

Author Changes: We have added this information to the methodology section: "Profit rates (rj) at the lowest level of the nest are computed based on price, cost, yield, and subsidy (if included) for land use types (crops, pasture, commercial forest); profit rates for land cover types are input into the model and are based on the value of land."

17. GCAM aggregates crops into some specific categories? How prices were generated for those categories. In many cases there is no data on crop prices?

Author Response: We use the weighted average of producer prices for aggregated commodities, weighting by the production.

Author Changes: We have added this information to Section 2.4.2: "Data was aggregated from individual crops to the GCAM/gcamland commodity groups, weighting crops by their production quantity"

18. The paper provides mixed messages on endogenous and exogenous variable. In determining targeted distribution parameters, what variables were targeted and what variables were determined in the model. It seems prices, areas, and yields were exogenous. Be more specific.

Author Response: Within gcamland, prices, costs, yields, subsidies, logit exponents, and expectation parameters are exogenous. In addition, the land area in the calibration

year is exogenous. Areas in subsequent years are endogenous. Within the experiment in this paper, we also varied logit exponents and expectation parameters as part of the ensemble sampling.

Author Changes: We have added this information to a footnote in the methodology section: "The land allocation mechanism within gcamland uses price, yield, cost, subsidy, logit exponents, expectation parameters, and initial land area as exogenous inputs and endogenously determines land area in subsequent years."

19. The whole practice implicitly assumes that other model parameters are accurate and valid. This is a strong assumption. The land supply parameters were determined while demand parameters held constant. The estimated supply parameters will be entirely wrong if the demand parameters (e.g. income and price elasticities for crops, livestock products, and forestry) are not valid. Any change in the demand parameters could alter your estimated parameters for the land supply. Can you test sensitivity of your results with respect to changes in other elasticities of the model?

Author Response: As noted in our response to comment #1, gcamland does not include a representation of demand for the exact reason you note here. We have chosen to isolate the land allocation mechanism in gcamland to ensure we get the right parameters for the right reasons and do not have cancelling errors. We cannot do a sensitivity on demand elasticities since they are not included in the model at all. Price is the only link to demand and we are using observed prices from FAO to ensure that demand-side sensitivity and errors do not affect the parameter estimation on the supply-side.

Author Changes: See response to #18.

20. The results are counterintuitive. Let me explain using figure 2. In the adaptive case, for the first two nests the values are about 0.4 and for the last nest (cropland) the value of is about 0.6. Given that limited land movements among land cover items occurred at national level in the US and lots of change occurred in the crop nest, one could justify this outcome. However, for the other three cases (hybrid, linear, and perfect) the

ranking is of values shows revers. Meaning that land conversion is easier at the land cover nests than the cropland land nest. These outcomes do not make sense. Am I missing something?

Author Response: Those outcomes also get lower NRMSE than the adaptive expectations, indicating that they do not explain historical land allocation as well as adaptive expectations. The parameter sets for hybrid, linear, and perfect minimize NRMSE if those expectation types are assumed, but the model with the lowest NRMSE includes adaptive expectations and parameters that match our intuition.

Author Changes: An explanation of the results and intuition is provided in section 4.1.

21. In showing the results, level variables were used to show errors. For example, figure 2 compares estimated harvested areas with their observations for four types of expectations. This hides the errors involved. It is better to calculate errors as percent difference between the estimated and observed areas.

Author Response: Thank you for the suggestion. We have considered adding a figure on percentage difference to the paper (attached). However, we feel that showing absolute values compared to observations is more informative since it gives a sense of scale, which differs significantly across crops in the USA.

Author Changes: We have added figures comparing model results to observations for all commodities to the supplemental material, but have opted not to add figures showing percentage difference.

22. The main manuscript only presents comparison of the projected and observed harvested areas and provided no comparison for other land types.

Author Response: The comparison of projected and observed area for other land types is included in the supplemental material for types where observations are available.

Author Changes: The comparison of projected and observed area for other land types is included in the supplemental material for types where observations are available. We

have also added a paragraph to the supplemental material stating which land types are in gcamland and explaining our choice of what to show where: "The main text of this paper focuses on four commodity groups (Corn, Wheat, OtherGrain, and OilCrop), as these four commodities represent the largest land area in the United States. However, gcamland includes twelve commodity groups in total, representing all crops reported by the FAO, and fallow or idled cropland (referred to as other arable land in gcamland). In addition, gcamland includes commercial forest and pasture, as well as several other land cover types, including forest, grassland, shrubland, tundra, rock/ice/desert, and urbanland. We include results for other agricultural commodities and the land cover types where observations are available in this section"

23. Results are highly aggregated into four groups of crops. how about the 12 categories of crops in GCAM?

Author Response: The main manuscript shows four of the 12 crop categories. We have included figures showing the other categories to the supplemental material.

Author Changes: The main manuscript shows four of the 12 crop categories. We have included figures showing the other categories to the supplemental material. We have also added an explicit reference to these figures in the main text.

24. The figure S5 of SI shows major errors for the change in forest area. This show that the model fails to represent changes in forest area correctly.

Author Response: Figure S5 had included the net change from 1990 to 2015 for the modeled data and the net change from 1992 to 2015 for observations for forest. Figure S7 shows the whole time series. As shown in S7, the time series tracks the observations fairly closely for the adaptive expectations. We do not think it is correct to say that the model fails to represent changes in forest area; however, we do think that Figure S5 was confusing given the unit used and the differences in time horizon.

Author Changes: We have removed Figure S5 as it did not add any new information

and was confusing (see response to comment #25).

25. The figure S5 show increases in all land cover types and harvested areas. How that could be possible?

Author Response: Figure S5 shows the ratio of land in 2015 to land in 1990. Values less than 1 indicate a reduction in land area over time. There are several land types where area declined and those land types have values less than 1. However, we do think the unit used in this figure was confusing and all of the information contained in this figure is also shown in Figure S7, so we have removed this figure.

Author Changes: See response #24.

26. Figure S7 shows no results for land cover items including grassland and shrubland for three types of expectations. Why?

Author Response: These results are included but they are covered by the line for adaptive expectations. All four expectation types produce similar values.

Author Changes: We have improved the figure so that the other expectation types are visible and added a note to the caption indicating they are overlapping.

27. Figure S7 shows major errors for grassland and shrubland in the adaptive approach project huge errors. Why?

Author Response: This is due to differences in the definition of grass and shrubland between gcamland and the CCI land cover product. Most notably, gcamland includes a large portion of what is categorized as grass or shrub in CCI land cover as pasture.

Author Changes: This was already stated in the paragraph immediately preceding figure S7: "Due to differences in definitions of land cover between gcamland and the CCI land cover product, Grassland and Shrubland do not match in absolute value between gcamland and the observation data." We have also added a panel to this figure showing the sum of pasture, shrubland and grassland; this summation improves upon the

data mismatch.

28. It seems the whole practice has failed to take care of land cover changes.

Author Response: Land cover changes are included in gcamland and are compared to observations in the Supplemental Material. Therefore, we do not think it is correct to say that we have failed to take care of land cover changes.

Author Changes: See changes in response to comments 12, 13, 16, 22, 24, 25, 26, 27.

29. The examined practice estimated a few parameters of the model for land use. A good way to test the outcomes of this practice is to run the GCAM model with the estimated parameters and compare the model results for land use changes, land cover changes, changes in crop prices, and changes in yield with actual observations over the examined period.

Author Response: We agree that this is a useful test and intend to do it in subsequent work. However, this paper is focused on parameter estimation and we think that adding those simulations to this paper would unnecessarily complicate the existing text and analysis.

Author Changes: None.

30. Finally, the whole work could be a valuable practice for the CGAM community. It uses "hindcast" to estimate the logit distribution parameters for this model. Hindcast Is not a new approach. The outcome of this practice may help the GCAM community to improve their work on land use modeling. However, the results of this practice may be not useable for other models. As they may follow very different modeling structure and assumptions. The author of this paper should make this point very carefully.

Author Response: We agree with the reviewer's notes here. Other models can use from our methodology and some of the takeaways from this paper (e.g., the choice of metric and which crops to include can alter the resulting parameters and model

performance); however, other models are unlikely to be able to use these parameters directly.

Author Changes: We've added this caveat to the discussion.

31. The abstract provides trivial information. It is not an abstract of this paper.

Author Response: We have revised the abstract.

Author Changes: We have revised the abstract.

32. Following a summary of land use change at the global scale, the second paragraph of the introduction begins with: "Similar trends occurred in the United States". This is not an accurate statement. The US land use change did not follow the global land use changes in terms of land conversion to crop production. No expansion in cropland has been observed for the cases of US.

Author Response: We have removed that sentence.

Author Changes: We have removed that sentence.
* * *
[Figure]

**Fig. 1.**

---

## Referee Report (RR1)

**Responses to authors**
**Revised mmanuscript entitled:**

**Modeling Land Use and Land Cover Change: Using a Hindcast to Estimate Economic Parameters in gcamland v2.0**

**Geoscientific Model Development**

*Anonymous*

In what follows I responds to the authors using green fonts

**Anonymous Referee #2**

Summary of the paper This paper uses gcamland/GCAM to calibrate/estimate/tune the land distribution parameters of a nested logit land allocation function used in this model. In the lack of econometrically estimated values for these parameters, it is an important effort to accomplish this task. However, the paper suffers from some important deficiencies and lack of clarity, in particular for those who do not know this mode.

**Author Response**: Thank you for your detailed review of our paper. Your comments have pointed out a number of places where we did not clearly describe our methodology.

**Author Changes**: We have made a number of changes to the paper in response to your comments and those of another reviewer. Those changes are detailed below.

Response: Thanks for your attention. Your responses helped me to understand your work better. However, in some cases your responses were not convincing. In what follows I highlight my comments using green fonts after each response.

Some important comments:
1.  I am not a GCAM modeler but is seems gcamland operates under GCAM. For nonGCAM community the links and interactions between these two models are not clear. How they linked and interact. A simple chart can help.

**Author Response**: GCAM and gcamland are completely separate models. GCAM includes representations of energy, water, land, and climate. It includes a land allocation mechanism where land use and land cover are calculated based on changes in profit. gcamland only includes this land allocation calculation. gcamland is not run when GCAM is run; GCAM is not run when gcamland is run. Instead, gcamland replicates the land allocation equations used in GCAM so that we can isolate that part of the code for analysis and uncertainty quantification. See also response to comment #19.

**Author Changes**: We have added a footnote to the methodology section clarifying the relationship between GCAM and gcamland: "GCAM and gcamland are separate models. While gcamland replicates the land allocation mechanism in GCAM, it is not run within GCAM. Similarly, GCAM is not run as a part of gcamland. gcamland only includes a representation of land allocation. GCAM includes representations of agricultural supply and demand, land allocation, and other sectors (energy, water, economy, climate). The land allocation mechanism within gcamland uses price, yield, cost, subsidy, logit exponents, expectation parameters, and initial land area as exogenous inputs and endogenously determines land area in subsequent years.

Changes in demand are explicitly represented in GCAM. In gcamland, changes in demand are captured through changes in price. For example, the increase in demand for corn and soybean due to biofuels policy is captured through changes in the prices of these goods."

Thanks, it seems the CGAM model plays no role in the estimation process of the estimated parameters and only the gcamland with no demand side has been used. If that is the case, then the paper needs revisions to make sure that CGAM has not been used in the estimation process. For example, here are a few places that need attention:

- Table 2, Description of Arable Land should refer to "gcamland" not "GCAM", as noted in footnote 5.
- Change title of 2.1.2 to "Economic approach in gcamland"
- Change title of section 2.3 to "Evaluation gcamland model performance"
- Change the end of line 204 to "… focused on annual time step in gcamland"

2. The model clearly uses a nesting logit format, perhaps three nests. Equation 1 of the paper shows only on nest. The formula should be replaced with a formula for the full nest.

**Author Response**: We have added information on how profit and shares are calculated for the other nests.

**Author Changes**: We have added additional text and an equation "In the three-level nest version, land allocation at each level is determined by a modified version of equation 1, where Y is replaced by the land allocated to that particular nest. The land allocated to a particular nest is dynamic and varies over time. Profit rates ($r_j$) at the lowest level of the nest are computed based on price, cost, yield, and subsidy (if included) for land use types (crops, pasture, commercial forest); profit rates for land cover types are input into the model and are based on the value of land. Profit rates for higher levels of the nest ($r_{node}$) are determined by: $r_{node} = [\sum_{j=1}^{n}(\lambda_j r_j)^\rho]^{1/\rho}$

The revision mentioned here is helpful, but it provides a broken formulation and misrepresent your work. To be fully transparent and to make sure that the readers understand that three exponent parameters (distribution parameters) and there sets of share parameters were estimated, I propose to use the following format:

$X_{jt}^C = \frac{(\lambda_{jo}^c r_{jt}^c)^{\rho^C}}{\sum_j (\lambda_{jo}^c r_{jt}^c)^{\rho^C}} \cdot Y_t^C$ where "C" represents the crop nest including rice, wheat, ……

$X_{jt}^A = \frac{(\lambda_{jo}^A r_{jt}^A)^{\rho^A}}{\sum_j (\lambda_{jo}^A r_{jt}^A)^{\rho^A}} \cdot Y_t^A$ where "A" represents the ag-forestry nest including cropland, forest, ….

$X_{jt}^R = \frac{(\lambda_{jo}^R r_{jt}^R)^{\rho^R}}{\sum_i (\lambda_{jo}^R r_{jt}^R)^{\rho^R}} \cdot Y_t^R$ where "R" represents the arable land nest including …..

The members of each nest should be clearly mentioned. In particular, for the crop nest all members of this nest including unused land, cropland pasture, CRP, other idled land should be specified. The information that presented in Table 2 is incomplete and very misleading. Of course, the names of variables should be described fully and maintain this convention through the paper.

3. How the land constraint/constraints is/are defined? Does a simple land constraint directly add all types of land: Total land =forest + pasture + corm + soy+ etc.? or each nest has its own land constraint?

**Author Response**: The only explicit constraint on land in the model is on total land. That is, we require the sum of all land types (forest, pasture, grassland, shrubland, urban, crops, etc.) to equal the total area in the United States. We parameterize the model to prevent expansion of cropland into non-arable lands (urban, tundra, and rock/ice/desert).

**Author Changes**: We have clarified this in section 2.1.2: "The land allocated to a particular nest is dynamic and varies over time."

Thanks for the clarifications that: 1) only one land constraint is used and 2) land transformation from non-arable lands to cropland is prohibited. However, these clarifications raised the following important concerns:

- First, when it goes to crops how do you distinguish between harvested area and planted area? There is no information for planted area for all crops. If harvested area is used for each crop, then what do you do with multiple cropping, crop failure. You cannot simply add up harvested area and say that the sum is total cropland. Please be very specific in your response.
- According to your response, non-arable land cannot be converted to cropland. You said some kinds of parameterizations were used. Please explain it more transparently and be very specific in your response. How about the revers land transformation: conversion of cropland to non-arable land? How do you model that? Please, again be precise in your response and avoid the general statement of based on profitability.

4. It is not clear how the estimation process is defined to estimate these parameters. Does the process estimate all the distribution parameters (s) simultaneously or individually?

**Author Response**: We are using Latin Hypercube Sampling, which estimates all parameters simultaneously.

**Author Changes**: We have added the following note to section 2.2: "Latin Hypercube Sampling draws all parameters simultaneously from uniform distributions."

Thanks, for this clarification. Please let me concentrate on the three logit distribution parameters and use the notation mentioned above. Based on this response and the response to my next comment, it seems the implemented process uses the following steps:

- Randomly selects three values for the three distribution parameters of $\rho^R, \rho^A and \rho^C$ from three uniform distributions (each ranged between 0.01 to 3).
- Given the selected parameters of $\rho^R, \rho^A and \rho^C$, land allocation for the whole time period is determined
- The above two steps repeated for 10,000 collections for each expectation type.
- A collection, including three distribution parameters, that minimizes the size of NRMSE is selected as the best estimate for the distribution parameters under each examined expectation type.

5. How distribution parameters () were perturbed? Are they coming from given distribution? If yes, what type of distribution? Is this a random selection of three values limited between 0.01 and 3?

**Author Response**: The parameters were sampled assuming a uniform distribution. For logit exponents, we choose values for each of the three exponents randomly between 0.01 and 3. Each of the three exponents can have a different value and we do 10,000 samples for each expectation type and model configuration.

**Author Changes**: We have added the following note to section 2.2: "Latin Hypercube Sampling draws all parameters simultaneously from uniform distributions."

Thanks for this clarification. See my previous response.

6.  Over time total area of agricultural land in the US has declined sharply, due to conversion to non-agricultural uses of land (urbanization, infrastructure, . . ..). How gacamland handles conversion of land to non-forestry-ag land. How land availability land has been taken care off over time? Is it an exogenous variable in each year?

**Author Response**: While urban land has grown over time, the definition of urban land in gcamland accounts for only 1% of total land area in the United States. We hold this area constant in the simulations presented here (equal to 1975, 1990 or 2005 values depending on the calibration year used in the simulation). Allowing this to change over time would not have a noticeable impact on results given how small the area is. Similarly, we hold tundra and rock/ice/desert constant in time, but they account for very small amounts of land in the United States (2.6% and 0.4%, respectively). All other land is included in the economic competition.

**Author Changes**: We have added a footnote to the methodology section: "A small amount of land (~4%) is considered non-arable in gcamland in the United States, including urban, tundra, rock, ice, and desert. This land is held constant throughout the simulation time period."

Thanks for this clarification. However, it does not help and makes further confusion.

- The paper does not provide a clear picture of the land categories in the model. While you are talking about non- arable land as a part of land use modeling, the logit nesting structure does not cover it. If this type of land is not modeled, why it is described in the paper.
- Table 1 introduces three nests: Arable land; Ag land. Forest, and other; and Cropland (or crops). This plus other information provided in the tale bring in mind the following nesting structure:

[Figure]

However, there is no "*cropland*" in the middle nest. What is the relationship between Ag land, cropland, and pasture land? It seems a nest is missing.
- It is noted FAO data with some modification is used to determine land data. If that is the case, then something is not correct. Please let me explain. Arable land is a subset of cropland in the FAO data. For example, in the FAO data for 2018, area of US cropland is

160,437 thousand hectares with sub component of 157,737 thousand hectare of arable land. This is in a sharp contrast with the nesting structure defined in the paper. Also, area of US Ag land is 2.6 times of its arable land. How Ag land could be a subset of arable land?

The paper should make proper clarifications on this issue.

Finally, let us follow the nesting structure outlined in the paper. According to the FAO data, total area of US arable land has declined from 180,630 thousand hectares in 1961 to 157,727 thousand hectares in 2018, a decline by 22,893 thousand hectares. How this reduction has been handled? It is not a small reduction.

7. A big issue in land use modeling is marginal cropland (idled land under CRP, cropland pasture, other types of idled). Area of idled cropland in the US have changed a lot. The definition of "cropland pasture" has also changed over time. How idled land is treated?

**Author Response**: Idled land is called "Other Arable Land" in gcamland. The amount of land in this category can change over time based on economic signals. Since idled cropland does not produce a product, its profit rate is exogenously specified like other land cover types (see response to comment #12). Note that this exogenous value is similar to a CRP payment, as it represents a marginal benefit of keeping land fallow.

**Author Changes**: We have added a footnote to the methodology section: "Fallow cropland (called other arable land in gcamland) is also included in this nest." We have also added text to the supplemental material indicating that idled cropland is included in gcamland.

This is a misleading response. FAO data base does not have other arable land. Please at least, provide a table and put all types of included land in each nest at list for three years, including cropland and its components.

8. It is noted that harvested area from FAO is used. FAO is missing many feed crops since 2011, including million hectares of those crops. without proper steps to cover missing crops in FAO, the estimated parameters will be subject to major issues and biases. Figure three suggest that those feed crops is missed. That is a major issue.

**Author Response**: gcamland includes fodder and feed crops, using data from FAO prior to 2011. We have excluded it from the comparison and statistics because the data is not available after 2011 as you noted, but it is included in the modelled results.

**Author Changes**: We have revised the Figure 3 caption to clarify this: "Note that fodder crops are included in gcamland but are excluded from total cropland area in this figure due to data limitations." We have also added figures to the supplemental material showing all crops, including fodder.

This is a misleading response. If you fixed the data (for after 2011) by estimating the missing data items, then that should be reported and included in your figures. Transparency is the key.

9. GCAM is using commodity price to model land allocation. It seems wholesale farm prices is used. That is a bad proxy for exporting crops such as cone and soybeans. For example, half of soybean is exported at much higher price farm price.

**Author Response**: The producer price is the relevant price signal to be used for planting decisions. The market price, and thus the producer and consumer prices, is a function of the demand sectors as well, which includes domestic demand and exports. However, the resulting equilibrium price paid to producers is the relevant price regardless of how the demand is determined. Therefore, we feel that producer price is the right input into gcamland for this analysis.

**Author Changes**: We have revised the manuscript to better document what is included in gcamland. See also the responses to comment #1 and comment #19.

Yes, as it is noted in this response "price paid to producers is the relevant price", but the FAO whole sale price does not reflect the price received by farmers. You need to address this as a major limitation of your work clearly. Please explain what revision you have made? And be more specific.

10.   In this paper, in one case, subsidy has been examined in a sensitivity test. Subsidy is the key item in deriving land use, land rent, and the price received by farmers. The distribution parameters of the logit should be evaluated with subsidies. Sensitivity test is meaningless. The key here is to capture all types of subsidies paid to famers in the estimation processes.

**Author Response**: As noted in the manuscript, the subsidy data does not improve the estimation either because these subsidies do not affect cropland area (as suggested by Weber and Key 2012 and discussed in Section 5.1) or due to the quality of the data. There are not continuous, complete, and consistent data sets for all types of subsidies paid to farmers. Additionally, we only found crop-specific information on direct payments, making the inclusion of other types of subsidies difficult.

**Author Changes**: We added more information to Section 2.4.2 describing the limitations of the data: "Subsidies are a reality of crop agriculture in the United States. However, there are not continuous, complete, and consistent data sets for all types of subsidies paid to farmers. Additionally, crop-specific information (of the type needed for gcamland) is only available for direct payments, making the inclusion of other types of subsidies difficult" and our choice to make this a sensitivity "Because this data is inconsistent and incomplete, we only use it as a sensitivity in this paper and do not include it in the primary analysis."

Thanks for this response. However, it is incomplete. The paper should clearly acknowledge that improper inclusion of subsidies could make major biases in the outcome of the estimation process and that a sensitivity test is not a proper way of fixing this issue.

11.   How biofuels were included in the simulations? Biofuels and biofuel policies were major drives of land use. How that included in your simulations

**Author Response**: Biofuels and biofuels policy are reflected in our model through changes in producer prices of crops.

**Author Changes**: We have added this to the footnote explaining differences between GCAM and gcamland: "Changes in demand are explicitly represented in GCAM. In gcamland, changes in demand are captured through changes in price. For example, the increase in demand for corn and soybean due to biofuels policy is captured through changes in the prices of these goods."

This response makes me more concern. I am not convinced that your work properly identifies source of piece changes. You approach only consider changes in the prices and send that as a signal to the land supply tree, without identifying the sources of price change. The source of price change could be demand shock (e.g. biofuels) or supply shock (reduction in yield or land supply). You approach does not distinguish these sources and simply consider price changes as signals to the land supply. Can you convince the readers that you do not need to identify the source of price change?

12.   Th dapper highlights that gcamland uses commodity prices in land allocation. But the model allocates land across land cover items. What prices are used for forest products, livestock products, etc.? The paper is silent on these prices. What prices were used for land cover items

**Author Response**: We calculate land rental prices for commercial forest and pasture using their product (forest or livestock products) prices and related productivity and cost information.  For non-commercial land cover only items, effective profit rates are derived during the calibration process to ensure that the amount of land area in the base year predicted by the logit equation matches the read in value. For subsequent years, these effective profit rates are held constant. This estimation is described in detail in Wise et al. (2014).

**Author Changes**: We have added this information to the methodology section: "Profit rates ($r_j$) at the lowest level of the nest are computed based on price, cost, yield, and subsidy (if included) for land use types (crops, pasture, commercial forest); profit rates for land cover types are input into the model and are based on the value of land."

- This response makes no clarification. For example, it is said that: "We calculate land rental prices for commercial forest and pasture using their  product (forest or livestock products) prices and related productivity and cost information." Please be very specific and answer the following: What are the prices of livestock products? How about prices of forest products? How do you measure yield for forest? How about yield for pasture land?

- If only arable land is modeled, why do you need profit rates for non-commercial land. As noted in my earlier comments, providing a table including all nests and their members could help.

- What is the effective rate for unmanaged forest? How do you determine it in the calibration prices? Calibration to what?

- The paper should clearly explain the above points

13.   Regarding forestry, how gcamland treats forest land. Is it operates based on managed forest? Managed + unmanaged? How it treats unmanaged forest with no economic output.

**Author Response**: We include both managed and unmanaged forestland. For managed forestland, we use price and yield to calculate profit. For unmanaged, see response to comment #12.

**Author Changes**: See response to comment #12.

What are the price and yield of managed forest? How do you determine them? Based on what data do you calculate price of forest and yield of managed forest? The paper should explain these.

14.  GCAM and gcamland are not forestry models. Forestry is not an annual crop. How these models take care of forestry in a dynamic setting. Do these models treat forestry as an annual crop?

**Author Response**: GCAM, gcamland, and other similar models assume that you need to set aside land at every timestep to ensure that you will have enough commercial forestland to meet harvest demand at the time the forest matures. To do this in GCAM, we assume that the amount of land needed for forest is equal to the wood product demand divided by the yield times the rotation length. So, if you need 1 Ha of land to meet wood product demand in a given year and the rotation length is 25 years, we set aside 25 Ha of land in that year. gcamland uses a similar paradigm, only we don't model demand, just yield and area. So, gcamland uses a yield that is equal to the harvest yield divided by the rotation length.

**Author Changes**: We have added this information to the methodology section: "Note that, since forestland is not an annually planted and harvested commodity, GCAM, gcamland, and other similar models assume that land must be set aside at every timestep to ensure enough commercial forestland is available to meet harvest demand at the time the forest matures. To do this in gcamland, we assume that the amount of land allocated to forest depends on the harvest yield and the rotation length."

Sorry for my ignorance. But I do not know any model that follow the assumption mentioned above. In your revision, please name some well-knows models that follow this approach. If I understand correctly, what mentioned above can be replicated by the following formula:

Forest area in year t = yield of forest in t * demand for forest in t * 25.

This formula over simplifies the way that a forest model works and has no root in real world. You need to put this formula in the paper and justify it to be transparent.

15.  In each case, the model is solved for a range of parameters. Then a set of parameters that minimizes NRMSE is selected. But NRMSE is defined for a single crop. How this variable is aggregated over crops? How NRMSE is calculated for non-cropland (e.g. forest, pasture, grass land, and etc.)? How cropland and non-cropland aggregated?

**Author Response**: We take the average of crop-specific NRMSE to get the aggregated NRMSE. As noted in the manuscript, we do not include non-cropland in our calculation of NRMSE.

**Author Changes**: We have clarified this in the methodology by adding "For NRMSE and RMSE" before the sentence that describes the averaging across crops.

Thanks for these clarifications. I have difficulties to find out in what part of the manuscript it is noted that you do not include non-cropland in the calculation of NRMSE. Please clearly mention this point at the beginning of section 2.3 so that the reader can see this important limitation. Also, it is important to explain why you do not include non-cropland in the calculation. This is a surprise for me? Why you do not follow the same approach for other nest that you determine their distribution parameters. Indeed, as I mentioned before, you determine a selection of three parameters of $\rho^R, \rho^A$ and $\rho^C$. But it seems you ignore how good are your estimated for $\rho^R$ and $\rho^A$

and only care about $\rho^C$. In fact, you select a mix of $\rho^R$, $\rho^A$ and $\rho^C$ to minimize NRMSE over the crop nest. So, this means that you may get bad errors for the other two nests while you minimize only over crops. This is a serous problem. Indeed, you do not optimize for two nests out of three nests. This could cause major errors in the projection of land use changes in the non-cropland nests, compared to the observed data.

16. How productivity of non-cropland is measured?

**Author Response**: We include productivity of pasture and forest, but not of other non-cropland types.

**Author Changes**: We have added this information to the methodology section: "Profit rates ($r_j$) at the lowest level of the nest are computed based on price, cost, yield, and subsidy (if included) for land use types (crops, pasture, commercial forest); profit rates for land cover types are input into the model and are based on the value of land."

Thanks for this clarification, but you did not answer the question completely. Please explain how did you calculated productivity of forest and pasture land? It is an important piece of information and the reader should know it.

17. GCAM aggregates crops into some specific categories? How prices were generated for those categories. In many cases there is no data on crop prices?

**Author Response**: We use the weighted average of producer prices for aggregated commodities, weighting by the production.

**Author Changes**: We have added this information to Section 2.4.2: "Data was aggregated from individual crops to the GCAM/gcamland commodity groups, weighting crops by their production quantity"

Thanks for the clarification, but for some crops, such as fodder crops, no data is available for quantity of production. Please explain what you did for those crops.

18. The paper provides mixed messages on endogenous and exogenous variable. In determining targeted distribution parameters, what variables were targeted and what variables were determined in the model. It seems prices, areas, and yields were exogenous. Be more specific.

**Author Response**: Within gcamland, prices, costs, yields, subsidies, logit exponents, and expectation parameters are exogenous. In addition, the land area in the calibration year is exogenous. Areas in subsequent years are endogenous. Within the experiment in this paper, we also varied logit exponents and expectation parameters as part of the ensemble sampling.

**Author Changes**: We have added this information to a footnote in the methodology section: "The land allocation mechanism within gcamland uses price, yield, cost, subsidy, logit exponents, expectation parameters, and initial land area as exogenous inputs and endogenously determines land area in subsequent years."

Thanks for this clarification.

19. The whole practice implicitly assumes that other model parameters are accurate and valid. This is a strong assumption. The land supply parameters were determined while demand parameters held constant. The estimated supply parameters will be entirely wrong if the demand parameters (e.g. income and price elasticities for crops, livestock products, and forestry) are not valid. Any change in the demand parameters could alter your estimated parameters for the land supply. Can you test sensitivity of your results with respect to changes in other elasticities of the model?

**Author Response**: As noted in our response to comment #1, gcamland does not include a representation of demand for the exact reason you note here. We have chosen to isolate the land allocation mechanism in gcamland to ensure we get the right parameters for the right reasons and do not have cancelling errors. We cannot do a sensitivity on demand elasticities since they are not included in the model at all. Price is the only link to demand and we are using observed prices from FAO to ensure that demand-side sensitivity and errors do not affect the parameter estimation on the supply-side.

**Author Changes**: See response to #18.

Please see my response to item # 11

20. The results are counterintuitive. Let me explain using figure 2. In the adaptive case, for the first two nests the values are about 0.4 and for the last nest (cropland) the value of is about 0.6. Given that limited land movements among land cover items occurred at national level in the US and lots of change occurred in the crop nest, one could justify this outcome. However, for the other three cases (hybrid, linear, and perfect) the ranking is of values shows revers. Meaning that land conversion is easier at the land cover nests than the cropland land nest. These outcomes do not make sense. Am I missing something?

**Author Response**: Those outcomes also get lower NRMSE than the adaptive expectations, indicating that they do not explain historical land allocation as well as adaptive expectations. The parameter sets for hybrid, linear, and perfect minimize NRMSE *if those expectation types are assumed*, but the model with the lowest NRMSE includes adaptive expectations and parameters that match our intuition.

**Author Changes**: An explanation of the results and intuition is provided in section 4.1.

Thanks for this clarification.

21. In showing the results, level variables were used to show errors. For example, figure 2 compares estimated harvested areas with their observations for four types of expectations. This hides the errors involved. It is better to calculate errors as percent difference between the estimated and observed areas.

**Author Response**: Thank you for the suggestion. We have considered adding a figure on percentage difference to the paper (attached). However, we feel that showing absolute values compared to observations is more informative since it gives a sense of scale, which differs significantly across crops in the USA.

**Author Changes**: We have added figures comparing model results to observations for all commodities to the supplemental material, but have opted not to add figures showing percentage

difference.

This is not an appropriate response. It is an essential task to inform the readers regarding the percent errors between projections and observed values. Comparing the level variables in a chart hides those errors. to be transparent and informative you need to provide data on errors. If it is hard to show it in charts, please show them in a table in the appendix. This is an essential validation check.

22. The main manuscript only presents comparison of the projected and observed harvested areas and provided no comparison for other land types.

**Author Response**: The comparison of projected and observed area for other land types is included in the supplemental material for types where observations are available.

**Author Changes**: The comparison of projected and observed area for other land types is included in the supplemental material for types where observations are available. We have also added a paragraph to the supplemental material stating which land types are in gcamland and explaining our choice of what to show where: "The main text of this paper focuses on four commodity groups (Corn, Wheat, OtherGrain, and OilCrop), as these four commodities represent the largest land area in the United States. However, gcamland includes twelve commodity groups in total, representing all crops reported by the FAO, and fallow or idled cropland (referred to as other arable land in gcamland). In addition, gcamland includes commercial forest and pasture, as well as several other land cover types, including forest, grassland, shrubland, tundra, rock/ice/desert, and urbanland. We include results for other agricultural commodities and the land cover types where observations are available in this section"

Sorry, I do not consider this as a satisfactory response. The paper and its supporting material provide inconstant and confusing information about the land cover types, land uses, and components of each of the three nests included in the model. Simply revise table 2 and clearly put all land types and land uses in that table for each nest.

For example, in the main text in table 2 pasture land is not a part of middle nest. But it appears in the SI in Figure 7 as a component in the mix of grassland, shrubland, and pasture. Very confusing.

The main text should clearly represent the nesting structure and the component of each nest. Do not refer to another paper. This is an essential information for this paper.

Figure S7 should show data for each land cover item including managed forest, unmanaged forest, pasture, grassland, shrubland, any other components of the non-cropland nest, and cropland as one land cover type. If your model does not trace changes in some land types, that type of land should not be included in the model nor in the paper.

As a subcategory of cropland, the projections for unused cropland and their corresponding observations should be presented and compares.

23. Results are highly aggregated into four groups of crops. how about the 12 categories of crops in GCAM?

**Author Response**: The main manuscript shows four of the 12 crop categories. We have included

figures showing the other categories to the supplemental material.

**Author Changes**: The main manuscript shows four of the 12 crop categories. We have included figures showing the other categories to the supplemental material. We have also added an explicit reference to these figures in the main text.

Please, add changes in unused land. That is an important piece of information.

24.  The figure S5 of SI shows major errors for the change in forest area. This show that the model fails to represent changes in forest area correctly.

**Author Response**: Figure S5 had included the net change from 1990 to 2015 for the modeled data and the net change from 1992 to 2015 for observations for forest. Figure S7 shows the whole time series. As shown in S7, the time series tracks the observations fairly closely for the adaptive expectations. We do not think it is correct to say that the model fails to represent changes in forest area; however, we do think that Figure S5 was confusing given the unit used and the differences in time horizon.

**Author Changes**: We have removed Figure S5 as it did not add any new information and was confusing (see response to comment #25).

First thanks for adding figure S7. Adding this figure is a step forward. As I mentioned in comment # 23 you need to extend this figure for all land cover items. In particular, it is important to show errors in %, not in levels. Level variables hide errors. Please show the percent errors, then we can judge the model performance in land cover items. Also remember that you failed to calculate the goodness of fit (in your language NRSME) for land cover items.

Figure S7 shows bad performances for land cover items that already are included in this figure. In particular, I see very large differences in level variables between the performance and observed items. You also could alter the scale of this figure to better see the errors for forest. You should show the errors in percent to show the model performance. I believe it is straight forward to calculate NRSME for these items. Why not?

25. The figure S5 show increases in all land cover types and harvested areas. How that could be possible?

**Author Response**: Figure S5 shows the ratio of land in 2015 to land in 1990. Values less than 1 indicate a reduction in land area over time. There are several land types where area declined and those land types have values less than 1. However, we do think the unit used in this figure was confusing and all of the information contained in this figure is also shown in Figure S7, so we have removed this figure.

**Author Changes**: See response #24.

Thanks, see my responses to item # 24

26.  Figure S7 shows no results for land cover items including grassland and shrubland for three types of expectations. Why?

**Author Response**: These results are included but they are covered by the line for adaptive expectations. All four expectation types produce similar values.

**Author Changes**: We have improved the figure so that the other expectation types are visible and added a note to the caption indicating they are overlapping.

Thanks

27. Figure S7 shows major errors for grassland and shrubland in the adaptive approach project huge errors. Why?

**Author Response**: This is due to differences in the definition of grass and shrubland between gcamland and the CCI land cover product. Most notably, gcamland includes a large portion of what is categorized as grass or shrub in CCI land cover as pasture.

**Author Changes**: This was already stated in the paragraph immediately preceding figure S7: "Due to differences in definitions of land cover between gcamland and the CCI land cover product, Grassland and Shrubland do not match in absolute value between gcamland and the observation data." We have also added a panel to this figure showing the sum of pasture, shrubland and grassland; this summation improves upon the data mismatch.

Sorry, but this response does not make sense. If you used different definitions in your modeling practice, you should adjust the actual observations based on your definition as well. please revise accordingly. Even the combined panel shows large differences. We are talking about millions of $km^2$.

28. It seems the whole practice has failed to take care of land cover changes.

**Author Response**: Land cover changes are included in gcamland and are compared to observations in the Supplemental Material. Therefore, we do not think it is correct to say that we have failed to take care of land cover changes.

**Author Changes**: See changes in response to comments 12, 13, 16, 22, 24, 25, 26, 27.

I already responded to these. Your work fails to calculate the goodness of fit for land cover items. That should be highlighted in the main manuscript, as I highlighted in several places in my responses. Also, errors should be presented in % differences. This is an essential item to validate this work. given the size of land items, even 5% error is huge. We are talking about million hectares of land.

29. The examined practice estimated a few parameters of the model for land use. A good way to test the outcomes of this practice is to run the GCAM model with the estimated parameters and compare the model results for land use changes, land cover changes, changes in crop prices, and changes in yield with actual observations over the examined period.

**Author Response**: We agree that this is a useful test and intend to do it in subsequent work. However, this paper is focused on parameter estimation and we think that adding those simulations to this paper would unnecessarily complicate the existing text and analysis.

**Author Changes**: None.

30.   Finally, the whole work could be a valuable practice for the CGAM community. It uses "hindcast" to estimate the logit distribution parameters for this model. Hindcast Is not a new approach. The outcome of this practice may help the GCAM community to improve their work on land use modeling. However, the results of this practice may be not useable for other models. As they may follow very different modeling structure and assumptions. The author of this paper should make this point very carefully.

**Author Response**: We agree with the reviewer's notes here. Other models can use from our methodology and some of the takeaways from this paper (e.g., the choice of metric and which crops to include can alter the resulting parameters and model performance); however, other models are unlikely to be able to use these parameters directly.

**Author Changes**: We've added this caveat to the discussion.

31.   The abstract provides trivial information. It is not an abstract of this paper.

**Author Response**: We have revised the abstract.

**Author Changes**: We have revised the abstract.

32.   Following a summary of land use change at the global scale, the second paragraph of the

introduction begins with: "Similar trends occurred in the United States". This is not an accurate statement. The US land use change did not follow the global land use changes in terms of land conversion to crop production. No expansion in cropland has been observed for the cases of US.

**Author Response**: We have removed that sentence.

**Author Changes**: We have removed that sentence.

Sorry, why you removed this part. You have to correct your statement and say that the United States have not followed the common trends in other countries and inform the reader why not.

---

## Referee Report (RR2)

**Responses to authors**
**Revised mmanuscript entitled:**

**Modeling Land Use and Land Cover Change: Using a Hindcast to Estimate Economic Parameters in gcamland v2.0**

**Geoscientific Model Development**

*Anonymous*

In what follows I responds to the authors second responses using black fonts with yellow background.

**Anonymous Referee #2**

Summary of the paper This paper uses gcamland/GCAM to calibrate/estimate/tune the land distribution parameters of a nested logit land allocation function used in this model. In the lack of econometrically estimated values for these parameters, it is an important effort to accomplish this task. However, the paper suffers from some important deficiencies and lack of clarity, in particular for those who do not know this mode.

**Author Response**: Thank you for your detailed review of our paper. Your comments have pointed out a number of places where we did not clearly describe our methodology.

**Author Changes**: We have made a number of changes to the paper in response to your comments and those of another reviewer. Those changes are detailed below.

Response: Thanks for your attention. Your responses helped me to understand your work better. However, in some cases your responses were not convincing. In what follows I highlight my comments using green fonts after each response.

**Author Response:** Thank you for your detailed review of our paper.

**Author Changes:** We have made a number of changes to the paper in response to your comments. Those changes are detailed below in red. Note that in response to many of the comments we have added several tables and figures to the Supplementary Material. We note these additions in the individual responses, but are including a list here to facilitate the review. Note that we are using the new table and figure numbering and not the numbering used in the previous submission. All other tables and figures are still included in the paper, but with new numbers.

Figure S1: Nesting diagram showing all land types included in gcamland and how they are grouped for the logit
Table S1: List of all gcamland land types, including a list of crops/categories included, and the data used for initialization, simulation, and observations.
Table S2: List of the gcamland node names (from the nesting diagram), along with the plain language description used in the paper and the logit exponent used in the analysis
Table S4: Absolute error, percentage error, and NRMSE for all crops and expectation types
Figure S10: Change in land area for forest, grassland, and shrubland in both the gcamland simulations and the observations
Figure S11: Change in land area for all gcamland land types

Thanks for these changes and additions. These help the reader to understand your model and work much better in transparent way. I will respond to each item in what follows.

Some important comments:
1. I am not a GCAM modeler but is seems gcamland operates under GCAM. For nonGCAM community the links and interactions between these two models are not clear. How they linked and interact. A simple chart can help.

**Author Response**: GCAM and gcamland are completely separate models. GCAM includes representations of energy, water, land, and climate. It includes a land allocation mechanism where land use and land cover are calculated based on changes in profit. gcamland only includes this land allocation calculation. gcamland is not run when GCAM is run; GCAM is not run when gcamland is run. Instead, gcamland replicates the land allocation equations used in GCAM so that we can isolate that part of the code for analysis and uncertainty quantification. See also response to comment #19.

**Author Changes**: We have added a footnote to the methodology section clarifying the relationship between GCAM and gcamland: "GCAM and gcamland are separate models. While gcamland replicates the land allocation mechanism in GCAM, it is not run within GCAM. Similarly, GCAM is not run as a part of gcamland. gcamland only includes a representation of land allocation. GCAM includes representations of agricultural supply and demand, land allocation, and other sectors (energy, water, economy, climate). The land allocation mechanism within gcamland uses price, yield, cost, subsidy, logit exponents, expectation parameters, and initial land area as exogenous inputs and endogenously determines land area in subsequent years. Changes in demand are explicitly represented in GCAM. In gcamland, changes in demand are captured through changes in price. For example, the increase in demand for corn and soybean due to biofuels policy is captured through changes in the prices of these goods."

Thanks, it seems the CGAM model plays no role in the estimation process of the estimated parameters and only the gcamland with no demand side has been used. If that is the case, then the paper needs revisions to make sure that CGAM has not been used in the estimation process. For example, here are a few places that need attention:

- Table 2, Description of Arable Land should refer to "gcamland" not "GCAM", as noted in footnote 5.
- Change title of 2.1.2 to "Economic approach in gcamland"
- Change title of section 2.3 to "Evaluation gcamland model performance"
- Change the end of line 204 to "… focused on annual time step in gcamland"

**Author Response:** That is correct. GCAM is only relevant here because it is where we derived the land allocation equations and data, and because we could use the parameters estimated in this paper in GCAM. Note that in order to use the parameters derived in this paper in GCAM, we would also need to estimate parameters for 31 other geopolitical regions and estimates of parameters governing demand since GCAM includes both supply and demand. See also response to #29
**Author Changes:** We have made the changes suggested by the reviewer.

Thanks for adopting the proposed changes. Not that, It is now very clear that in your calibration process only gcamland is used. And thanks for declaring that you need to repeat the calibration process proposed in this paper for 31 other geographical regions of GCAM model to be able to

run it for a historical simaution (as noted in comment # 29 as well) and a validation test.

2. The model clearly uses a nesting logit format, perhaps three nests. Equation 1 of the paper shows only on nest. The formula should be replaced with a formula for the full nest.

**Author Response**: We have added information on how profit and shares are calculated for the other nests.

**Author Changes**: We have added additional text and an equation "In the three-level nest version, land allocation at each level is determined by a modified version of equation 1, where Y is replaced by the land allocated to that particular nest. The land allocated to a particular nest is dynamic and varies over time. Profit rates ($r_j$) at the lowest level of the nest are computed based on price, cost, yield, and subsidy (if included) for land use types (crops, pasture, commercial forest); profit rates for land cover types are input into the model and are based on the value of land. Profit rates for higher levels of the nest ($r_{node}$) are determined by: $r_{node} = [\sum_{j=1}^{n}(\lambda_j r_j)^\rho]^{1/\rho}$

The revision mentioned here is helpful, but it provides a broken formulation and misrepresent your work. To be fully transparent and to make sure that the readers understand that three exponent parameters (distribution parameters) and there sets of share parameters were estimated, I propose to use the following format:

$X_{jt}^{C} = \frac{(\lambda_{jo}^{C} r_{jt}^{C})^{\rho^C}}{\sum_j (\lambda_{jo}^{C} r_{jt}^{C})^{\rho^C}} \cdot Y_t^C$ where "C" represents the crop nest including rice, wheat, ……

$X_{jt}^{A} = \frac{(\lambda_{jo}^{A} r_{jt}^{A})^{\rho^A}}{\sum_j (\lambda_{jo}^{A} r_{jt}^{A})^{\rho^A}} \cdot Y_t^A$ where "A" represents the ag-forestry nest including cropland, forest, ….

$X_{jt}^{R} = \frac{(\lambda_{jo}^{R} r_{jt}^{R})^{\rho^R}}{\sum_i (\lambda_{jo}^{R} r_{jt}^{R})^{\rho^R}} \cdot Y_t^R$ where "R" represents the arable land nest including …..

The members of each nest should be clearly mentioned. In particular, for the crop nest all members of this nest including unused land, cropland pasture, CRP, other idled land should be specified. The information that presented in Table 2 is incomplete and very misleading. Of course, the names of variables should be described fully and maintain this convention through the paper.

**Author Response:** Please note that the equation did not transfer well from the PDF of the review and thus appears strangely in the comment and response above. We have used the original PDF with the correct formatting when reading and reacting to this comment. The same equation is used in all three places; given the confusion this seems to have caused, we have now repeated it as suggested by the reviewer. Given the number of crop categories in gcamland, listing all crops in Table 2 made it difficult to read the table, so we have opted to provide the full list in a footnote. We also include a complete nesting diagram with all information in the same figure in the SM. To facilitate readability, we have opted to use plain language descriptors in the paper rather than the abbreviated terminology used with the gcamland code. However, we do agree that we need to specify the mapping between plain language and terminology, so we have added a table to the SM.

**Author Changes:** We have repeated the logit equation as suggested by the reviewer. We have added information on what is contained in the nests, using the node names (now defined in Table S2) in the main text and the complete list of crops in footnote 6. We have added a nesting diagram to the SM (Figure S1) which indicates which members are in which nests. We have

added a table to the SM (Table S2) that maps the plain language descriptors used in the main text of the paper to the precise terminology in the gcamland model.

Sorry that the pdf file including my comments was not presenting the suggested formulas correctly. The pdf version that I posted on the Journal website shows the formulas correctly. Anyway, I see that you correctly specified and defined the proposed equations. Thanks for that and thanks for adding Figure S1 and Tables S2. Your work is now more transparent and traceable.

However, I see that table S2 in defining A, R, and C refers to equation 1 which is wrong. It should refer to equation 2.

Figure S1 represents 7 nests for the supply structure of gcamland. On the other hand, Table S2 shows that your calibration process estimates distribution parameters for only three of them. The rest (including 4 distribution parameters) were given some ad hoc values. For the root nest, it makes sense to use a zero value, as you assumed no change in the areas of members of this nest. However, it is not clear from where the values of 2.7, 0.05 and 1.575 are come from. Explain how those values were determined? The paper should explain the sources of these ad hoc values.

3.   How the land constraint/constraints is/are defined? Does a simple land constraint directly add all types of land: Total land =forest + pasture + corm + soy+ etc.? or each nest has its own land constraint?

**Author Response**: The only explicit constraint on land in the model is on total land. That is, we require the sum of all land types (forest, pasture, grassland, shrubland, urban, crops, etc.) to equal the total area in the United States. We parameterize the model to prevent expansion of cropland into non-arable lands (urban, tundra, and rock/ice/desert).

**Author Changes**: We have clarified this in section 2.1.2: "The land allocated to a particular nest is dynamic and varies over time."

Thanks for the clarifications that: 1) only one land constraint is used and 2) land transformation from non-arable lands to cropland is prohibited. However, these clarifications raised the following important concerns:

- First, when it goes to crops how do you distinguish between harvested area and planted area? There is no information for planted area for all crops. If harvested area is used for each crop, then what do you do with multiple cropping, crop failure. You cannot simply add up harvested area and say that the sum is total cropland. Please be very specific in your response.
- According to your response, non-arable land cannot be converted to cropland. You said some kinds of parameterizations were used. Please explain it more transparently and be very specific in your response. How about the revers land transformation: conversion of cropland to non-arable land? How do you model that? Please, again be precise in your response and avoid the general statement of based on profitability.

**Author Response:** gcamland tracks both harvested area and planted area. We have a fixed ratio of harvested to planted area for each crop, estimated in the base year and held constant throughout the simulation. We have focused our results on harvested area since that is the comparison data we used from FAO. Planted area is used in the constraint on total land.

To limit the transformation of non-arable land (urban, tundra, and rock/ice/desert) to cropland, we set the logit exponent above that nest to zero. With a logit exponent of zero, land shares are held constant at their base year values. Such a parameterization also means that cropland cannot be converted to non-arable land.

Note that I have continued to use "arable" and "non-arable" in this response. However, based on our response to your comment #6, we are no longer using these terms in the paper.

**Author Changes:** We have added a couple of sentences to 2.1.2 describing the gcamland approach to planted and harvested area: "gcamland tracks both planted and harvested area for crops. Planted area is determined by the logit-base land allocation scheme described in this section. Harvested area is calculated using planted area and a fixed harvested-to-planted area ratio, estimated in the base year, and held constant in the future."

In response to the second point, we have elaborated on the footnote, which now states: "This land is held constant throughout the simulation time period by setting the logit exponent dictating competition between these land types to zero. Such a parameterization means that no cropland can be converted to urban, rock/ice/desert or tundra and no urban, rock/ice/desert or tundra can be converted to cropland."

Thanks for these clarifications. The paper is now more transparent, and one can follow it better.

Based on this response and also according to Table S1, it seems gcamland uses FAO data for harvested area (for each crop) and uses a fixed ratio between harvested area and planted area (again for each crop) to trace planted area. It is also noted that the ratio between harvested area and planted area (again for each crop) is estimated in the bas year and remained constant over time. However, it is not clear how the ratio between harvested area and cropland is determined in the base year for each crop. For the case of US economy, one can find this ratio for some crops using USDA data but not for all crops. The paper should be transparent and reveal how the ratio is calculated for each crop category and what assumptions were used. Please declare these in a proper way

The clarifications made in this response reveals that the gcamland misses two important factors. The first missing factor is the reduction in crop failure due to technological progress over time. The second missing factor is multiple cropping. With the revealed set up, it is very clear that gcamland (and hence GCAM) ignores both of these factors. While missing the first factor may not be significant, the second one is a major failure. According to the USDA repots (see the USDA Economic Information Bulletin Number 178) up to 4% of the US total harvested area is related to double cropping. This is a large area to miss. For example, it could be up to about 12.5 million acres in 2012. This is a big missing item. This basically means that gcamland (and hence CGAM) badly over estimate demand for cropland. It is something that cannot be ignored. It seems the author of this paper plan to do the same calibration practice for other GCAM regions. Missing multiple cropping in other countries (e.g., India, China, Brazil, and many other countries) would be a disaster, as the rates of multiple cropping in these countries are really large. Failing to address this issue is a big mistake. I am not asking the authors of this paper to fix this huge problem in the current paper, but I expect the authors to address this issue and present their plan to fix it for future work.

4. It is not clear how the estimation process is defined to estimate these parameters. Does the process estimate all the distribution parameters (s) simultaneously or individually?

**Author Response**: We are using Latin Hypercube Sampling, which estimates all parameters simultaneously.

**Author Changes**: We have added the following note to section 2.2: "Latin Hypercube Sampling draws all parameters simultaneously from uniform distributions."

Thanks, for this clarification. Please let me concentrate on the three logit distribution parameters and use the notation mentioned above. Based on this response and the response to my next comment, it seems the implemented process uses the following steps:

- Randomly selects three values for the three distribution parameters of $\rho^R, \rho^A$ and $\rho^C$ from three uniform distributions (each ranged between 0.01 to 3).
- Given the selected parameters of $\rho^R, \rho^A$ and $\rho^C$, land allocation for the whole time period is determined
- The above two steps repeated for 10,000 collections for each expectation type.
- A collection, including three distribution parameters, that minimizes the size of NRMSE is selected as the best estimate for the distribution parameters under each examined expectation type.

**Author Response:** Please note that the equation did not transfer well from the PDF of the review and thus appears strangely in the comment and response above. We have used the original PDF with the correct formatting when reading and reacting to this comment. This is mostly correct. For the first step, we are drawing three distribution parameters plus parameters governing how expectations are formed for the non-Perfect expectation models. Additionally, given subsequent comments, we do think we should elaborate on the second and fourth steps. For the second step, "Land allocation for the whole time period is determined" *by running gcamland over the historical period with each set of randomly chosen parameters.* For the fourth step, NRMSE is calculated by comparing simulated land area from gcamland over time with observations of land area (that is, we calculate land area in the historical period with gcamland and compare those outputs to observations to determine NRMSE). We have separated this step into two to make this clearer in the paper.

**Author Changes:** We have added an elaborated version of these steps to the beginning of the methodology section and are using these steps to help the reader navigate the subsections of the methodology:

"The overall methodology used in this paper is as follows:
1. Randomly select a set of parameters from uniform distributions (see Table 2 for list of parameters and their values, Section 2.2 for how the parameters are selected, and Sections 2.1.2 and 2.1.3 on how those parameters are used in gcamland).
2. Land allocation for the whole time period is estimated by running gcamland over the historical period with each set of randomly chosen parameters (see 2.1 for a description of gcamland and how it simulates land allocation and section 2.4 for the data used in gcamland).
3. Repeat steps 1 and 2 for 10,000 parameter draws (see Section 2.2).
4. Calculate a variety of metrics of goodness of fit from simulated land allocation from gcamland and observations of land allocation (see Section 2.3 for the metrics used and Section 2.4.3 for a description of the observation data).
5. Determine the "best" set of parameters by choosing the set that minimizes a given goodness of fit metric."

Thanks for the clarifications and confirmation.

5.  How distribution parameters () were perturbed? Are they coming from given distribution? If yes, what type of distribution? Is this a random selection of three values limited between 0.01 and 3?

**Author Response**: The parameters were sampled assuming a uniform distribution. For logit exponents, we choose values for each of the three exponents randomly between 0.01 and 3. Each of the three exponents can have a different value and we do 10,000 samples for each expectation type and model configuration.

**Author Changes**: We have added the following note to section 2.2: "Latin Hypercube Sampling draws all parameters simultaneously from uniform distributions."

Thanks for this clarification. See my previous response.

6.  Over time total area of agricultural land in the US has declined sharply, due to conversion to non-agricultural uses of land (urbanization, infrastructure, . . ..). How gacamland handles conversion of land to non-forestry-ag land. How land availability land has been taken care off over time? Is it an exogenous variable in each year?

**Author Response**: While urban land has grown over time, the definition of urban land in gcamland accounts for only 1% of total land area in the United States. We hold this area constant in the simulations presented here (equal to 1975, 1990 or 2005 values depending on the calibration year used in the simulation). Allowing this to change over time would not have a noticeable impact on results given how small the area is. Similarly, we hold tundra and rock/ice/desert constant in time, but they account for very small amounts of land in the United States (2.6% and 0.4%, respectively). All other land is included in the economic competition.

**Author Changes**: We have added a footnote to the methodology section: "A small amount of land (~4%) is considered non-arable in gcamland in the United States, including urban, tundra, rock, ice, and desert. This land is held constant throughout the simulation time period."

Thanks for this clarification. However, it does not help and makes further confusion.

-   The paper does not provide a clear picture of the land categories in the model. While you are talking about non- arable land as a part of land use modeling, the logit nesting structure does not cover it. If this type of land is not modeled, why it is described in the paper.
-   Table 1 introduces three nests: Arable land; Ag land. Forest, and other; and Cropland (or crops). This plus other information provided in the tale bring in mind the following nesting structure:

[Figure]

However, there is no *"cropland"* in the middle nest. What is the relationship between Ag land, cropland, and pasture land? It seems a nest is missing.

- It is noted FAO data with some modification is used to determine land data. If that is the case, then something is not correct. Please let me explain. Arable land is a subset of cropland in the FAO data. For example, in the FAO data for 2018, area of US cropland is 160,437 thousand hectares with sub component of 157,737 thousand hectare of arable land. This is in a sharp contrast with the nesting structure defined in the paper. Also, area of US Ag land is 2.6 times of its arable land. How Ag land could be a subset of arable land?

The paper should make proper clarifications on this issue.

Finally, let us follow the nesting structure outlined in the paper. According to the FAO data, total area of US arable land has declined from 180,630 thousand hectares in 1961 to 157,727 thousand hectares in 2018, a decline by 22,893 thousand hectares. How this reduction has been handled? It is not a small reduction.

**Author Response:** In terms of the nesting, we had focused this part of methodology section on the aspects of gcamland that we perturbed in this experiment. It is clear from the reviewer's response that this has caused confusion.

For "arable", we were using this word in the colloquial sense ("suitable for growing crops") and not in the way that FAO defines it ("areas under temporary crops, temporary meadows and pastures, and land with temporary fallow"). We can see that this caused confusion and given the paper's dependence on FAO data we have decided to remove the word "arable" in almost all instances in the paper. The one exception is when we define the categories in gcamland; in this case, we think it is important to use and define the terminology in gcamland. gcamland only uses the word "arable" in the name of "OtherArableLand" category, so this is the only case in which "arable" will appear in the paper. Note that throughout the paper we have on occasion used plain language descriptions or lengthier titles for categories and names in gcamland to make the paper easier to read; within gcamland, we use shorthand and abbreviations. We have now added a table to the SM (Table S2) mapping the precise terminology used in gcamland to the plain language descriptors used in the paper.

Thanks for these additions and clarification. They help to understand your work.

Since several items identified and mentioned in table S2 are not standard data and you guesstimated them, they have to be revealed and presented in Table S2. I propose to add a column to table S2 and show the area of each type of land for at least one representative year.

7. A big issue in land use modeling is marginal cropland (idled land under CRP, cropland pasture, other types of idled). Area of idled cropland in the US have changed a lot. The definition of "cropland pasture" has also changed over time. How idled land is treated?

**Author Response**: Idled land is called "Other Arable Land" in gcamland. The amount of land in this category can change over time based on economic signals. Since idled cropland does not produce a product, its profit rate is exogenously specified like other land cover types (see response to comment #12). Note that this exogenous value is similar to a CRP payment, as it

represents a marginal benefit of keeping land fallow.

**Author Changes**: We have added a footnote to the methodology section: "Fallow cropland (called other arable land in gcamland) is also included in this nest." We have also added text to the supplemental material indicating that idled cropland is included in gcamland.

This is a misleading response. FAO data base does not have other arable land. Please at least, provide a table and put all types of included land in each nest at list for three years, including cropland and its components.

**Author Response:** We derive the OtherArableLand category from harvested area data from FAO and total cropland area data from the land use harmonization (LUH) product. LUH provides total land cover classified as cropland. We assume the difference between land planted in crops and land cover of crops is idled land. See also the response to #3. We are unclear what the reviewer meant by 'at list [least?] for three years' and thus have not responded directly to that.

**Author Changes:** We have added a nesting diagram (Figure S1) and two tables to the SM (Table S1 and S2). These figures and tables include all land types in gcamland, where they are nested, what we call them in the paper, and where we get the initialization and simulation data. We believe that this provides full transparency for readers in clear, easy-to-understand forms.

Thanks for the clarifications and sorry for the typo ("least" not "list"). I think it was clear that I am asking for a table including your land data and you refused to provide it. Your new table S2 reveal that many items in your land data (in particular for non-cropland) are just guesstimates. You have to reveal your data at least for a one representative year. Table S2 is a good place to do that.

8.   It is noted that harvested area from FAO is used. FAO is missing many feed crops since 2011, including million hectares of those crops. without proper steps to cover missing crops in FAO, the estimated parameters will be subject to major issues and biases. Figure three suggest that those feed crops is missed. That is a major issue.

**Author Response**: gcamland includes fodder and feed crops, using data from FAO prior to 2011. We have excluded it from the comparison and statistics because the data is not available after 2011 as you noted, but it is included in the modelled results.

**Author Changes**: We have revised the Figure 3 caption to clarify this: "Note that fodder crops are included in gcamland but are excluded from total cropland area in this figure due to data limitations." We have also added figures to the supplemental material showing all crops, including fodder.

This is a misleading response. If you fixed the data (for after 2011) by estimating the missing data items, then that should be reported and included in your figures. Transparency is the key.

**Author Response:** We did not fix the data. gcamland only needs land cover data in the base year (1975, 1990, or 2005 depending on the historical time period to be simulated); FAO has data on fodder crops for all three of those years. With that quantity for use as a base year, we then simulate fodder crop land cover throughout the simulation time horizon (1975-2015, 1990-2015, or 2005-2015 depending), but we do not use FAO data in our simulations after the base year and

we do not fix the FAO data at all. Therefore, while we have simulated fodder crop land cover for the entire simulation time horizon, including years beyond 2011, we cannot compare the later years of the simulation to observations because they are not provided by FAO after 2011. This is why fodder crops are excluded from the NRMSE calculations.

**Author Changes:** We have added a table to the SM (Table S1) that lists all gcamland types, along with the source of data for initialization and for comparison.

Thanks for the clarifications. You need to revise the paper to make this clarification.

In addition, it seems Figure 3 shows harvested area not cropland. Change the title and the legend to harvested area. As you correctly highlighted in your introduction, in the US, cropland area remained flat while harvested area has increased over time with some fluctuations.

9.   GCAM is using commodity price to model land allocation. It seems wholesale farm prices is used. That is a bad proxy for exporting crops such as cone and soybeans. For example, half of soybean is exported at much higher price farm price.

**Author Response**: The producer price is the relevant price signal to be used for planting decisions. The market price, and thus the producer and consumer prices, is a function of the demand sectors as well, which includes domestic demand and exports. However, the resulting equilibrium price paid to producers is the relevant price regardless of how the demand is determined. Therefore, we feel that producer price is the right input into gcamland for this analysis.

**Author Changes**: We have revised the manuscript to better document what is included in gcamland. See also the responses to comment #1 and comment #19.

Yes, as it is noted in this response "price paid to producers is the relevant price", but the FAO whole sale price does not reflect the price received by farmers. You need to address this as a major limitation of your work clearly. Please explain what revision you have made? And be more specific.

**Author Response:** We are not using wholesale prices. We are using producer prices from FAO, which FAO defines as "prices received by farmers" for the reasons above. We are not clear on what in the manuscript led the reviewer to think we were using wholesale price, but we are using producer prices (as indicated by the reference to the FAO producer price dataset).

**Author Changes:** We have added "producer" as a qualifier on "price" in section 2.4.2 to clarify this.

Sorry for the confusion between the producer price and whole sale price. Yes, I understand that you used the FAO "producer price" data. But that is not the point controversy. The point is that price received by farmers (including subsidies) is the relevant price and you used the producer price (farmgate price). Producer price is not equal to price received by farmers.

10.   In this paper, in one case, subsidy has been examined in a sensitivity test. Subsidy is the key item in deriving land use, land rent, and the price received by farmers. The distribution parameters of the logit should be evaluated with subsidies. Sensitivity test is meaningless. The key here is to capture all types of subsidies paid to famers in the estimation processes.

**Author Response**: As noted in the manuscript, the subsidy data does not improve the estimation either because these subsidies do not affect cropland area (as suggested by Weber and Key 2012 and discussed in Section 5.1) or due to the quality of the data. There are not continuous, complete, and consistent data sets for all types of subsidies paid to farmers. Additionally, we only found crop-specific information on direct payments, making the inclusion of other types of subsidies difficult.

**Author Changes**: We added more information to Section 2.4.2 describing the limitations of the data: "Subsidies are a reality of crop agriculture in the United States. However, there are not continuous, complete, and consistent data sets for all types of subsidies paid to farmers. Additionally, crop-specific information (of the type needed for gcamland) is only available for direct payments, making the inclusion of other types of subsidies difficult" and our choice to make this a sensitivity "Because this data is inconsistent and incomplete, we only use it as a sensitivity in this paper and do not include it in the primary analysis."

Thanks for this response. However, it is incomplete. The paper should clearly acknowledge that improper inclusion of subsidies could make major biases in the outcome of the estimation process and that a sensitivity test is not a proper way of fixing this issue.

**Author Response:** We agree with the reviewer that including subsidies (or any changes to the calculation of profit or the equations determining land allocation) could alter the errorminimizing parameters resulting from this analysis. However, the simulations that included subsidies did not result in a change in different estimates of error-minimizing parameter values but did increase NRMSE for those parameter sets relative to the simulations without subsidies.
Finally, sensitivity tests are a common way in modeling studies to test whether a particular factor matters. In this case, our sensitivity test concluded it did not. Furthermore, we have found previous literature that supports this finding (noted in section 5.1). Thus, while it is possible that including subsidies could change results or lead to biases, it is not guaranteed.

**Author Changes:** We have added information clarifying the effect of subsidies in this study and indicating that the inclusion of subsidy data *could* result in different parameters, though that did not happen here. In particular, we've expanded footnote 9 to say (added text in italics): "Note that our choice to use it as a sensitivity and not the default is because it does not improve NRMSE *and did not alter the parameter set that minimized NRMSE between simulated and observed land allocation* (as discussed in Section 5).".

We've revised the sentence describing the results of this sensitivity in Section 5.1 to clarify what does and doesn't change with subsidies: "Varying these assumptions results in differences in cropland area (Figure 4) and in parameters for the "Same Parameters" sensitivity; however, the parameters for the "With Subsidies" sensitivity are identical to the Default model (Table S2)."

To the discussion, we have added: "In theory, any change in the data or in the profit calculation, like the inclusion of subsidies, could alter the error and the set of parameters that minimize error (i.e., conversely, the exclusion of those factors could introduce biases in the estimated parameters). However, in our study, we found that the inclusion of subsidies increased NRMSE, but did not alter the parameters that minimized NRMSE."

Thanks for these clarifications and additions

11.  How biofuels were included in the simulations? Biofuels and biofuel policies were major drives of land use. How that included in your simulations

**Author Response**: Biofuels and biofuels policy are reflected in our model through changes in producer prices of crops.

**Author Changes**: We have added this to the footnote explaining differences between GCAM and gcamland: "Changes in demand are explicitly represented in GCAM. In gcamland, changes in demand are captured through changes in price. For example, the increase in demand for corn and soybean due to biofuels policy is captured through changes in the prices of these goods."

This response makes me more concern. I am not convinced that your work properly identifies source of piece changes. You approach only consider changes in the prices and send that as a signal to the land supply tree, without identifying the sources of price change. The source of price change could be demand shock (e.g. biofuels) or supply shock (reduction in yield or land supply). You approach does not distinguish these sources and simply consider price changes as signals to the land supply. Can you convince the readers that you do not need to identify the source of price change?

**Author Response:** Our model was not designed to identify the source of price changes as we focus on supply responses. However, we do represent supply and demand shocks differently. The direct effect of a supply shock (e.g., a yield change) is explicitly represented in gcamland; demand shocks (e.g., biofuel policy) are signaled by changes in prices. This information is enough for landowners to make land allocation decisions based on the relative rental profits, particularly under perfect foresight. In the cases with imperfect expectations (e.g., adaptive expectation), we allow a different expectation coefficient for biofuel crops (i.e., Corn and OilCrop) to reflect that the price expectations of these biofuel crops could be different than other crops. In other words, biofuel shocks in our model indirectly affect price signals and/or expectation schemes. As discussed in Section 4.1, the results indicate that Corn and OilCrop rely less on past information than other crops in the Adaptive expectations.

**Author Changes:** None.

Thanks for this explanation. You need to include this justification in the paper.

12.  Th dapper highlights that gcamland uses commodity prices in land allocation. But the model allocates land across land cover items. What prices are used for forest products, livestock products, etc.? The paper is silent on these prices. What prices were used for land cover items

**Author Response**: We calculate land rental prices for commercial forest and pasture using their product (forest or livestock products) prices and related productivity and cost information.  For non-commercial land cover only items, effective profit rates are derived during the calibration process to ensure that the amount of land area in the base year predicted by the logit equation matches the read in value. For subsequent years, these effective profit rates are held constant. This estimation is described in detail in Wise et al. (2014).

**Author Changes**: We have added this information to the methodology section: "Profit rates ($r_j$) at the lowest level of the nest are computed based on price, cost, yield, and subsidy (if included) for land use types (crops, pasture, commercial forest); profit rates for land cover types are input into the model and are based on the value of land."

- This response makes no clarification. For example, it is said that: "We calculate land rental prices for commercial forest and pasture using their product (forest or livestock products) prices and related productivity and cost information." Please be very specific and answer the following: What are the prices of livestock products? How about prices of forest products? How do you measure yield for forest? How about yield for pasture land?

- If only arable land is modeled, why do you need profit rates for non-commercial land. As noted in my earlier comments, providing a table including all nests and their members could help.

- What is the effective rate for unmanaged forest? How do you determine it in the calibration prices? Calibration to what?

- The paper should clearly explain the above points

**Author Response:** We estimate the yield for forest from its vegetation carbon density and the yield for pasture is set to the yield of hay. Forest prices are calculated from FAO's export data. Pasture price data is not available; instead, we set these prices to 70% of the alfalfa price (used for FodderHerb).

For arable land, as noted in our response to #6, the choice of this word has caused confusion. We include all land use and land cover categories in gcamland and thus need profit rates for each. The effective profit rate is the profit rate that would be needed to return the land allocation in the base year. The calibration process ensures that land allocation in the base year exactly matches the input values. To do this, we determine what the profit rate would need to be for the logit to predict the input (referred to as the "effective profit rate" above). For non-commercial land, we hold these effective profit rates constant in the future. As noted above, this procedure is described in Wise et al. (2014). We cannot repeat everything from that paper; thus, we refer the reader to that paper for a discussion of the calibration procedure, as is common in science.

**Author Changes:** We have added a nesting diagram (Figure S1) and a table (Table S1) that specifies each land type in gcamland to the SM. This table includes the type and sources of all input and comparison data used.

Thanks for the clarifications. Based on Table S1 and its footnote # 5, no data is available to calculate profitability of non-commercial land types. Therefore, you used an unknown approached to calculate profits for these types of land in the base year land allocation. The calibrated profits of these land types (including other arable land) for the base year should be revealed. Table S2 is a good place to show them along with the calculated profits for the commercial land types.

Let us assume that the calibrated profits for these land types are credible. How do you determine changes in these profits after the base year? Please be very specific in your response.

13. Regarding forestry, how gcamland treats forest land. Is it operates based on managed forest? Managed + unmanaged? How it treats unmanaged forest with no economic output.

**Author Response**: We include both managed and unmanaged forestland. For managed forestland, we use price and yield to calculate profit. For unmanaged, see response to comment #12.

**Author Changes**: See response to comment #12.

What are the price and yield of managed forest? How do you determine them? Based on what data do you calculate price of forest and yield of managed forest? The paper should explain these.

See also my response to comment 12 as well.

**Author Response:** The yield of managed forest is derived from its carbon density. The price is derived from its export value and volume, as reported by FAO.

**Author Changes:** We have added a table to the SM (Table S1) with information on data sources for all gcamland land types.

Thanks for the clarifications. See my response to comment # 13.

14. GCAM and gcamland are not forestry models. Forestry is not an annual crop. How these models take care of forestry in a dynamic setting. Do these models treat forestry as an annual crop?

**Author Response**: GCAM, gcamland, and other similar models assume that you need to set aside land at every timestep to ensure that you will have enough commercial forestland to meet harvest demand at the time the forest matures. To do this in GCAM, we assume that the amount of land needed for forest is equal to the wood product demand divided by the yield times the rotation length. So, if you need 1 Ha of land to meet wood product demand in a given year and the rotation length is 25 years, we set aside 25 Ha of land in that year. gcamland uses a similar paradigm, only we don't model demand, just yield and area. So, gcamland uses a yield that is equal to the harvest yield divided by the rotation length.

**Author Changes**: We have added this information to the methodology section: "Note that, since forestland is not an annually planted and harvested commodity, GCAM, gcamland, and other similar models assume that land must be set aside at every timestep to ensure enough commercial forestland is available to meet harvest demand at the time the forest matures. To do this in gcamland, we assume that the amount of land allocated to forest depends on the harvest yield and the rotation length."

Sorry for my ignorance. But I do not know any model that follow the assumption mentioned above. In your revision, please name some well-knows models that follow this approach. If I understand correctly, what mentioned above can be replicated by the following formula:

Forest area in year t = yield of forest in t * demand for forest in t * 25.

This formula over simplifies the way that a forest model works and has no root in real world. You need to put this formula in the paper and justify it to be transparent.

**Author Response:** First, the discussion above was about "commercial forestland" (also referred to as managed forest in these models) as indicated in that response and in the text added to the paper. Second, as noted in Grassi et al. (2021, https://doi.org/10.1038/s41558-021-01033-6), models like GCAM (e.g., IMAGE, MESSAGE, AIM, REMIND) calculate managed forest area based on: "(1) forest product demand (mostly based on FAO statistics and then projections into the future), (2) carbon density of forests and/or timber that can be harvested per hectare increments and (3) estimates on length of rotation cycles and/or year to maturity." These

calculations assume "the area of managed forest represents the area required to provided historic and future demand for wood products in continuous harvest rotations". This approach was (and still is) described in Section 2.1.2 of the paper.

**Author Changes:** We have added a footnote to the SM table (Table S1) describing data sources indicating how managed forest area is calculated in gcamland.

Thanks for these responses.

15.  In each case, the model is solved for a range of parameters. Then a set of parameters that minimizes NRMSE is selected. But NRMSE is defined for a single crop. How this variable is aggregated over crops? How NRMSE is calculated for non-cropland (e.g. forest, pasture, grass land, and etc.)? How cropland and non-cropland aggregated?

**Author Response**: We take the average of crop-specific NRMSE to get the aggregated NRMSE. As noted in the manuscript, we do not include non-cropland in our calculation of NRMSE.

**Author Changes**: We have clarified this in the methodology by adding "For NRMSE and RMSE" before the sentence that describes the averaging across crops.

Thanks for these clarifications. I have difficulties to find out in what part of the manuscript it is noted that you do not include non-cropland in the calculation of NRMSE. Please clearly mention this point at the beginning of section 2.3 so that the reader can see this important limitation. Also, it is important to explain why you do not include non-cropland in the calculation. This is a surprise for me? Why you do not follow the same approach for other nest that you determine their distribution parameters. Indeed, as I mentioned before, you determine a selection of three parameters of $\rho^R, \rho^A and \rho^C$. But it seems you ignore how good are your estimated for $\rho^R$ and $\rho^A$ and only care about $\rho^C$. In fact, you select a mix of $\rho^R, \rho^A and \rho^C$ to minimize NRMSE over the crop nest. So, this means that you may get bad errors for the other two nests while you minimize only over crops. This is a serous problem. Indeed, you do not optimize for two nests out of three nests. This could cause major errors in the projection of land use changes in the non-cropland nests, compared to the observed data.

**Author Response:** In a nested logit model, all logit exponents above a particular leaf play a role in determining the land area of that leaf. That is, the amount of land allocated to a particular crop depends on the cropland logit, as well as the two logits in the nests above cropland. For this reason, we have to vary all three logits to do a sensitivity on the area for an individual crop.

As for why we did not include non-cropland, we had included land types where we have complete time series of observation data, covering the entire simulation period, in our default NRMSE calculations. We do not have land cover data for the entire time series since the satellite data record does not go that far back in time. We have done a sensitivity where we include land cover to demonstrate the effect of it on NRMSE and the parameters that minimize NRMSE. However, we are keeping this as a sensitivity since we do not have a complete time series of observation data.

**Author Changes:** We have added a nesting diagram (Figure S1) which shows which nests and logits affect the area allocated to crops. We have also revised the discussion of the nested logit in the methodology section adding additional equations and description: "In a nested logit, the area of a particular land type is determined by not just the logit of its nest, but also by the logit of the

nests above that".

We have added a discussion of a sensitivity simulation which includes land cover types to Section 5.2.2: "Finally, including all dynamic land cover types where observations are available for any period of the simulation years (e.g., non-fodder crops, grassland, shrubland, and forest) in the calculation of NRMSE increases the NRMSE substantially (from 1.4 to 75) due to definitional differences in land cover types. The change in land area for land cover types is reasonably consistent with observations (Figure S10); however, the absolute value for grassland and shrubland differs substantially (Figure S9). Despite the increase in NRMSE, the inclusion of land cover types does not alter the parameter sets that minimize NRMSE."

We have also added a sentence to the methodology section clarifying our choice to focus NRMSE calculations on crops: "By default, we include any land type where we have observations for the full time series of the simulation, which effectively means all crops excluding fodder crops (see Section 2.4.3 and Table S1); however, we include a sensitivity on the set of land types included in Section 5.2.2.". Finally, we've added text on the availability of observation data in both Section 2.4.3 and Table S1.

Thanks for these clarifications

16. How productivity of non-cropland is measured?

**Author Response**: We include productivity of pasture and forest, but not of other non-cropland types.

**Author Changes**: We have added this information to the methodology section: "Profit rates ($r_j$) at the lowest level of the nest are computed based on price, cost, yield, and subsidy (if included) for land use types (crops, pasture, commercial forest); profit rates for land cover types are input into the model and are based on the value of land."

Thanks for this clarification, but you did not answer the question completely. Please explain how did you calculated productivity of forest and pasture land? It is an important piece of information and the reader should know it.

**Author Response:** We calculate productivity of forest from its vegetation carbon density and the yield for pasture is set to the yield of hay.

**Author Changes:** We have added information on the source of yields for all land types to a table in the SM.

Thanks for the clarification.

17. GCAM aggregates crops into some specific categories? How prices were generated for those categories. In many cases there is no data on crop prices?

**Author Response**: We use the weighted average of producer prices for aggregated commodities, weighting by the production.

**Author Changes**: We have added this information to Section 2.4.2: "Data was aggregated from individual crops to the GCAM/gcamland commodity groups, weighting crops by their production

quantity"

Thanks for the clarification, but for some crops, such as fodder crops, no data is available for quantity of production. Please explain what you did for those crops.

**Author Response:** The reviewer is correct that our previous answer was incomplete. For most crops, we use the weighted average of producer prices as the aggregate commodity price, weighting by production. For fodder crops, we do not have complete time series of price or production data. Instead, we use alfalfa prices from USDA for FodderHerb prices and we set the FodderGrass prices at 70% of the FodderHerb prices.

**Author Changes:** We have added a table to the SM that includes the data sources for prices. We explain how prices are weighted for most crops and include the source for the fodder prices.

Thanks

18.  The paper provides mixed messages on endogenous and exogenous variable. In determining targeted distribution parameters, what variables were targeted and what variables were determined in the model. It seems prices, areas, and yields were exogenous. Be more specific.

**Author Response**: Within gcamland, prices, costs, yields, subsidies, logit exponents, and expectation parameters are exogenous. In addition, the land area in the calibration year is exogenous. Areas in subsequent years are endogenous. Within the experiment in this paper, we also varied logit exponents and expectation parameters as part of the ensemble sampling.

**Author Changes**: We have added this information to a footnote in the methodology section: "The land allocation mechanism within gcamland uses price, yield, cost, subsidy, logit exponents, expectation parameters, and initial land area as exogenous inputs and endogenously determines land area in subsequent years."

Thanks for this clarification.

19.  The whole practice implicitly assumes that other model parameters are accurate and valid. This is a strong assumption. The land supply parameters were determined while demand parameters held constant. The estimated supply parameters will be entirely wrong if the demand parameters (e.g. income and price elasticities for crops, livestock products, and forestry) are not valid. Any change in the demand parameters could alter your estimated parameters for the land supply. Can you test sensitivity of your results with respect to changes in other elasticities of the model?

**Author Response**: As noted in our response to comment #1, gcamland does not include a representation of demand for the exact reason you note here. We have chosen to isolate the land allocation mechanism in gcamland to ensure we get the right parameters for the right reasons and do not have cancelling errors. We cannot do a sensitivity on demand elasticities since they are not included in the model at all. Price is the only link to demand and we are using observed prices from FAO to ensure that demand-side sensitivity and errors do not affect the parameter estimation on the supply-side.

**Author Changes**: See response to #18.

20.   The results are counterintuitive. Let me explain using figure 2. In the adaptive case, for the first two nests the values are about 0.4 and for the last nest (cropland) the value of is about 0.6. Given that limited land movements among land cover items occurred at national level in the US and lots of change occurred in the crop nest, one could justify this outcome. However, for the other three cases (hybrid, linear, and perfect) the ranking is of values shows revers. Meaning that land conversion is easier at the land cover nests than the cropland land nest. These outcomes do not make sense. Am I missing something?

**Author Response**: Those outcomes also get lower NRMSE than the adaptive expectations, indicating that they do not explain historical land allocation as well as adaptive expectations. The parameter sets for hybrid, linear, and perfect minimize NRMSE *if those expectation types are assumed*, but the model with the lowest NRMSE includes adaptive expectations and parameters that match our intuition.

**Author Changes**: An explanation of the results and intuition is provided in section 4.1.

Thanks for this clarification.

21.   In showing the results, level variables were used to show errors. For example, figure 2 compares estimated harvested areas with their observations for four types of expectations. This hides the errors involved. It is better to calculate errors as percent difference between the estimated and observed areas.

**Author Response**: Thank you for the suggestion. We have considered adding a figure on percentage difference to the paper (attached). However, we feel that showing absolute values compared to observations is more informative since it gives a sense of scale, which differs significantly across crops in the USA.

**Author Changes**: We have added figures comparing model results to observations for all commodities to the supplemental material, but have opted not to add figures showing percentage difference.

This is not an appropriate response. It is an essential task to inform the readers regarding the percent errors between projections and observed values. Comparing the level variables in a chart hides those errors. to be transparent and informative you need to provide data on errors. If it is hard to show it in charts, please show them in a table in the appendix. This is an essential validation check.

**Author Response:** See response to comment #24
**Author Changes:** See changes in response to comment #24

See my response to comment # 24

22.   The main manuscript only presents comparison of the projected and observed harvested areas and provided no comparison for other land types.

**Author Response**: The comparison of projected and observed area for other land types is

included in the supplemental material for types where observations are available.

**Author Changes**: The comparison of projected and observed area for other land types is included in the supplemental material for types where observations are available. We have also added a paragraph to the supplemental material stating which land types are in gcamland and explaining our choice of what to show where: "The main text of this paper focuses on four commodity groups (Corn, Wheat, OtherGrain, and OilCrop), as these four commodities represent the largest land area in the United States. However, gcamland includes twelve commodity groups in total, representing all crops reported by the FAO, and fallow or idled cropland (referred to as other arable land in gcamland). In addition, gcamland includes commercial forest and pasture, as well as several other land cover types, including forest, grassland, shrubland, tundra, rock/ice/desert, and urbanland. We include results for other agricultural commodities and the land cover types where observations are available in this section"

Sorry, I do not consider this as a satisfactory response. The paper and its supporting material provide inconstant and confusing information about the land cover types, land uses, and components of each of the three nests included in the model. Simply revise table 2 and clearly put all land types and land uses in that table for each nest.

For example, in the main text in table 2 pasture land is not a part of middle nest. But it appears in the SI in Figure 7 as a component in the mix of grassland, shrubland, and pasture. Very confusing.

The main text should clearly represent the nesting structure and the component of each nest. Do not refer to another paper. This is an essential information for this paper.

Figure S7 should show data for each land cover item including managed forest, unmanaged forest, pasture, grassland, shrubland, any other components of the non-cropland nest, and cropland as one land cover type. If your model does not trace changes in some land types, that type of land should not be included in the model nor in the paper.

As a subcategory of cropland, the projections for unused cropland and their corresponding observations should be presented and compares.

**Author Response:** We have added a nesting diagram and a table to the SM with information on every land type included in the paper. We have also included a figure in the SM with model results and observations (where available) for all land types. We have included this an additional figure rather than an expansion of the original Figure S7 (now Figure S9) because in some cases the observations we have are for a sum of two gcamland land types. The new figure shows land by type for each individual type included in gcamland.

**Author Changes:** We have added a nesting diagram (Figure S1) and a table to the SM (Table S1) with information on every land type included in the paper. We have also included a figure in the SM (Figure S11) with model results and observations (where available) for all gcamland land types. We have also clarified when observations are available and when they are not, including the information in Table S1 and a description of what is included in the various figures and why: "Finally, Figures S6-S10 and Table S4 focus on comparing gcamland simulation results to observations for categories or sums of categories where observation data is present. However, there are other land types included in gcamland (see Figure S1 and Table S1). Figure S11 shows the evolution of all individual land categories in gcamland for the default simulations, with

observation data plotted when it is available for the individual category."

Thanks for these Changes. Some of these changes are good. Indeed, Figure S11 mentioned in this response, provides no additional information. The scales used for these panels hide the variation and difference between alternative cases over time. In most cases, except for grassland and shrubland, the areas presented by these panels are smaller than 50 km$^2$, while the max value of the vertical axis in each panel sets to show 3000 km$^2$. To be transparent and help the readers to evaluate your work, it is critical to show the numerical results presented in the panel of figure S11 and their corresponding actual observations in a simple plain table. The table should include the simulations results for all land types listed in Table S2 and their corresponding observations. Providing an Excel file attachment is a better option.

23. Results are highly aggregated into four groups of crops. how about the 12 categories of crops in GCAM?

**Author Response**: The main manuscript shows four of the 12 crop categories. We have included figures showing the other categories to the supplemental material.

**Author Changes**: The main manuscript shows four of the 12 crop categories. We have included figures showing the other categories to the supplemental material. We have also added an explicit reference to these figures in the main text.

Please, add changes in unused land. That is an important piece of information.

**Author Response:** I am not clear on what the reviewer means by "unused land", but we have added information on all land types included in gcamland.

**Author Changes:** We have added information and results for all land types included in gcamland, including a nesting diagram (Figure S1), a table with data sources (Table S1), and a figure showing changes in land allocation for all gcamland land types (Figure S11).

Unused land is a common terminology and refers to the cropland which has not been used for crop production (CRP, cropland pasture, set aside land, fallow land, ….). In the nesting structure provided in the new revision you named it "other arable land".

24. The figure S5 of SI shows major errors for the change in forest area. This show that the model fails to represent changes in forest area correctly.

**Author Response**: Figure S5 had included the net change from 1990 to 2015 for the modeled data and the net change from 1992 to 2015 for observations for forest. Figure S7 shows the whole time series. As shown in S7, the time series tracks the observations fairly closely for the adaptive expectations. We do not think it is correct to say that the model fails to represent changes in forest area; however, we do think that Figure S5 was confusing given the unit used and the differences in time horizon.

**Author Changes**: We have removed Figure S5 as it did not add any new information and was confusing (see response to comment #25).

First thanks for adding figure S7. Adding this figure is a step forward. As I mentioned in comment # 23 you need to extend this figure for all land cover items. In particular, it is

important to show errors in %, not in levels. Level variables hide errors. Please show the percent errors, then we can judge the model performance in land cover items. Also remember that you failed to calculate the goodness of fit (in your language NRSME) for land cover items.

Figure S7 shows bad performances for land cover items that already are included in this figure. In particular, I see very large differences in level variables between the performance and observed items. You also could alter the scale of this figure to better see the errors for forest. You should show the errors in percent to show the model performance. I believe it is straight forward to calculate NRSME for these items. Why not?

**Author Response:** We have added a calculation of NRMSE including dynamic land cover types. As noted in our response to comment #15, we are keeping this as a sensitivity since we do not have a complete time series of observations for land cover types.

As for percentage errors, percentage errors can mask as much as absolute errors (like an unnormalized rms) for this sort of multi-target work. Percentage errors often overemphasize errors in land types with small land allocations. For example, the largest error in percentage terms in any of the gcamland simulations is for PalmFruit. In absolute terms, this error is virtually zero. NRMSE uses the standard deviation of observation to normalize errors rather than using percentage errors to address this point. For this reason, many model validation studies rely on NRMSE as a measure of error. We have now added a table showing absolute error, percentage error, and NRMSE; we have also included a discussion on our choice to focus on NRMSE instead of percentage errors.

**Author Changes:** We have added a calculation of NRMSE including dynamic land cover types where observation data is available (see also response to comment #15). We have added a table to the SM (Table S4) showing absolute error, percentage error, and NRMSE, as well as a discussion on the different metrics: "Finally, Table S4 summarizes the error (simulation minus observation) in both absolute (million km2) and percentage terms, as well as including NRMSE for each expectation type and crop. We include all three metrics in this table; however, in this study, we primarily use NRMSE. Normalized measures of error are key for interpreting whether a simulated data set acceptably replicates available observational data. While normalizing to present errors in terms of percentages is common, this can result in large magnitude percentage errors when dealing with multiple variables (land types) with a wide range of magnitudes. Given the significant difference in land historically allocated to different uses in the United States (e.g, the PalmFruit vs Corn commodities in gcamland) and the fact that we are seeking parameter sets to minimize error measures across these commodities, this can lead to misleading results. Rather, we follow the literature normalizing by the standard deviations of observations (Nash and Suttcliff 1970; Willmott 1981; Legates and McCabe 1999; Willmott et al 2012; Tebaldi et al 2020), captured in our NRMSE. This allows a benchmark of whether the discrepancies between simulated and observed data fall within the natural variability of the observed data, giving a statistically justifiable benchmark to determine whether those discrepancies are acceptably sized."

Thanks for these clarifications. Please provide the table mentioned in comment # 22 to resolve all the issues.

25. The figure S5 show increases in all land cover types and harvested areas. How that could

be possible?

**Author Response**: Figure S5 shows the ratio of land in 2015 to land in 1990. Values less than 1 indicate a reduction in land area over time. There are several land types where area declined and those land types have values less than 1. However, we do think the unit used in this figure was confusing and all of the information contained in this figure is also shown in Figure S7, so we have removed this figure.

**Author Changes**: See response #24.

Thanks, see my responses to item # 24

26.  Figure S7 shows no results for land cover items including grassland and shrubland for three types of expectations. Why?

**Author Response**: These results are included but they are covered by the line for adaptive expectations. All four expectation types produce similar values.

**Author Changes**: We have improved the figure so that the other expectation types are visible and added a note to the caption indicating they are overlapping.

Thanks

27.  Figure S7 shows major errors for grassland and shrubland in the adaptive approach project huge errors. Why?

**Author Response**: This is due to differences in the definition of grass and shrubland between gcamland and the CCI land cover product. Most notably, gcamland includes a large portion of what is categorized as grass or shrub in CCI land cover as pasture.

**Author Changes**: This was already stated in the paragraph immediately preceding figure S7: "Due to differences in definitions of land cover between gcamland and the CCI land cover product, Grassland and Shrubland do not match in absolute value between gcamland and the observation data." We have also added a panel to this figure showing the sum of pasture, shrubland and grassland; this summation improves upon the data mismatch.

Sorry, but this response does not make sense. If you used different definitions in your modeling practice, you should adjust the actual observations based on your definition as well. please revise accordingly. Even the combined panel shows large differences. We are talking about millions of $km^2$.

**Author Response:** We do not think it is appropriate to adjust the observation data. However, we agree that it makes it difficult to compare results. We cannot initialize gcamland to the observation data because the CCI data is not available for all possible gcamland base years (1975, 1990, 2005). The initialization data we do use (now reported in Table S1) is a reconstruction based on potential vegetation and land use data; since it is not a single data product, we did not think it was a good source of observation data.

**Author Changes:** As noted, we do not think it is appropriate to adjust observation data, but we have added a figure (Figure S10) showing changes in land area for land cover types to compare

the model to observations. In addition, we've added sources for initialization and observation data to the new table in the SM (Table S1), along with an explanation of why we do not use the CCI data for initialization: "The CCI satellite data is only available starting in 1992. For this reason, we cannot use it as initialization data since we need that data for 1975, 1990, and 2005."

Thanks for these clarifications and additions.

28.  It seems the whole practice has failed to take care of land cover changes.

**Author Response**: Land cover changes are included in gcamland and are compared to observations in the Supplemental Material. Therefore, we do not think it is correct to say that we have failed to take care of land cover changes.

**Author Changes**: See changes in response to comments 12, 13, 16, 22, 24, 25, 26, 27.

I already responded to these. Your work fails to calculate the goodness of fit for land cover items. That should be highlighted in the main manuscript, as I highlighted in several places in my responses. Also, errors should be presented in % differences. This is an essential item to validate this work. given the size of land items, even 5% error is huge. We are talking about million hectares of land.

**Author Response:** We have addressed these comments in our responses to 21, 24, and 27.

Thanks for the responses and clarifications. But they do not show that your work provides satisfying outcomes for land cover. Just look at your figure S10.

29.  The examined practice estimated a few parameters of the model for land use. A good way to test the outcomes of this practice is to run the GCAM model with the estimated parameters and compare the model results for land use changes, land cover changes, changes in crop prices, and changes in yield with actual observations over the examined period.

**Author Response**: We agree that this is a useful test and intend to do it in subsequent work. However, this paper is focused on parameter estimation and we think that adding those simulations to this paper would unnecessarily complicate the existing text and analysis.

**Author Changes**: None.

I am glad that you consider this test as a useful test. But it is not only useful. This is an essential task. Validation of estimated parameters is a crucial task. Indeed, without this test you do not know how good the estimated parameters for projection are. Running a validation test may require more work, but it is an essential task.

I observe that the revised version noted that: "we have focused on the historical period. However, these models and parameter estimates could be used in a simulation of future land use and land cover change to better understand their implications." Even you have not shown how your model project historical data. You only used historical data to estimate the logit parameters. But failed to test how good are the estimated parameters to replicate the historical. That should be clearly acknowledged in the paper.

If you choose not running this test, then the paper should clearly acknowledge that you have not

examined the validity of your practice in the abstract, discussion section, and conclusion.

**Author Response:** We believe the reviewers response to our previous answer stems from the confusion between gcamland and GCAM, as discussed in comment #1. We have used the parameters in gcamland, re-run the model, and compared the results to observations. That is, we have shown how gcamland simulates land area in the historical period using the new parameters (see Figures 3-6). We have not run GCAM, which would require us to replicate this analysis for 31 other regions (GCAM includes 32 regions). In addition, GCAM includes models of agricultural demand and links to the energy system, all of which will affect land allocation. We do think running the updated parameters through GCAM is useful future work, but it is outside the scope of this paper.

**Author Changes:** We have revised the final paragraph clarifying what we have done and why we are leaving the GCAM analysis to future work. The first sentence now reads (new text in italics): "In this paper, we have focused on the historical period, *simulating land allocation in gcamland over this period and comparing it to observations*." And, we've added this sentence about the potential GCAM validation test: "However, for such a study, we would need parameter estimates for all thirty-two GCAM regions and not just the United States." We have also added an overview of the methodology, including the simulation of historical land using gcamland and comparison with observations to the beginning of Section 2 (see response to comment #4).

Thanks for the clarifications and additions. However, to be clear, this paper basically is nothing more than an incomplete calibration practice based on some ad hoc assumptions under several limitations. The outcome of this calibration process is note validated. This paper provided no evidence regarding the outcome of GCAM for replications of historical observations using the calibrated parameters.

30. Finally, the whole work could be a valuable practice for the CGAM community. It uses "hindcast" to estimate the logit distribution parameters for this model. Hindcast Is not a new approach. The outcome of this practice may help the GCAM community to improve their work on land use modeling. However, the results of this practice may be not useable for other models. As they may follow very different modeling structure and assumptions. The author of this paper should make this point very carefully.

**Author Response**: We agree with the reviewer's notes here. Other models can use from our methodology and some of the takeaways from this paper (e.g., the choice of metric and which crops to include can alter the resulting parameters and model performance); however, other models are unlikely to be able to use these parameters directly.

**Author Changes**: We've added this caveat to the discussion.

Thanks

31. The abstract provides trivial information. It is not an abstract of this paper.

**Author Response**: We have revised the abstract.

**Author Changes**: We have revised the abstract.

With all due respects, the revised abstract needs a major work. The fist six lines provide a lecture to justify this work. Those should be included in the main text not in the abstract. Then it is noted that: "We run gcamland simulations with these parameter sets over the historical period in the United States to quantify land use and land cover, determine how well the model reproduces observations". This is not what you did. you have not quantified "land use and land cover" over time. You used historical data to estimate some model parameters. You have not determined how your model produces actual observations for the estimated parameters. This is the validation test that you refused to accomplish. You have not highlighted your findings on the sizes of the estimated parameters. You have not highlighted the limitations of this work. You need to revise the abstract.

**Author Response:** First, we are following the guidance provided by the Nature journals on how to write an abstract or summary paragraph (https://www.nature.com/documents/nature-summaryparagraph.pdf). Such guidance recommends providing an introduction to the field (both general and specific) to motivate the work before describing the specific contribution of the paper. It does not include limitations of the work in the abstract, instead that is included in the discussion.

Second, we have quantified land use and land cover in the historical period (see also response to comment #29). The statement in the abstract is correct as written. For each randomly sampled set of parameters, we initialize gcamland to a base year of data (e.g. 1990) and then use the parameter sample set to simulate land use and land cover in subsequent years (e.g. 1991-2015) without incorporating additional historical data. We then compare the gcamland simulated

Thanks for saying that the abstract is prepared based on the guidance provided by the Nature journal. But the guidance does not ask you to write about general materials that are the core of your paper. The abstract consists of about 280 words. One third of it describes materials that are not really the core of your paper. Your aper is not about future changes in land use, it is not about water use, it is not about uncertain socioeconomic condition, and it is not about technological progress that you referred to. These are important issues, but not directly presented in your work. after all, this paper basically uses a calibration process under uncertainty in crop prices. Indeed, this paper calibrates a few parameters that govern land supply. You need to tune your abstract to say the importance of this calibration practice.

Again, your work does not simulate historical observation. It uses historical observation to calibrate land allocation parameters. Running GCAM model with the calibrated parameters to replicate historical observation is a valuable practice, but you clearly admitted that you are not able to perform this valuable practice with incomplete calibration process. You should not oversell your work or claim for something that has not achieved. The abstract should say: 1) the paper proposes a calibration method for the GCAM land allocation module; 2) the proposed method is tested for the case of US; 3) what are the calibrated and selected parameters; 4) what limitations are involved; 5) what changes are needed to run the calibration for other regions.

32. Following a summary of land use change at the global scale, the second paragraph of the introduction begins with: "Similar trends occurred in the United States". This is not an accurate statement. The US land use change did not follow the global land use changes in terms of land conversion to crop production. No expansion in cropland has been observed for the cases of US.

**Author Response**: We have removed that sentence.

**Author Changes**: We have removed that sentence.

Sorry, why you removed this part. You have to correct your statement and say that the United States have not followed the common trends in other countries and inform the reader why not.

**Author Response:** We removed that sentence because the entire next paragraph elaborates on the trends in the United States: "In the United States, crop production has increased substantially in the last several decades, but much of that increase in production is due to increases in yields (Babcock, 2015; Fuglie, 2010). Total cropland area in the United States has remained relatively constant between 1975 and 2015. Instead, there has been a shift in crop distribution, with an increasing share of corn and soybeans and a decreasing share of wheat and other grains (Figure 1, (FAO, 2020a; Taheripour and Tyner, 2013))."
**Author Changes:** No changes.

Thanks for brining my attention to these sentences.

---

## Referee Report (RR3)

**Reviewer number 2**

**Author Response:** Thank you for the helpful comments. We have revised the paper in response to your comments. We are adding our responses in purple font in the document below. We have also removed any prior comment that is now resolved to simplify the document.

**Author Changes:** We have revised the paper in response to your comments. We include a point- by-point description of the changes below.

Thanks for detailed responses. The revised version is much improved. The response to one of the comments needs additional attention. In what follows I highlighted the comment which needs additional work. I also added a few minor new comments on the abstract. My responses are in red fonts in blue background

1.     The model clearly uses a nesting logit format, perhaps three nests. Equation 1 of the paper shows only on nest. The formula should be replaced with a formula for the full nest.

**Author Response**: We have added information on how profit and shares are calculated for the other nests.

**Author Changes**: We have added additional text and an equation "In the three-level nest version, land allocation at each level is determined by a modified version of equation 1, where Y is replaced by the land allocated to that particular nest. The land allocated to a particular nest is dynamic and varies over time. Profit rates ($r_j$) at the lowest level of the nest are computed based on price, cost, yield, and subsidy (if included) for land use types (crops, pasture, commercial forest); profit rates for land cover types are input into the model and are based on the value of land. Profit rates for higher levels of the nest ($r_{node}$) are determined by: $r_{node} = [\sum_{j=1}^{n}(\lambda_j r_j)^\rho]^{1/\rho}$

The revision mentioned here is helpful, but it provides a broken formulation and misrepresent your work. To be fully transparent and to make sure that the readers understand that three exponent parameters (distribution parameters) and there sets of share parameters were estimated, I propose to use the following format:

$X_{jt}^C = \frac{(\lambda_{jo}^C r_{jt}^C)^{\rho^C}}{\sum_j (\lambda_{jo}^C r_{jt}^C)^{\rho^C}} \cdot Y_t^C$ where "C" represents the crop nest including rice, wheat, ……

$X_{jt}^A = \frac{(\lambda_{jo}^A r_{jt}^A)^{\rho^A}}{\sum_j (\lambda_{jo}^A r_{jt}^A)^{\rho^A}} \cdot Y_t^A$ where "A" represents the ag-forestry nest including cropland, forest, ….

$X_{jt}^R = \frac{(\lambda_{jo}^R r_{jt}^R)^{\rho^R}}{\sum_i (\lambda_{jo}^R r_{jt}^R)^{\rho^R}} \cdot Y_t^R$ where "R" represents the arable land nest including …..

The members of each nest should be clearly mentioned. In particular, for the crop nest all members of this nest including unused land, cropland pasture, CRP, other idled land should be specified. The information that presented in Table 2 is incomplete and very misleading. Of course, the names of variables should be described fully and maintain this convention through the paper.

**Author Response:** Please note that the equation did not transfer well from the PDF of the review and thus appears strangely in the comment and response above. We have used the original PDF with the correct formatting when reading and reacting to this comment. The same equation is used in all three places; given the confusion this seems to have caused, we have now repeated it

as suggested by the reviewer. Given the number of crop categories in gcamland, listing all crops in Table 2 made it difficult to read the table, so we have opted to provide the full list in a footnote. We also include a complete nesting diagram with all information in the same figure in the SM. To facilitate readability, we have opted to use plain language descriptors in the paper rather than the abbreviated terminology used with the gcamland code. However, we do agree that we need to specify the mapping between plain language and terminology, so we have added a table to the SM.

**Author Changes:** We have repeated the logit equation as suggested by the reviewer. We have added information on what is contained in the nests, using the node names (now defined in Table S2) in the main text and the complete list of crops in footnote 6. We have added a nesting diagram to the SM (Figure S1) which indicates which members are in which nests. We have added a table to the SM (Table S2) that maps the plain language descriptors used in the main text of the paper to the precise terminology in the gcamland model.

Sorry that the pdf file including my comments was not presenting the suggested formulas correctly. The pdf version that I posted on the Journal website shows the formulas correctly. Anyway, I see that you correctly specified and defined the proposed equations. Thanks for that and thanks for adding Figure S1 and Tables S2. Your work is now more transparent and traceable.

However, I see that table S2 in defining A, R, and C refers to equation 1 which is wrong. It should refer to equation 2.

Figure S1 represents 7 nests for the supply structure of gcamland. On the other hand, Table S2 shows that your calibration process estimates distribution parameters for only three of them. The rest (including 4 distribution parameters) were given some ad hoc values. For the root nest, it makes sense to use a zero value, as you assumed no change in the areas of members of this nest. However, it is not clear from where the values of 2.7, 0.05 and 1.575 are come from. Explain how those values were determined? The paper should explain the sources of these ad hoc values.

**Author Response:** We have corrected the reference to equations in table S2. For the other logit exponents, we have used the default values used in GCAM. We have chosen not to vary these in this exercise as they do not directly impact the amount of cropland area, which was the output we focused on in the main text. We have done an additional sensitivity analysis quantifying the impact of these logits on cropland area. In this analysis, we doubled each logit one at a time. The area allocated to each crop changes by less than 1% (the largest change in magnitude is -0.12%). These logits do have a larger impact on other land types. For example, doubling the ForestLand logit results in a shift in the distribution of commercial and non-commercial forest, with commercial forest increasing by as much as 27%. However, total forest is largely unchanged (maximum change of 0.22%). We have added this information to the supplementary material, near Table S2.

For the default values, we realize in re-reading the previous GCAM papers that while we have documented the approach to selecting logit exponents in Wise et al. (2014) and the specific numbers used are available on GitHub (github.com/jgcri/gcam-core) we have not documented specifically how those numbers were chosen. We have now added this to the supplementary material of this paper.

**Author Changes:** We have corrected the reference to equations in table S2. We have clarified which logit exponents were varied (and which were not) in the methodology section. We have

also added a discussion of the other logit exponents, including why we did not vary them, how the default values were chosen, and the effect of changing those values on the outputs used in this study to the supplemental material:

"Table S2 provides information the gcamland nodes, the total land area for each node in 1990, and logit exponents used in this study. As noted in the main text, three of the logit exponents used in gcamland are varied as part of the analysis in this paper. For the remaining logit exponents (root, Pastureland, Grass/shrubs, Forestland), we use the default values used in GCAM. These values were chosen based on heuristics, where larger values are used for land types that are more substitutable. For the root, this is set to zero, as we do not allow conversion into or out of urban, tundra, or rock/ice/desert. For grass/shrubs, the decision to shift between grassland and shrubland is unlikely to be an economic choice; for this reason, we set the logit exponent to a very low value, effectively preserving the shares of grassland vs shrubland in the initial model year. Both the Forestland and Pastureland logit exponents govern substitution between commercial and non-commercial land types; a shift between these land types is not a land conversion (i.e., it does not require re-planting) but a shift in use (i.e., either moving livestock or engaging in logging activities). For this reason, higher logit exponents are chosen. A higher logit exponent governs Pastureland than Forestland as the shift in use of pastureland is likely to be easier than the change in use of Forestland. We have chosen not to vary these in this exercise as they do not directly impact the amount of cropland area, which was the output we focused on in the main text. We have done an additional sensitivity analysis quantifying the impact of these logits on cropland area. In this analysis, we doubled each logit one at a time. The area allocated to each crop changes by less than 1% (the largest change in magnitude is -0.12%). These logits do have a larger impact on other land types. For example, doubling the ForestLand logit results in a shift in the distribution of commercial and non-commercial forest, with commercial forest increasing by as much as 27%. However, total forest is largely unchanged (maximum change of 0.22%)."

Thanks for the additional sensitivity test. I appreciate your efforts. I also appreciate your note which says you have "not documented specifically how those numbers were chosen". Your response also says: "We have now added this to the supplementary material of this paper". I see that you tried to explain and justify the role of these parameters in explaining Table S2. But you ignored to clearly mention that those values are ad hoc values. I checked Wise et al. (2014). This reference says nothing regarding the selected values under discussion. You have to clearly add the following phrase, or something similar, to the main manuscript not the supporting document:

"The logit values assigned to pastureland, grass/shrubs, and forest land nests have not been obtained from an explicit statistical approach nor a calibration process. They have been selected based on the authors' value judgment".

Additional new comments on abstract:

1) The abstract says: "In this study, we demonstrate a more systematic and empirically-based approach to estimating model parameters for an economic model of land use and land cover change, gcamland".

I propose the following modification as you only calibrate (not estimate) a few parameters assuming other model parameters are valid:

"In this study, we demonstrate a more systematic and empirically based approach to calibrating a few selected parameters of an economic model of land use and land cover change, gcamland.

2) The abstract says: "we generate a large set of model parameter perturbations and run gcamland simulations with these parameter sets over the historical period in the United States to quantify land use and land cover, determine how well the model reproduces observations, and identify parameter combinations that best replicate observations".

I propose the following modifications as you calibrate a small set of parameters assuming other parameters are valid. It is important to say what parameters you calibrated. Your paper is all about it, but the abstract says nothing about that.

"we generate a large set of model parameter perturbations on the selected parameters and run gcamland simulations with these parameter sets over the historical period in the United States to quantify land use and land cover, determine how well the model reproduces observations, and identify parameter combinations that best replicate observations, assuming other model parameters are valid. In particular, 3 parameters out of 7 parameters that govern land allocation in gcanland were calibrated only for the case US.

---

## Editor Decision (ED1)

Editor's comments:

I also recommend streamlining the presentation of the study and providing more detail. E.g., the introduction of the Methods section lists 5 bullet points while the study is in fact more comprehensive. You may consider providing a study design schematic that covers all these aspects and renders them better traceable throughout the manuscript. Also missing in this list and subsequent descriptions is how the expectation types relate to the parameter perturbations. Are they part of the LHC sampling or is each of the 10000 parameter sets run for each expectation type?

Another point of confusion may be the calibration of a gcamland parameter for one year and the subsequent sensitivity analysis/calibration using further parameter perturbations. The manuscript may profit here from clear descriptions of the whole study process as pointed out above.

As pointed out by the reviewer, more detail and clarity on further steps or options for validating such a selection of a "numerically optimal parameter set" that serves "to guide the selection of parameters for an economic model", i.e. a calibration, would be important, including the state-of-the-art in the field (e.g. relating to the Introduction) in case an actual out-of-bag validation is hampered by data availability or other limitations. This is touched upon in parts of the Discussion but should deserve more elaboration.

Finally, I also agree that the abstract should provide more technical information to better target the readership of GMD. E.g., it would be helpful to mention the expectation types tested, the perturbed parameter types, etc. L6-9 in turn are not necessarily relevant for a technical paper.

Specific comments:

- L122: GGE needs to be spelled out
- L136: planted to harvested area ratio provides the ratio of crop failures or non-harvested crops. Do you mean physical cropland to harvested area, which determines multi-cropping, or actual crop failure ratios?
- Footnotes 4/8: Provide citations for R packages
- Table 2: "Logit exponent (p) ..." should be included in column two for consistency
- L184: What are the implications of the acceptable NRMSE for the study results?
- L259: European Space Agency
- L386/L431/L503: reconsider use of "you/we"

---

## Author Response (AR3)

Editor's comments:

**Author Response:** Thank you for the helpful comments. We have revised the paper in response to your comments and the comments from reviewer #2. We are adding our responses in purple font in the document below. We have also removed any prior comment that is now resolved to simplify the document.

**Author Changes:** We have revised the paper in response to your comments and the comments from reviewer #2. We include a point-by-point description of the changes below.

I also recommend streamlining the presentation of the study and providing more detail. E.g., the introduction of the Methods section lists 5 bullet points while the study is in fact more comprehensive. You may consider providing a study design schematic that covers all these aspects and renders them better traceable throughout the manuscript. Also missing in this list and subsequent descriptions is how the expectation types relate to the parameter perturbations. Are they part of the LHC sampling or is each of the 10000 parameter sets run for each expectation type?

**Author Response:** The parameters used to form the expectation types are included in the LHS sampling. The parameters sampled include 3 logit exponents and 6 parameters related to expectations (3 for adaptive and 3 for linear). We have reorganized the methodology section to address this comment. In particular, we have elaborated on and added to the bullets at the beginning to ensure they are comprehensive, reorganized the remainder of the methodology section to align with these bullets, added a schematic, and added details to address the specific comments noted (e.g., how LHS sampling relates to expectation types).

**Author Changes:** We have reorganized the methodology section to address this comment. In particular, we have elaborated on the bullets at the beginning to ensure they are comprehensive, reorganized the remainder of the methodology section to align with these bullets, added a schematic, and added details to address the specific comments noted (e.g., how LHS sampling relates to expectation types). The updated bullets are:

"The steps implemented are is as follows (see also Figure 2):
1.  Sample Parameters: Using Latin Hypercube Sampling, randomly select a set of parameters from uniform distributions (see Section 2.2.1).
2.  Run gcamland ensemble: Land allocation in the United States for the whole time period is estimated by running gcamland over the historical period (i.e., as a hindcast simulation) with each set of randomly chosen parameters (see Section 2.1 for a description of gcamland and Section 2.2.2 for a description of the ensemble).
3.  Compare to Observations: Calculate a variety of metrics of goodness of fit from simulated land allocation from gcamland and observations of land allocation (see Section 2.2.3).
4.  Select best parameters per expectation type: Determine the "best" set of parameters by choosing the set that optimizes a given goodness of fit metric for each expectation type (see Section 2.2.4).

5. Select overall best model: Select expectation type and parameter set combination that optimizes a given goodness of fit metric across all expectation types (see Section 2.2.5).
6. Repeat Steps 1 through 6 for different model specifications (see Section 2.2.6)."

Another point of confusion may be the calibration of a gcamland parameter for one year and the subsequent sensitivity analysis/calibration using further parameter perturbations. The manuscript may profit here from clear descriptions of the whole study process as pointed out above.

**Author Response:** We have restructured the methodology section in part to clarify the difference between calibration, perturbed parameter ensembles, and sensitivity analysis and improved the description of the whole study. We have also changed the language around the calibration year to reduce confusion; we now refer to it as the "initial model year" when referring to the internal gcamland process using one year of data.

**Author Changes:** We have restructured the methodology section.The calibration of gcamland parameters are now discussed Section 2.1 (describing gcamland), Section 2.2.1 (describing all gcamland parameters and where their values come from for this study), and Section 2.2.2 describing the hindcast process and its use of data. We have also changed the phrase "calibration year" to "initial model year".

As pointed out by the reviewer, more detail and clarity on further steps or options for validating such a selection of a "numerically optimal parameter set" that serves "to guide the selection of parameters for an economic model", i.e. a calibration, would be important, including the state-of-the-art in the field (e.g. relating to the Introduction) in case an actual out-of-bag validation is hampered by data availability or other limitations. This is touched upon in parts of the Discussion but should deserve more elaboration.

**Author Response:** This paper demonstrates how to explore structural sensitivities (expectation types, subsidies accounted for or not) and how to estimate parameters that cannot be calculated directly from existing data (logit values and parameters for expectation types) for an economic model of land use and land cover change, gcamland, in a region of interest. This is an improvement over existing studies which rely on heuristics. With each set of perturbed parameters, gcamland simulates the land allocation in the historical period under different structural sensitivities, and the resulting simulated land allocations are validated against existing observational data that was not used as part of the simulation. Exploring structural sensitivities and estimating parameter values for each region in gcamland is an independent exercise: given historical price and yield data for that region, the land allocation model can have decision parameters estimated independently in each region. The exploration of different expectation structures under perturbed parameters is a key part of this study, and an addition to the economic land use modeling literature.

We focus on the United States in this paper to establish this methodology and, importantly, to identify the sensitivity to different factors, including to data sources that may not be available in other regions. In particular, this study explored many sensitivities (such as subsidies, initial model years, metrics to compare with observations, etc) in a data rich region to determine which factors were important. Taking subsidies as an example, had this study in the United States led to

the conclusion that they are so fundamentally important that they cannot be neglected even when data is not available, we would face a fundamentally different challenge in looking at other regions. Therefore, this USA-centric study establishes a methodology and provides a streamlined roadmap for exploring structural economic sensitivities (expectations) and estimating appropriate parameters for each of those structures in additional regions of gcamland in the future.

**Author Changes:** We have clarified in the abstract, introduction and discusion that this study is focusing on gcamland as an independent economic model to illustrate a methodology for exploring structural economic sensitivities and estimating appropriate parameters for those assumptions in the different regions modeled in gcamland. By focusing on a data-rich region (the United States), we can examine many sensitivities that would be more challenging in regions with less data available.

We have also clarified the discussion to note that, because gcamland implements the same land allocation equations and structure as GCAM, expectation structures and parameter values estimated for gcamland can have utility in GCAM experiments, when they have been estimated for all 32 regions shared by gcamland and GCAM. Specifically, it would not be computationally feasible to perform this  exploration of economic expectations and estimating associated parameters directly in GCAM. In the future, the methodology established in this paper will be repeated for all 32 regions, and the resulting optimal parameters will be run through GCAM in a hindcast to see if the data-informed economic expectations and parameters result in an overall better evaluation of model performance than the default decision parameters and expectation structure (as in Calvin et al, 2017; Snyder et al 2017).

Finally, I also agree that the abstract should provide more technical information to better target the readership of GMD. E.g., it would be helpful to mention the expectation types tested, the perturbed parameter types, etc. L6-9 in turn are not necessarily relevant for a technical paper.

**Author Response:** We have revised the abstract in response to this comment and that of the reviewer.

**Author Changes:** We have revised the abstract in response to this comment and that of the reviewer: "Future changes in land use and cover have important implications for agriculture, energy, water use, and climate. Estimates of future land use and land cover differ significantly across economic models as a result of differences in drivers, model structure, and model parameters; however, these models often rely on heuristics to determine model parameters. In this study, we demonstrate a more systematic and empirically-based approach to estimating model parameters for an economic model of land use and land cover change, gcamland. Specifically, we generate a large set of model parameter perturbations and run gcamland simulations with these parameter sets over the historical period in the United States to quantify land use and land cover, determine how well the model reproduces observations, and identify parameter combinations that best replicate observations. We also test alternate methods for forming expectations about uncertain crop yields and prices, including adaptive, perfect, linear, and hybrid approaches. We find that an adaptive expectation approach minimizes the error between simulated outputs and observations, with parameters that suggest that for most crops,

landowners put a significant weight on previous information. Interestingly, for corn, where ethanol policies have led to a rapid growth in demand, the resulting parameters show that a larger weight is placed on more recent information. We examine the change in model parameters as the metric of model error changes, finding that the measure of model fitness affects the choice of parameter sets. Finally, we discuss how the methodology and results used in this study could be used for other regions or economic models to improve projections of future land use and land cover change."

Specific comments:
- L122: GGE needs to be spelled out

**Author Response:** We have spelled out this acronym.

**Author Changes:** We have spelled out this acronym.

- L136: planted to harvested area ratio provides the ratio of crop failures or non-harvested crops. Do you mean physical cropland to harvested area, which determines multi-cropping, or actual crop failure ratios?

**Author Response:** We mean physical cropland.

**Author Changes:** We have modified the text to say "gcamland tracks both physical area and harvested area for crops"

- Footnotes 4/8: Provide citations for R packages

**Author Response:** We have added these citations.

**Author Changes:** We have added citations for R packages.

- Table 2: "Logit exponent (p) ..." should be included in column two for consistency

**Author Response:** We have added $\rho$ to column three to be consistent with other parameters.

**Author Changes:** We have added $\rho$ to column three to be consistent with other parameters.

- L184: What are the implications of the acceptable NRMSE for the study results?

**Author Response:** We have added a discussion of this (see "author changes" below).

**Author Changes:** We have updated the text to read "While $NRMSE = 0$ corresponds to perfect model performance, any $NRMSE < 1$, is considered acceptable model performance (e.g. Tebaldi et al 2020 and the review of metrics in Legates and McCabe, 1999). Using the standard deviation of observation as an error baseline puts the deviations between simulation and observation for each crop in the context of that crop's historical variations. If errors are greater than the historic

standard deviation, then by definition, simply using the 1990-2015 mean value of land allocation in every simulated year 1990-2015 would have resulted in better errors than the model under consideration. Note, however, that in a hindcast approach, one would not actually have access to the 1990-2015 mean observed land allocation to use as a model to simulation 1990-2015 land allocation; it is simply an easy conceptual counterfactual model. Even when comparing two different model results that each have $NRMSE < 1$, the model with the smaller $NRMSE$ value is considered better. "

- L259: European Space Agency

**Author Response:** We have corrected this.

**Author Changes:** We have corrected this.

- L386/L431/L503: reconsider use of "you/we"

**Author Response:** We have removed the instance of "you" and changed this to "we".

**Author Changes:** We have removed the instance of "you" and changed this to "we".

**Reviewer number 2**

In what follows I responds to the authors second responses using black fonts with yellow background.

**Author Response:** Thank you for the helpful comments. We have revised the paper in response to your comments. We are adding our responses in purple font in the document below. We have also removed any prior comment that is now resolved to simplify the document.

**Author Changes:** We have revised the paper in response to your comments. We include a point-by-point description of the changes below.

Some important comments:
1.      Resolved.

2.      The model clearly uses a nesting logit format, perhaps three nests. Equation 1 of the paper shows only on nest. The formula should be replaced with a formula for the full nest.

**Author Response**: We have added information on how profit and shares are calculated for the other nests.

**Author Changes**: We have added additional text and an equation "In the three-level nest version, land allocation at each level is determined by a modified version of equation 1, where Y is replaced by the land allocated to that particular nest. The land allocated to a particular nest is dynamic and varies over time. Profit rates ($r_j$) at the lowest level of the nest are computed based on price, cost, yield, and subsidy (if included) for land use types (crops, pasture, commercial forest); profit rates for land cover types are input into the model and are based on the value of land. Profit rates for higher levels of the nest ($r_{node}$) are determined by: $rr_{nnnnnnnn} = [\sum_{j=1}^{n}(\lambda_j r_j)^\rho]^1/_{\rho\rho}$

The revision mentioned here is helpful, but it provides a broken formulation and misrepresent your work. To be fully transparent and to make sure that the readers understand that three exponent parameters (distribution parameters) and there sets of share parameters were estimated, I propose to use the following format:

$XX_{jjjjCC} = \sum \underline{\qquad} (_{jj}^{\lambda\lambda}(\lambda\lambda^{cc}{}_{jjjj}{}_{ccjjjj}rr_{jjjj}rr^{cc}{}_{jjjjcc})^{\rho\rho})_{\rho\rho}{}^{CC}{}_{CC} \cdot YY_{jj}{}^{CC}$ where "C" represents the crop nest including rice, wheat, ……

$XX_{jjjjAA} = \sum \underline{\qquad} (_{jj}^{\lambda\lambda}(\lambda\lambda^{AA}{}_{jjjj}{}_{jjjjAA}rr_{jjjj}rr^{AA}{}_{jjjjAA})^{\rho\rho})_{\rho\rho}{}^{AA}{}_{AA} \cdot YY_{jj}{}^{AA}$ where "A" represents the ag-forestry nest including cropland, forest, ….

$XX_{jjjj}{}^{RR} = \sum \underline{\qquad} (_{ii}^{\lambda\lambda}(\lambda\lambda^{RR}{}_{jjjj}{}_{RRjjjj}rr_{jjjj}rr^{RR}{}_{jjjjRR})^{\rho\rho})_{\rho\rho}{}^{RR}{}_{RR} \cdot YY_{jj}{}^{RR}$ where "R" represents the arable land nest including …..

The members of each nest should be clearly mentioned. In particular, for the crop nest all members of this nest including unused land, cropland pasture, CRP, other idled land should be specified. The information that presented in Table 2 is incomplete and very misleading. Of course, the names of variables should be described fully and maintain this convention through the paper.

**Author Response:** Please note that the equation did not transfer well from the PDF of the review and thus appears strangely in the comment and response above. We have used the original PDF with the correct formatting when reading and reacting to this comment. The same equation is used in all three places; given the confusion this seems to have caused, we have now repeated it as suggested by the reviewer. Given the number of crop categories in gcamland, listing all crops in Table 2 made it difficult to read the table, so we have opted to provide the full list in a footnote. We also include a complete nesting diagram with all information in the

same figure in the SM. To facilitate readability, we have opted to use plain language descriptors in the paper rather than the abbreviated terminology used with the gcamland code. However, we do agree that we need to specify the mapping between plain language and terminology, so we have added a table to the SM.

**Author Changes:** We have repeated the logit equation as suggested by the reviewer. We have added information on what is contained in the nests, using the node names (now defined in Table S2) in the main text and the complete list of crops in footnote 6. We have added a nesting diagram to the SM (Figure S1) which indicates which members are in which nests. We have added a table to the SM (Table S2) that maps the plain language descriptors used in the main text of the paper to the precise terminology in the gcamland model.

Sorry that the pdf file including my comments was not presenting the suggested formulas correctly. The pdf version that I posted on the Journal website shows the formulas correctly. Anyway, I see that you correctly specified and defined the proposed equations. Thanks for that and thanks for adding Figure S1 and Tables S2. Your work is now more transparent and traceable.

However, I see that table S2 in defining A, R, and C refers to equation 1 which is wrong. It should refer to equation 2.

Figure S1 represents 7 nests for the supply structure of gcamland. On the other hand, Table S2 shows that your calibration process estimates distribution parameters for only three of them. The rest (including 4 distribution parameters) were given some ad hoc values. For the root nest, it makes sense to use a zero value, as you assumed no change in the areas of members of this nest. However, it is not clear from where the values of 2.7, 0.05 and 1.575 are come from. Explain how those values were determined? The paper should explain the sources of these ad hoc values.

**Author Response:** We have corrected the reference to equations in table S2. For the other logit exponents, we have used the default values used in GCAM. We have chosen not to vary these in this exercise as they do not directly impact the amount of cropland area, which was the output we focused on in the main text. We have done an additional sensitivity analysis quantifying the impact of these logits on cropland area. In this analysis, we doubled each logit one at a time. The area allocated to each crop changes by less than 1% (the largest change in magnitude is -0.12%). These logits do have a larger impact on other land types. For example, doubling the ForestLand logit results in a shift in the distribution of commercial and non-commercial forest, with commercial forest increasing by as much as 27%. However, total forest is largely unchanged (maximum change of 0.22%). We have added this information to the supplementary material, near Table S2.

For the default values, we realize in re-reading the previous GCAM papers that while we have documented the approach to selecting logit exponents in Wise et al. (2014) and the specific numbers used are available on GitHub (github.com/jgcri/gcam-core) we have not documented specifically how those numbers were chosen. We have now added this to the supplementary material of this paper.

**Author Changes:** We have corrected the reference to equations in table S2. We have clarified which logit exponents were varied (and which were not) in the methodology section. We have also added a discussion of the other logit exponents, including why we did not vary them, how the default values were chosen, and the effect of changing those values on the outputs used in this study to the supplemental material:

"Table S2 provides information the gcamland nodes, the total land area for each node in 1990, and logit exponents used in this study. As noted in the main text, three of the logit exponents used in gcamland are varied as part of the analysis in this paper. For the remaining logit exponents (root, Pastureland, Grass/shrubs, Forestland), we use the default values used in GCAM. These values were chosen based on heuristics, where larger values are used for land types that are more substitutable. For the root, this is set to zero, as we do not allow conversion into or out of urban, tundra, or rock/ice/desert. For grass/shrubs, the decision to shift between grassland and shrubland is unlikely to be an economic choice; for this reason, we set the logit exponent to a very low value, effectively preserving the shares of grassland vs shrubland in the initial model year. Both the Forestland and Pastureland logit exponents govern substitution between commercial and non-commercial land types; a shift between these land types is not a land conversion (i.e., it does not require re-planting) but a shift in use (i.e., either moving livestock or engaging in logging activities). For this reason, higher logit exponents are chosen. A higher logit exponent governs Pastureland than Forestland as the shift in use of pastureland is likely to be easier than the change in use of Forestland. We have chosen not to vary these in this exercise as they do not directly impact the amount of cropland area, which was the output we focused on in the main text. We have done an additional sensitivity analysis quantifying the impact of these logits on cropland area. In this analysis, we doubled each logit one at a time. The area allocated to each crop changes by less than 1% (the largest change in magnitude is -0.12%). These logits do have a larger impact on other land types. For example, doubling the ForestLand logit results in a shift in the distribution of commercial and non-commercial forest, with commercial forest increasing by as much as 27%. However, total forest is largely unchanged (maximum change of 0.22%)."

3. How the land constraint/constraints is/are defined? Does a simple land constraint directly add all types of land: Total land = forest + pasture + corm + soy+ etc.? or each nest has its own land constraint?

**Author Response**: The only explicit constraint on land in the model is on total land. That is, we require the sum of all land types (forest, pasture, grassland, shrubland, urban, crops, etc.) to equal the total area in the United States. We parameterize the model to prevent expansion of cropland into non-arable lands (urban, tundra, and rock/ice/desert).

**Author Changes**: We have clarified this in section 2.1.2: "The land allocated to a particular nest is dynamic and varies over time."

Thanks for the clarifications that: 1) only one land constraint is used and 2) land transformation from non-arable lands to cropland is prohibited. However, these clarifications raised the following important concerns:

- First, when it goes to crops how do you distinguish between harvested area and planted area? There is no information for planted area for all crops. If harvested area is used for each crop, then what do you do with multiple cropping, crop failure. You cannot simply add up harvested area and say that the sum is total cropland. Please be very specific in your response.
- According to your response, non-arable land cannot be converted to cropland. You said some kinds of parameterizations were used. Please explain it more transparently and be very specific in your response. How about the revers land transformation: conversion of cropland to non-arable land? How do you model that? Please, again be precise in your response and avoid the general statement of based on profitability.

**Author Response:** gcamland tracks both harvested area and planted area. We have a fixed ratio of harvested to planted area for each crop, estimated in the base year and held constant throughout the simulation. We have focused our results on harvested area since that is the comparison data we used from FAO. Planted area is used in the constraint on total land.

To limit the transformation of non-arable land (urban, tundra, and rock/ice/desert) to cropland, we set the logit exponent above that nest to zero. With a logit exponent of zero, land shares are held constant at their base year values. Such a parameterization also means that cropland cannot be converted to non-arable land.

Note that I have continued to use "arable" and "non-arable" in this response. However, based on our response to your comment #6, we are no longer using these terms in the paper.

**Author Changes:** We have added a couple of sentences to 2.1.2 describing the gcamland approach to planted and harvested area: "gcamland tracks both planted and harvested area for crops. Planted area is determined by the logit-base land allocation scheme described in this section. Harvested area is calculated using planted area and a fixed harvested-to-planted area ratio, estimated in the base year, and held constant in the future."

In response to the second point, we have elaborated on the footnote, which now states: "This land is held constant throughout the simulation time period by setting the logit exponent dictating competition between these land types to zero. Such a parameterization means that no cropland can be converted to urban, rock/ice/desert or tundra and no urban, rock/ice/desert or tundra can be converted to cropland."

Thanks for these clarifications. The paper is now more transparent, and one can follow it better.

Based on this response and also according to Table S1, it seems gcamland uses FAO data for harvested area (for each crop) and uses a fixed ratio between harvested area and planted area (again for each crop) to trace planted area. It is also noted that the ratio between harvested area and planted area (again for each crop) is estimated in the bas year and remained constant over time. However, it is not clear how the ratio between harvested area and cropland is determined in the base year for each crop. For the case of US economy, one can find this ratio for some crops using USDA data but not for all crops. The paper should be transparent and reveal how the ratio

is calculated for each crop category and what assumptions were used. Please declare these in a proper way

The clarifications made in this response reveals that the gcamland misses two important factors.
The first missing factor is the reduction in crop failure due to technological progress over time. The second missing factor is multiple cropping. With the revealed set up, it is very clear that gcamland (and hence GCAM) ignores both of these factors. While missing the first factor may not be significant, the second one is a major failure. According to the USDA repots (see the USDA Economic Information Bulletin Number 178) up to 4% of the US total harvested area is related to double cropping. This is a large area to miss. For example, it could be up to about 12.5 million acres in 2012. This is a big missing item. This basically means that gcamland (and hence CGAM) badly over estimate demand for cropland. It is something that cannot be ignored. It seems the author of this paper plan to do the same calibration practice for other GCAM regions. Missing multiple cropping in other countries (e.g., India, China, Brazil, and many other countries) would be a disaster, as the rates of multiple cropping in these countries are really large. Failing to address this issue is a big mistake. I am not asking the authors of this paper to fix this huge problem in the current paper, but I expect the authors to address this issue and present their plan to fix it for future work.

**Author Response:** First, based on comments from the editor, we think that it is better to use the term "physical" and not "planted" area. The ratio between harvested area and physical area in gcamland is used to reconcile the difference between land cover and FAO harvested area.

The reviewer is correct that we are missing the reduction in crop failure due to technological progress. We capture many aspects of technological progress through changes in yield in gcamland; however, those yields are derived from FAO production and harvested area and thus only capture changes in production on land reported as harvested.

Multiple cropping is captured by the harvested to physical land ratio implemented within gcamland. This ratio varies across crops in the USA, but ranges from 1.01 (harvested area is 1% larger than physical land) to 1.05 (harvested area is 5% higher than physical land). The lower values are for FiberCrop, FodderGrass, and Rice. The higher values are for Wheat and MiscCrop.

While the ratio of harvested to physical areas are low in the USA, the reviewer is correct that these factors are much more important in some other countries/regions. Within gcamland, a discrepancy in the harvested to physical ratio would not impact harvested land by crop, as profits are driven by total crop sold in a year (regardless of the number of harvests) and the harvested area can be preserved/recovered through the ratio of harvested to physical area. However, the physical land area allocated to crops does depend on this ratio; as a result, the amount of land available for other land types will be influenced by the choice of harvested to physical area used within each region. As a result, the implications for cropland harvested area in this study are not directly influenced by this ratio (or any errors in it), but the ability to accurately model land cover could depend on the choice of this ratio. We do note that the ratio used in gcamland is constrained by the land cover observations in the initial model year, minimizing the effect of this

ratio on land cover. That is, total physical area is derived from land cover estimates. The largest effect of this ratio will be on the amount of OtherArableLand, which will increase if the harvested to physical area ratio increases.

**Author Changes:** We have updated the text to use "physical" instead of "planted". We have added both physical and harvested land to the additional excel table of result so the user can see the difference between these values for all commodities. We have also added both physical land and harvested land in 1990 to Table S1. Finally, we've added a couple of sentences to the discussion to highlight the potential importance of this outside of the USA: "). For multi-cropping, gcamland includes both harvested and physical area; however, the ratio between the two is held constant. Intensification through multi-cropping could be more important when extending the study outside of the United States; however, additional data and investigation is needed."

4.      Resolved

5.      Resolved

6.      Over time total area of agricultural land in the US has declined sharply, due to conversion to non-agricultural uses of land (urbanization, infrastructure, . . ..). How gacamland handles conversion of land to non-forestry-ag land. How land availability land has been taken care off over time? Is it an exogenous variable in each year?

**Author Response**: While urban land has grown over time, the definition of urban land in gcamland accounts for only 1% of total land area in the United States. We hold this area constant in the simulations presented here (equal to 1975, 1990 or 2005 values depending on the calibration year used in the simulation). Allowing this to change over time would not have a noticeable impact on results given how small the area is. Similarly, we hold tundra and rock/ice/desert constant in time, but they account for very small amounts of land in the United States (2.6% and 0.4%, respectively). All other land is included in the economic competition.

**Author Changes**: We have added a footnote to the methodology section: "A small amount of land (~4%) is considered non-arable in gcamland in the United States, including urban, tundra, rock, ice, and desert. This land is held constant throughout the simulation time period."

Thanks for this clarification. However, it does not help and makes further confusion.

- The paper does not provide a clear picture of the land categories in the model. While you are talking about non- arable land as a part of land use modeling, the logit nesting structure does not cover it. If this type of land is not modeled, why it is described in the paper.
- Table 1 introduces three nests: Arable land; Ag land. Forest, and other; and Cropland (or crops). This plus other information provided in the tale bring in mind the following nesting structure:

[Figure]

Rice wheat corn soy, ….

Forest    Ag land    other

Arable land

However, there is no "*cropland*" in the middle nest. What is the relationship between Ag land, cropland, and pasture land? It seems a nest is missing.

- It is noted FAO data with some modification is used to determine land data. If that is the case, then something is not correct. Please let me explain. Arable land is a subset of cropland in the FAO data. For example, in the FAO data for 2018, area of US cropland is 160,437 thousand hectares with sub component of 157,737 thousand hectare of arable land. This is in a sharp contrast with the nesting structure defined in the paper. Also, area of US Ag land is 2.6 times of its arable land. How Ag land could be a subset of arable land?

The paper should make proper clarifications on this issue.

Finally, let us follow the nesting structure outlined in the paper. According to the FAO data, total area of US arable land has declined from 180,630 thousand hectares in 1961 to 157,727 thousand hectares in 2018, a decline by 22,893 thousand hectares. How this reduction has been handled? It is not a small reduction.

**Author Response:** In terms of the nesting, we had focused this part of methodology section on the aspects of gcamland that we perturbed in this experiment. It is clear from the reviewer's response that this has caused confusion.

For "arable", we were using this word in the colloquial sense ("suitable for growing crops") and not in the way that FAO defines it ("areas under temporary crops, temporary meadows and pastures, and land with temporary fallow"). We can see that this caused confusion and given the paper's dependence on FAO data we have decided to remove the word "arable" in almost all instances in the paper. The one exception is when we define the categories in gcamland; in this case, we think it is important to use and define the terminology in gcamland. Gcamland only uses the word "arable" in the name of "OtherArableLand" category, so this is the only case in which "arable" will appear in the paper. Note that throughout the paper we have on occasion used plain language descriptions or lengthier titles for categories and names in gcamland to make the paper easier to read; within gcamland, we use shorthand and abbreviations. We have now added a table to the SM (Table S2) mapping the precise terminology used in gcamland to the plain language descriptors used in the paper.

Thanks for these additions and clarification. They help to understand your work.

Since several items identified and mentioned in table S2 are not standard data and you guesstimated them, they have to be revealed and presented in Table S2. I propose to add a column to table S2 and show the area of each type of land for at least one representative year.

**Author Response:** We have added a column to Table S1 and S2 with area in 1990 (the default calibration year). Note that the areas listed in Table S2 are the sum of all land area in that node.

**Author Changes:** We have added a column to Table S1 and S2 with both physical and harvested area in 1990.

7. A big issue in land use modeling is marginal cropland (idled land under CRP, cropland pasture, other types of idled). Area of idled cropland in the US have changed a lot. The definition of "cropland pasture" has also changed over time. How idled land is treated?

**Author Response**: Idled land is called "Other Arable Land" in gcamland. The amount of land in this category can change over time based on economic signals. Since idled cropland does not produce a product, its profit rate is exogenously specified like other land cover types (see response to comment #12). Note that this exogenous value is similar to a CRP payment, as it represents a marginal benefit of keeping land fallow.

**Author Changes**: We have added a footnote to the methodology section: "Fallow cropland (called other arable land in gcamland) is also included in this nest." We have also added text to the supplemental material indicating that idled cropland is included in gcamland.

This is a misleading response. FAO data base does not have other arable land. Please at least, provide a table and put all types of included land in each nest at list for three years, including cropland and its components.

**Author Response:** We derive the OtherArableLand category from harvested area data from FAO and total cropland area data from the land use harmonization (LUH) product. LUH provides total land cover classified as cropland. We assume the difference between land planted in crops and land cover of crops is idled land. See also the response to #3. We are unclear what the reviewer meant by 'at list [least?] for three years' and thus have not responded directly to that.

**Author Changes:** We have added a nesting diagram (Figure S1) and two tables to the SM (Table S1 and S2). These figures and tables include all land types in gcamland, where they are nested, what we call them in the paper, and where we get the initialization and simulation data. We believe that this provides full transparency for readers in clear, easy-to-understand forms.

Thanks for the clarifications and sorry for the typo ("least" not "list"). I think it was clear that I am asking for a table including your land data and you refused to provide it. Your new table S2 reveal that many items in your land data (in particular for non-cropland) are just guesstimates.

You have to reveal your data at least for a one representative year. Table S2 is a good place to do that.

**Author Response:** Thank you for the clarification on your comment. We have added 1990 land area to Tables S1 and S2 and are providing land area for all simulations in a separate excel file.

**Author Changes:** Thank you for the clarification on your comment. We have added 1990 land area to Tables S1 and S2 and have provided land area for all simulation years in a separate excel file. In Table S1 and the excel file, we provide both physical and harvested area where appropriate.

8. It is noted that harvested area from FAO is used. FAO is missing many feed crops since 2011, including million hectares of those crops. Without proper steps to cover missing crops in FAO, the estimated parameters will be subject to major issues and biases. Figure three suggest that those feed crops is missed. That is a major issue.

**Author Response**: gcamland includes fodder and feed crops, using data from FAO prior to 2011. We have excluded it from the comparison and statistics because the data is not available after 2011 as you noted, but it is included in the modelled results.

**Author Changes**: We have revised the Figure 3 caption to clarify this: "Note that fodder crops are included in gcamland but are excluded from total cropland area in this figure due to data limitations." We have also added figures to the supplemental material showing all crops, including fodder.

This is a misleading response. If you fixed the data (for after 2011) by estimating the missing data items, then that should be reported and included in your figures. Transparency is the key.

**Author Response:** We did not fix the data. Gcamland only needs land cover data in the base year (1975, 1990, or 2005 depending on the historical time period to be simulated); FAO has data on fodder crops for all three of those years. With that quantity for use as a base year, we then simulate fodder crop land cover throughout the simulation time horizon (1975-2015, 1990-2015, or 2005-2015 depending), but we do not use FAO data in our simulations after the base year and we do not fix the FAO data at all. Therefore, while we have simulated fodder crop land cover for the entire simulation time horizon, including years beyond 2011, we cannot compare the later years of the simulation to observations because they are not provided by FAO after 2011. This is why fodder crops are excluded from the NRMSE calculations.

**Author Changes:** We have added a table to the SM (Table S1) that lists all gcamland types, along with the source of data for initialization and for comparison.

Thanks for the clarifications. You need to revise the paper to make this clarification.

In addition, it seems Figure 3 shows harvested area not cropland. Change the title and the legend to harvested area. As you correctly highlighted in your introduction, in the US, cropland area remained flat while harvested area has increased over time with some fluctuations.

**Author Response:** We have included a footnote to Table S1 that describes the fodder data specifically. We have also clarified the approach to calibration and sensitivity analysis, including the differences in data used, in response to this comment and one by the editor. The caption to Figure 3 did indicate this was harvested area; however, we've added this to the title of the cropland facet to make this clearer.

**Author Changes:** We have included a footnote to Table S1 that describes the fodder data specifically: "FAO only includes fodder data through 2011. We use this information to initialize the model, but do not use it beyond initialization since it is unavailable for all simulation years." We've adjusted the title of the cropland facet to say: "Harvested crop area". Finally, we've clarified the approach to calibration and sensitivity analysis as part of the restructure and edit to the methodology section.

9. GCAM is using commodity price to model land allocation. It seems wholesale farm prices is used. That is a bad proxy for exporting crops such as cone and soybeans. For example, half of soybean is exported at much higher price farm price.

**Author Response:** The producer price is the relevant price signal to be used for planting decisions. The market price, and thus the producer and consumer prices, is a function of the demand sectors as well, which includes domestic demand and exports. However, the resulting equilibrium price paid to producers is the relevant price regardless of how the demand is determined. Therefore, we feel that producer price is the right input into gcamland for this analysis.

**Author Changes:** We have revised the manuscript to better document what is included in gcamland. See also the responses to comment #1 and comment #19.

Yes, as it is noted in this response "price paid to producers is the relevant price", but the FAO whole sale price does not reflect the price received by farmers. You need to address this as a major limitation of your work clearly. Please explain what revision you have made? And be more specific.

**Author Response:** We are not using wholesale prices. We are using producer prices from FAO, which FAO defines as "prices received by farmers" for the reasons above. We are not clear on what in the manuscript led the reviewer to think we were using wholesale price, but we are using producer prices (as indicated by the reference to the FAO producer price dataset).

**Author Changes:** We have added "producer" as a qualifier on "price" in section 2.4.2 to clarify this.

Sorry for the confusion between the producer price and whole sale price. Yes, I understand that you used the FAO "producer price" data. But that is not the point controversy. The point is that price received by farmers (including subsidies) is the relevant price and you used the producer price (farmgate price). Producer price is not equal to price received by farmers.

**Author Response:** Thank you for the clarification. We have added this information to the text, clarifying what is and isn't included in the price used in gcamland. This has been added to the methodology section just prior to our discussion on subsidies and subsidy data.

**Author Changes:** We have added a clarification on the price data used: "The producer prices used in gcamland are defined as "prices received by farmers…at the point of initial sale" or "prices paid at the farm-gate" (FAO, 2018a) and thus do not reflect subsidies. However, subsidies are a reality…"

10. Resolved

11. How biofuels were included in the simulations? Biofuels and biofuel policies were major drives of land use. How that included in your simulations

**Author Response**: Biofuels and biofuels policy are reflected in our model through changes in producer prices of crops.

**Author Changes**: We have added this to the footnote explaining differences between GCAM and gcamland: "Changes in demand are explicitly represented in GCAM. In gcamland, changes in demand are captured through changes in price. For example, the increase in demand for corn and soybean due to biofuels policy is captured through changes in the prices of these goods."

This response makes me more concern. I am not convinced that your work properly identifies source of piece changes. You approach only consider changes in the prices and send that as a signal to the land supply tree, without identifying the sources of price change. The source of price change could be demand shock (e.g. biofuels) or supply shock (reduction in yield or land supply). You approach does not distinguish these sources and simply consider price changes as signals to the land supply. Can you convince the readers that you do not need to identify the source of price change?

**Author Response:** Our model was not designed to identify the source of price changes as we focus on supply responses. However, we do represent supply and demand shocks differently. The direct effect of a supply shock (e.g., a yield change) is explicitly represented in gcamland; demand shocks (e.g., biofuel policy) are signaled by changes in prices. This information is enough for landowners to make land allocation decisions based on the relative rental profits, particularly under perfect foresight. In the cases with imperfect expectations (e.g., adaptive expectation), we allow a different expectation coefficient for biofuel crops (i.e., Corn and OilCrop) to reflect that the price expectations of these biofuel crops could be different than other crops. In other words, biofuel shocks in our model indirectly affect price signals and/or expectation schemes. As discussed in Section 4.1, the results indicate that Corn and OilCrop rely less on past information than other crops in the Adaptive expectations.

**Author Changes:** None.

Thanks for this explanation. You need to include this justification in the paper.

**Author Response:** Thank you. We have added this information to the paper.

**Author Changes:** We noted how demand shocks are included in the study in the methodology section: "In gcamland, changes in demand are captured through changes in price. For example, the increase in demand for corn and soybean due to biofuels policy is captured through changes in the prices of these goods." We have also added information to the discussion: "Finally, our study focused on land supply responses and did not identify the sources of price changes. Future studies could extend our model structurally to explicitly identify demand shocks, responses, and their effects on prices."

12.     Th dapper highlights that gcamland uses commodity prices in land allocation. But the model allocates land across land cover items. What prices are used for forest products, livestock products, etc.? The paper is silent on these prices. What prices were used for land cover items

**Author Response**: We calculate land rental prices for commercial forest and pasture using their product (forest or livestock products) prices and related productivity and cost information. For non-commercial land cover only items, effective profit rates are derived during the calibration process to ensure that the amount of land area in the base year predicted by the logit equation matches the read in value. For subsequent years, these effective profit rates are held constant. This estimation is described in detail in Wise et al. (2014).

**Author Changes**: We have added this information to the methodology section: "Profit rates ($r_j$) at the lowest level of the nest are computed based on price, cost, yield, and subsidy (if included) for land use types (crops, pasture, commercial forest); profit rates for land cover types are input into the model and are based on the value of land."

- This response makes no clarification. For example, it is said that: "We calculate land rental prices for commercial forest and pasture using their product (forest or livestock products) prices and related productivity and cost information." Please be very specific and answer the following: What are the prices of livestock products? How about prices of forest products? How do you measure yield for forest? How about yield for pasture land?

- If only arable land is modeled, why do you need profit rates for non-commercial land. As noted in my earlier comments, providing a table including all nests and their members could help.

- What is the effective rate for unmanaged forest? How do you determine it in the calibration prices? Calibration to what?

- The paper should clearly explain the above points

**Author Response:** We estimate the yield for forest from its vegetation carbon density and the yield for pasture is set to the yield of hay. Forest prices are calculated from FAO's export data. Pasture price data is not available; instead, we set these prices to 70% of the alfalfa price (used for FodderHerb).

For arable land, as noted in our response to #6, the choice of this word has caused confusion. We include all land use and land cover categories in gcamland and thus need profit rates for each. The effective profit rate is the profit rate that would be needed to return the land allocation in the base year. The calibration process ensures that land allocation in the base year exactly matches the input values. To do this, we determine what the profit rate would need to be for the logit to predict the input (referred to as the "effective profit rate" above). For non-commercial land, we hold these effective profit rates constant in the future. As noted above, this procedure is described in Wise et al. (2014). We cannot repeat everything from that paper; thus, we refer the reader to that paper for a discussion of the calibration procedure, as is common in science.

**Author Changes:** We have added a nesting diagram (Figure S1) and a table (Table S1) that specifies each land type in gcamland to the SM. This table includes the type and sources of all input and comparison data used.

Thanks for the clarifications. Based on Table S1 and its footnote # 5, no data is available to calculate profitability of non-commercial land types. Therefore, you used an unknown approached to calculate profits for these types of land in the base year land allocation. The calibrated profits of these land types (including other arable land) for the base year should be revealed. Table S2 is a good place to show them along with the calculated profits for the commercial land types.

Let us assume that the calibrated profits for these land types are credible. How do you determine changes in these profits after the base year? Please be very specific in your response.

**Author Response:** The approach is not unknown and has been documented in previous papers. As mentioned above, we hold the profits of non-commercial land constant in future periods. However, with the nested logit-sharing framework, this implies a land supply curve for non-commercial land. This approach offers more modelling flexibilities with traceable results compared to approaches of assuming non-commercial lands to be inaccessible and fixed over time or aggregating non-commercial lands with commercial lands.

**Author Changes:** We have added a column to Table S1 with profits in 1990. We have also added the following sentences in Section 2.1.2: "Profit rates for commercial lands evolve over time as price, cost, yield, and subsidy change. Profit rates for non-commercial lands are constant over time. The logit approach effectively depicts a supply curve for non-commercial land with the land supply elasticity implicitly determined by the logit exponent and the assumed rental profit rates (i.e., implying a cost of land transition). The supply curve approach, which views the amount of land available as endogenous, offers more modelling flexibilities with traceable results compared to approaches of assuming non-commercial lands to be inaccessible and fixed over time or aggregating non-commercial lands with commercial lands (Dixon et al., 2016)."

13.     Resolved

14.     Resolved

15.     Resolved

16.     Resolved

17.      Resolved

18.     Resolved

19.     Resolved

20.     Resolved

21.     See comment #24

22. The main manuscript only presents comparison of the projected and observed harvested areas and provided no comparison for other land types.

**Author Response**: The comparison of projected and observed area for other land types is included in the supplemental material for types where observations are available.

**Author Changes**: The comparison of projected and observed area for other land types is included in the supplemental material for types where observations are available. We have also added a paragraph to the supplemental material stating which land types are in gcamland and explaining our choice of what to show where: "The main text of this paper focuses on four commodity groups (Corn, Wheat, OtherGrain, and OilCrop), as these four commodities represent the largest land area in the United States. However, gcamland includes twelve commodity groups in total, representing all crops reported by the FAO, and fallow or idled cropland (referred to as other arable land in gcamland). In addition, gcamland includes commercial forest and pasture, as well as several other land cover types, including forest, grassland, shrubland, tundra, rock/ice/desert, and urbanland. We include results for other agricultural commodities and the land cover types where observations are available in this section"

Sorry, I do not consider this as a satisfactory response. The paper and its supporting material provide inconstant and confusing information about the land cover types, land uses, and components of each of the three nests included in the model. Simply revise table 2 and clearly put all land types and land uses in that table for each nest.

For example, in the main text in table 2 pasture land is not a part of middle nest. But it appears in the SI in Figure 7 as a component in the mix of grassland, shrubland, and pasture. Very confusing.

The main text should clearly represent the nesting structure and the component of each nest. Do not refer to another paper. This is an essential information for this paper.

Figure S7 should show data for each land cover item including managed forest, unmanaged forest, pasture, grassland, shrubland, any other components of the non-cropland nest, and cropland as one land cover type. If your model does not trace changes in some land types, that type of land should not be included in the model nor in the paper.

As a subcategory of cropland, the projections for unused cropland and their corresponding observations should be presented and compares.

**Author Response:** We have added a nesting diagram and a table to the SM with information on every land type included in the paper. We have also included a figure in the SM with model results and observations (where available) for all land types. We have included this an additional figure rather than an expansion of the original Figure S7 (now Figure S9) because in some cases the observations we have are for a sum of two gcamland land types. The new figure shows land by type for each individual type included in gcamland.

**Author Changes:** We have added a nesting diagram (Figure S1) and a table to the SM (Table S1) with information on every land type included in the paper. We have also included a figure in the SM (Figure S11) with model results and observations (where available) for all gcamland land types. We have also clarified when observations are available and when they are not, including the information in Table S1 and a description of what is included in the various figures and why: "Finally, Figures S6-S10 and Table S4 focus on comparing gcamland simulation results to observations for categories or sums of categories where observation data is present. However, there are other land types included in gcamland (see Figure S1 and Table S1). Figure S11 shows the evolution of all individual land categories in gcamland for the default simulations, with observation data plotted when it is available for the individual category."

Thanks for these Changes. Some of these changes are good. Indeed, Figure S11 mentioned in this response, provides no additional information. The scales used for these panels hide the variation and difference between alternative cases over time. In most cases, except for grassland and shrubland, the areas presented by these panels are smaller than 50 km$^2$, while the max value of the vertical axis in each panel sets to show 3000 km$^2$. To be transparent and help the readers to evaluate your work, it is critical to show the numerical results presented in the panel of figure S11 and their corresponding actual observations in a simple plain table. The table should include the simulations results for all land types listed in Table S2 and their corresponding observations. Providing an Excel file attachment is a better option.

**Author Response:** We have added an excel file with all of the results from Figure S11. We have also added 1990 land areas to Tables S1 and S2.

**Author Changes:** We have added an excel file with all of the results from Figure S11. We have also added 1990 land areas to Tables S1 and S2.

23. Results are highly aggregated into four groups of crops. How about the 12 categories of crops in GCAM?

**Author Response**: The main manuscript shows four of the 12 crop categories. We have included figures showing the other categories to the supplemental material.

**Author Changes**: The main manuscript shows four of the 12 crop categories. We have included figures showing the other categories to the supplemental material. We have also added an explicit reference to these figures in the main text.

Please, add changes in unused land. That is an important piece of information.

**Author Response:** I am not clear on what the reviewer means by "unused land", but we have added information on all land types included in gcamland.

**Author Changes:** We have added information and results for all land types included in gcamland, including a nesting diagram (Figure S1), a table with data sources (Table S1), and a figure showing changes in land allocation for all gcamland land types (Figure S11).

Unused land is a common terminology and refers to the cropland which has not been used for crop production (CRP, cropland pasture, set aside land, fallow land, ….). In the nesting structure provided in the new revision you named it "other arable land".

**Author Response:** Thank you for the clarification. We'd prefer to keep the terminology as is in the paper to minimize confusion.

**Author Changes:** No changes.

24. The figure S5 of SI shows major errors for the change in forest area. This show that the model fails to represent changes in forest area correctly.

**Author Response**: Figure S5 had included the net change from 1990 to 2015 for the modeled data and the net change from 1992 to 2015 for observations for forest. Figure S7 shows the whole time series. As shown in S7, the time series tracks the observations fairly closely for the adaptive expectations. We do not think it is correct to say that the model fails to represent changes in forest area; however, we do think that Figure S5 was confusing given the unit used and the differences in time horizon.

**Author Changes**: We have removed Figure S5 as it did not add any new information and was confusing (see response to comment #25).

First thanks for adding figure S7. Adding this figure is a step forward. As I mentioned in comment # 23 you need to extend this figure for all land cover items. In particular, it is important to show errors in %, not in levels. Level variables hide errors. Please show the percent errors, then we can judge the model performance in land cover items. Also remember that you failed to calculate the goodness of fit (in your language NRSME) for land cover items.

Figure S7 shows bad performances for land cover items that already are included in this figure. In particular, I see very large differences in level variables between the performance and observed items. You also could alter the scale of this figure to better see the errors for forest. You should show the errors in percent to show the model performance. I believe it is straight forward to calculate NRSME for these items. Why not?

**Author Response:** We have added a calculation of NRMSE including dynamic land cover types. As noted in our response to comment #15, we are keeping this as a sensitivity since we do not have a complete time series of observations for land cover types.

As for percentage errors, percentage errors can mask as much as absolute errors (like an unnormalized rms) for this sort of multi-target work. Percentage errors often overemphasize errors in land types with small land allocations. For example, the largest error in percentage terms in any of the gcamland simulations is for PalmFruit. In absolute terms, this error is virtually zero. NRMSE uses the standard deviation of observation to normalize errors rather than using percentage errors to address this point. For this reason, many model validation studies rely on NRMSE as a measure of error. We have now added a table showing absolute error, percentage error, and NRMSE; we have also included a discussion on our choice to focus on NRMSE instead of percentage errors.

**Author Changes:** We have added a calculation of NRMSE including dynamic land cover types where observation data is available (see also response to comment #15). We have added a table to the SM (Table S4) showing absolute error, percentage error, and NRMSE, as well as a discussion on the different metrics: "Finally, Table S4 summarizes the error (simulation minus observation) in both absolute (million $km_2$) and percentage terms, as well as including NRMSE for each expectation type and crop. We include all three metrics in this table; however, in this study, we primarily use NRMSE. Normalized measures of error are key for interpreting whether a simulated data set acceptably replicates available observational data. While normalizing to present errors in terms of percentages is common, this can result in large magnitude percentage errors when dealing with multiple variables (land types) with a wide range of magnitudes. Given the significant difference in land historically allocated to different uses in the United States (e.g, the PalmFruit vs Corn commodities in gcamland) and the fact that we are seeking parameter sets to minimize error measures across these commodities, this can lead to misleading results. Rather, we follow the literature normalizing by the standard deviations of observations (Nash and Suttcliff 1970; Willmott 1981; Legates and McCabe 1999; Willmott et al 2012;

Tebaldi et al 2020), captured in our NRMSE. This allows a benchmark of whether the discrepancies between simulated and observed data fall within the natural variability of the observed data, giving a statistically justifiable benchmark to determine whether those discrepancies are acceptably sized."

Thanks for these clarifications All the issues.                  Please provide the table mentioned in comment # 22 to resolve

**Author Response:** We have added an excel file with all of the results from Figure S11. We have also added 1990 land areas to Tables S1 and S2.

**Author Changes:** We have added an excel file with all of the results from Figure S11. We have also added 1990 land areas to Tables S1 and S2.

25. Resolved

26. Resolved

27. Resolved

28. Resolved

29. The examined practice estimated a few parameters of the model for land use. A good way to test the outcomes of this practice is to run the GCAM model with the estimated parameters and compare the model results for land use changes, land cover changes, changes in crop prices, and changes in yield with actual observations over the examined period.

**Author Response**: We agree that this is a useful test and intend to do it in subsequent work. However, this paper is focused on parameter estimation and we think that adding those simulations to this paper would unnecessarily complicate the existing text and analysis.

**Author Changes**: None.

I am glad that you consider this test as a useful test. But it is not only useful. This is an essential task. Validation of estimated parameters is a crucial task. Indeed, without this test you do not know how good the estimated parameters for projection are. Running a validation test may require more work, but it is an essential task.

I observe that the revised version noted that: "we have focused on the historical period. However, these models and parameter estimates could be used in a simulation of future land use and land cover change to better understand their implications." Even you have not shown how your model project historical data. You only used historical data to estimate the logit parameters.

But failed to test how good are the estimated parameters to replicate the historical. That should be clearly acknowledged in the paper.

If you choose not running this test, then the paper should clearly acknowledge that you have not examined the validity of your practice in the abstract, discussion section, and conclusion.

**Author Response:** We believe the reviewers response to our previous answer stems from the confusion between gcamland and GCAM, as discussed in comment #1. We have used the parameters in gcamland, re-run the model, and compared the results to observations. That is, we have shown how gcamland simulates land area in the historical period using the new parameters (see Figures 3-6). We have not run GCAM, which would require us to replicate this analysis for 31 other regions (GCAM includes 32 regions). In addition, GCAM includes models of agricultural demand and links to the energy system, all of which will affect land allocation. We do think running the updated parameters through GCAM is useful future work, but it is outside the scope of this paper.

**Author Changes:** We have revised the final paragraph clarifying what we have done and why we are leaving the GCAM analysis to future work. The first sentence now reads (new text in italics): "In this paper, we have focused on the historical period, *simulating land allocation in gcamland over this period and comparing it to observations*." And, we've added this sentence about the potential GCAM validation test: "However, for such a study, we would need parameter estimates for all thirty-two GCAM regions and not just the United States." We have also added an overview of the methodology, including the simulation of historical land using gcamland and comparison with observations to the beginning of Section 2 (see response to comment #4).

Thanks for the clarifications and additions. However, to be clear, this paper basically is nothing more than an incomplete calibration practice based on some ad hoc assumptions under several limitations. The outcome of this calibration process is note validated. This paper provided no evidence regarding the outcome of GCAM for replications of historical observations using the calibrated parameters.

**Author Response:** This paper demonstrates how to estimate parameters for an economic model of land use and land cover change, gcamland, in a region of interest. This is an improvement over existing studies which rely on heuristics.

**Author Changes:** We have clarified the goal of the paper in the abstract and have noted in the discussion what is required to to implement this in GCAM.

30. Resolved

31. The abstract provides trivial information. It is not an abstract of this paper.

**Author Response**: We have revised the abstract.

**Author Changes**: We have revised the abstract.
With all due respects, the revised abstract needs a major work. The fist six lines provide a lecture to justify this work. Those should be included in the main text not in the abstract. Then it is noted that: "We run gcamland simulations with these parameter sets over the historical period in the United States to quantify land use and land cover, determine how well the model reproduces observations". This is not what you did. you have not quantified "land use and land cover" over time. You used historical data to estimate some model parameters. You have not determined how your model produces actual observations for the estimated parameters. This is the validation test that you refused to accomplish. You have not highlighted your findings on the sizes of the estimated parameters. You have not highlighted the limitations of this work. You need to revise the abstract.

**Author Response:** First, we are following the guidance provided by the Nature journals on how to write an abstract or summary paragraph (https://www.nature.com/documents/naturesummaryparagraph.pdf). Such guidance recommends providing an introduction to the field (both general and specific) to motivate the work before describing the specific contribution of the paper. It does not include limitations of the work in the abstract, instead that is included in the discussion.

Second, we have quantified land use and land cover in the historical period (see also response to comment #29). The statement in the abstract is correct as written. For each randomly sampled set of parameters, we initialize gcamland to a base year of data (e.g. 1990) and then use the parameter sample set to simulate land use and land cover in subsequent years (e.g. 1991-2015) without incorporating additional historical data. We then compare the gcamland simulated

Thanks for saying that the abstract is prepared based on the guidance provided by the Nature journal. But the guidance does not ask you to write about general materials that are the core of your paper. The abstract consists of about 280 words. One third of it describes materials that are not really the core of your paper. Your aper is not about future changes in land use, it is not about water use, it is not about uncertain socioeconomic condition, and it is not about technological progress that you referred to. These are important issues, but not directly presented in your work. after all, this paper basically uses a calibration process under uncertainty in crop prices. Indeed, this paper calibrates a few parameters that govern land supply. You need to tune your abstract to say the importance of this calibration practice.

Again, your work does not simulate historical observation. It uses historical observation to calibrate land allocation parameters. Running GCAM model with the calibrated parameters to replicate historical observation is a valuable practice, but you clearly admitted that you are not able to perform this valuable practice with incomplete calibration process. You should not oversell your work or claim for something that has not achieved. The abstract should say: 1) the paper proposes a calibration method for the GCAM land allocation module; 2) the proposed method is tested for the case of US; 3) what are the calibrated and selected parameters; 4) what limitations are involved; 5) what changes are needed to run the calibration for other regions.

**Author Response:** We have revised the abstract based on your comments and those of the editor.

**Author Changes:** We have revised the abstract based on your comments and those of the editor: "Future changes in land use and cover have important implications for agriculture, energy, water use, and climate. Estimates of future land use and land cover differ significantly across economic models as a result of differences in drivers, model structure, and model parameters; however, these models often rely on heuristics to determine model parameters. In this study, we demonstrate a more systematic and empirically-based approach to estimating model parameters for an economic model of land use and land cover change, gcamland. Specifically, we generate a large set of model parameter perturbations and run gcamland simulations with these parameter sets over the historical period in the United States to quantify land use and land cover, determine how well the model reproduces observations, and identify parameter combinations that best replicate observations. We also test alternate methods for forming expectations about uncertain crop yields and prices, including adaptive, perfect, linear, and hybrid approaches. We find that an adaptive expectation approach minimizes the error between simulated outputs and observations, with parameters that suggest that for most crops, landowners put a significant weight on previous information. Interestingly, for corn, where ethanol policies have led to a rapid growth in demand, the resulting parameters show that a larger weight is placed on more recent information. We examine the change in model parameters as the metric of model error changes, finding that the measure of model fitness affects the choice of parameter sets. Finally, we discuss how the methodology and results used in this study could be used for other regions or economic models to improve projections of future land use and land cover change."

32.    Resolved

---

## Author Response (AR4)

**Reviewer number 2**

Author Response: Thank you for the helpful comments. We have revised the paper in response to your comments. We are adding our responses in purple font in the document below. We have also removed any prior comment that is now resolved to simplify the document.

Author Changes: We have revised the paper in response to your comments. We include a point- by-point description of the changes below.

Thanks for detailed responses. The revised version is much improved. The response to one of the comments needs additional attention. In what follows I highlighted the comment which needs additional work. I also added a few minor new comments on the abstract. My responses are in red fonts in blue background

Author Response: Thank you for the comments. We have revised the paper in response to your comments. We are adding our responses in white font with navy blue background.

Author Changes: We have revised the paper and include a point-by-point description of the changes below.

1.    The model clearly uses a nesting logit format, perhaps three nests. Equation 1 of the paper shows only on nest. The formula should be replaced with a formula for the full nest.

Author Response: We have added information on how profit and shares are calculated for the other nests.

Author Changes: We have added additional text and an equation "In the three-level nest version, land allocation at each level is determined by a modified version of equation 1, where Y is replaced by the land allocated to that particular nest. The land allocated to a particular nest is dynamic and varies over time. Profit rates ($r_j$) at the lowest level of the nest are computed based on price, cost, yield, and subsidy (if included) for land use types (crops, pasture, commercial forest); profit rates for land cover types are input into the model and are based on the value of land. Profit rates for higher levels of the nest ($r_{node}$) are determined by: $rr_{nnnnnnnn}$
$= [\sum_{j=1}^{n} (\lambda_j r_j)^\rho]^1{}_{/\rho\rho}$

The revision mentioned here is helpful, but it provides a broken formulation and misrepresent your work. To be fully transparent and to make sure that the readers understand that three exponent parameters (distribution parameters) and there sets of share parameters were estimated, I propose to use the following format:

$XX_{jjjjCC} = \sum \underline{\qquad\qquad}_{(jj}{}^{\lambda\lambda}{}_{(\lambda\lambda cc jjjj}{}_{ccjjjj}{}^{rr jjjj}{}_{rr^{cc}jjjjcc)}{}^{\rho\rho}{}_{)\rho\rho}{}^{CC}{}_{CC} \cdot YY_{jj}{}^{CC}$ where "C" represents the crop nest including rice, wheat, ……

$XX_{jjjjAA} = \sum \underline{\qquad\qquad}_{(jj}{}^{\lambda\lambda}{}_{(\lambda\lambda AAjjjj}{}_{AAjjjj}{}^{rr jjjj}{}_{rr^{AA}jjjjAA)}{}^{\rho\rho}{}_{)\rho\rho}{}^{AA}{}_{AA} \cdot YY_{jj}{}^{AA}$ where "A" represents the ag-forestry nest including cropland, forest, ….

$XX_{jjjj}{}^{RR} = \sum\_\_\_\_\_\_(\lambda\lambda_{(\lambda\lambda^{RRjjjj}{}_{RR}rr_{jjjj}rr^{RR}{}_{jjjjRR})\rho\rho})_{\rho\rho}{}^{RR}{}_{RR} \cdot YY_{jj}{}^{RR}$ where "R" represents the arable land nest including …..
$\quad\quad{}_{ii}\quad{}_{jjjj}$

The members of each nest should be clearly mentioned. In particular, for the crop nest all members of this nest including unused land, cropland pasture, CRP, other idled land should be specified. The information that presented in Table 2 is incomplete and very misleading. Of course, the names of variables should be described fully and maintain this convention through the paper.

**Author Response:** Please note that the equation did not transfer well from the PDF of the review and thus appears strangely in the comment and response above. We have used the original PDF with the correct formatting when reading and reacting to this comment. The same equation is used in all three places; given the confusion this seems to have caused, we have now repeated it as suggested by the reviewer. Given the number of crop categories in gcamland, listing all crops in Table 2 made it difficult to read the table, so we have opted to provide the full list in a footnote. We also include a complete nesting diagram with all information in the same figure in the SM. To facilitate readability, we have opted to use plain language descriptors in the paper rather than the abbreviated terminology used with the gcamland code. However, we do agree that we need to specify the mapping between plain language and terminology, so we have added a table to the SM.

**Author Changes:** We have repeated the logit equation as suggested by the reviewer. We have added information on what is contained in the nests, using the node names (now defined in Table S2) in the main text and the complete list of crops in footnote 6. We have added a nesting diagram to the SM (Figure S1) which indicates which members are in which nests. We have added a table to the SM (Table S2) that maps the plain language descriptors used in the main text of the paper to the precise terminology in the gcamland model.

Sorry that the pdf file including my comments was not presenting the suggested formulas correctly. The pdf version that I posted on the Journal website shows the formulas correctly. Anyway, I see that you correctly specified and defined the proposed equations. Thanks for that and thanks for adding Figure S1 and Tables S2. Your work is now more transparent and traceable.

However, I see that table S2 in defining A, R, and C refers to equation 1 which is wrong. It should refer to equation 2.

Figure S1 represents 7 nests for the supply structure of gcamland. On the other hand, Table S2 shows that your calibration process estimates distribution parameters for only three of them. The rest (including 4 distribution parameters) were given some ad hoc values. For the root nest, it makes sense to use a zero value, as you assumed no change in the areas of members of this nest. However, it is not clear from where the values of 2.7, 0.05 and 1.575 are come from. Explain how those values were determined? The paper should explain the sources of these ad hoc values.

**Author Response:** We have corrected the reference to equations in table S2. For the other logit exponents, we have used the default values used in GCAM. We have chosen not to vary these in this exercise as they do not directly impact the amount of cropland area, which was the output we focused on in the main text. We have done an additional sensitivity analysis

quantifying the impact of these logits on cropland area. In this analysis, we doubled each logit one at a time. The area allocated to each crop changes by less than 1% (the largest change in magnitude is -0.12%). These logits do have a larger impact on other land types. For example, doubling the ForestLand logit results in a shift in the distribution of commercial and noncommercial forest, with commercial forest increasing by as much as 27%. However, total forest is largely unchanged (maximum change of 0.22%). We have added this information to the supplementary material, near Table S2.

For the default values, we realize in re-reading the previous GCAM papers that while we have documented the approach to selecting logit exponents in Wise et al. (2014) and the specific numbers used are available on GitHub (github.com/jgcri/gcam-core) we have not documented specifically how those numbers were chosen. We have now added this to the supplementary material of this paper.

**Author Changes:** We have corrected the reference to equations in table S2. We have clarified which logit exponents were varied (and which were not) in the methodology section. We have also added a discussion of the other logit exponents, including why we did not vary them, how the default values were chosen, and the effect of changing those values on the outputs used in this study to the supplemental material:

"Table S2 provides information the gcamland nodes, the total land area for each node in 1990, and logit exponents used in this study. As noted in the main text, three of the logit exponents used in gcamland are varied as part of the analysis in this paper. For the remaining logit exponents (root, Pastureland, Grass/shrubs, Forestland), we use the default values used in GCAM. These values were chosen based on heuristics, where larger values are used for land types that are more substitutable. For the root, this is set to zero, as we do not allow conversion into or out of urban, tundra, or rock/ice/desert. For grass/shrubs, the decision to shift between grassland and shrubland is unlikely to be an economic choice; for this reason, we set the logit exponent to a very low value, effectively preserving the shares of grassland vs shrubland in the initial model year. Both the Forestland and Pastureland logit exponents govern substitution between commercial and non-commercial land types; a shift between these land types is not a land conversion (i.e., it does not require re-planting) but a shift in use (i.e., either moving livestock or engaging in logging activities). For this reason, higher logit exponents are chosen. A higher logit exponent governs Pastureland than Forestland as the shift in use of pastureland is likely to be easier than the change in use of Forestland. We have chosen not to vary these in this exercise as they do not directly impact the amount of cropland area, which was the output we focused on in the main text. We have done an additional sensitivity analysis quantifying the impact of these logits on cropland area. In this analysis, we doubled each logit one at a time. The area allocated to each crop changes by less than 1% (the largest change in magnitude is -0.12%). These logits do have a larger impact on other land types. For example, doubling the ForestLand logit results in a shift in the distribution of commercial and non-commercial forest, with commercial forest increasing by as much as 27%. However, total forest is largely unchanged (maximum change of 0.22%)."

Thanks for the additional sensitivity test. I appreciate your efforts. I also appreciate your note which says you have "not documented specifically how those numbers were chosen". Your response also says: "We have now added this to the supplementary material of this paper". I see that you tried to explain and justify the role of these parameters in explaining Table S2. But you ignored to clearly mention that those

values are ad hoc values. I checked Wise et al. (2014). This reference says nothing regarding the selected values under discussion. You have to clearly add the following phrase, or something similar, to the main manuscript not the supporting document:

"The logit values assigned to pastureland, grass/shrubs, and forest land nests have not been obtained from an explicit statistical approach nor a calibration process. They have been selected based on the authors' value judgment".

**Author Response:** We have added this information to the main text.

**Author Changes:** We have added this information to section 2.2.1: "The values of those four logit exponents have not been obtained from an explicit statistical analysis and instead were selected based on authors' judgment (see Section S1)."

Additional new comments on abstract:

1) The abstract says: "In this study, we demonstrate a more systematic and empirically-based approach to estimating model parameters for an economic model of land use and land cover change, gcamland".

   I propose the following modification as you only calibrate (not estimate) a few parameters assuming other model parameters are valid:

   "In this study, we demonstrate a more systematic and empirically based approach to calibrating a few selected parameters of an economic model of land use and land cover change, gcamland.

**Author Response:** We have revised the abstract to clarify that only a few key parameters were included. We have opted not to use the word "calibrate" as the editor noted that there was confusion in the previous draft between the internal model calibration in gcamland and the estimation of parameters using perturbed parameter analysis.

**Author Changes:** We have revised the abstract to state: "we demonstrate a more systematic and empirically-based approach to estimating *a few key* parameters…"

2) The abstract says: "we generate a large set of model parameter perturbations and run gcamland simulations with these parameter sets over the historical period in the United States to quantify land use and land cover, determine how well the model reproduces observations, and identify parameter combinations that best replicate observations".

   I propose the following modifications as you calibrate a small set of parameters assuming other parameters are valid. It is important to say what parameters you calibrated. Your paper is all about it, but the abstract says nothing about that.

   "we generate a large set of model parameter perturbations on the selected parameters and run gcamland simulations with these parameter sets over the historical period in the United States to quantify land use and land cover, determine how well the model reproduces observations, and identify parameter combinations that best replicate observations, assuming other model parameters are valid. In particular, 3 parameters out of 7 parameters that govern land allocation in gcanland were calibrated only for the case US.

**Author Response:** We have revised the abstract to clarify the parameters estimated. We have chosen to be more precise on the parameters noting that we estimated 6 parameters governing expectations and 3 of 7 logit exponents, instead of only referencing the logit exponents. We have chosen to do this as all 9 parameters influence land allocation in gcamland. We've decided to use 'fixed' instead of 'valid' as the word 'valid' could imply something beyond what is intended in this analysis.

**Author Changes:** We have revised the abstract to state: "Specifically, we generate a large set of model parameter perturbations *for the selected parameters* and run gcamland simulations with these parameter sets over the historical period in the United States to quantify land use and land cover, determine how well the model reproduces observations, and identify parameter combinations that best replicate observations, *assuming other model parameters are fixed*." We have also added the following sentence: "In particular, we estimate parameters for six parameters used in the formation of expectations and three of seven logit exponents for the USA only."